# Improved Algorithms for Neural Active Learning

Yikun Ban*, Yuheng Zhang*, Hanghang Tong, Arindam Banerjee, and Jingrui He

University of Illinois Urbana-Champaign
{yikunb2, yuhengz2, htong, arindamb, jingrui}@illinois.edu

## Abstract

We improve the theoretical and empirical performance of neural-network(NN)-based active learning algorithms for the non-parametric streaming setting. In particular, we introduce two regret metrics by minimizing the population loss that are more suitable in active learning than the one used in state-of-the-art (SOTA) related work. Then, the proposed algorithm leverages the powerful representation of NNs for both exploitation and exploration, has the query decision-maker tailored for $k$-class classification problems with the performance guarantee, utilizes the full feedback, and updates parameters in a more practical and efficient manner. These careful designs lead to an instance-dependent regret upper bound, roughly improving by a multiplicative factor $O(\log T)$ and removing the curse of input dimensionality. Furthermore, we show that the algorithm can achieve the same performance as the Bayes-optimal classifier in the long run under the hard-margin setting in classification problems. In the end, we use extensive experiments to evaluate the proposed algorithm and SOTA baselines, to show the improved empirical performance.

## 1 Introduction

The Neural Network (NN) is one of the indispensable paradigms in machine learning and is widely used in multifarious supervised-learning tasks [23]. As more and more complicated NNs are developed, the requirement of the training procedure on the labeled data grows, incurring significant cost of label annotation. Active learning investigates effective techniques on a much smaller labeled data set while attaining the comparable generalization performance to passive learning [19]. In this paper, we focus on the classification problem in the streaming setting of active learning with NN models. At every round, the learner receives an instance and is compelled to decide on-the-fly whether or not to observe the label associated with this instance. This problem seeks to maximize the generalization capability of learned NNs in a sequence of rounds, such that the model has robust performance on the unseen data from the same distribution [40].

In active learning, given access to the i.i.d. generated instances from a distribution $\mathcal{D}$, suppose there exist a class of functions $\mathcal{F}$ that formulate the mapping from instances to their labels. In the parametric setting, i.e., $\mathcal{F}$ has finite VC-dimension [25], existing works [24, 14, 7] have shown that the active learning algorithms can achieve the convergence rate of $\widetilde{\mathcal{O}}(1/\sqrt{N})$ to the best population loss in $\mathcal{F}$, where $N$ is the number of label queries. In the non-parametric setting, recent works [34, 35] provide the similar convergence results while suffering from the curse of input dimensionality. Unfortunately, most of NN-based approaches to active learning do not come with the performance guarantee, despite having powerful empirical results.

The first performance guarantee for neural active learning has been established in a recent work by [48], and the analysis is for over-parameterized neural networks with the assistance of Neural Tangent Kernel (NTK). We carefully investigate the limitations of [48], which turn into the main

---

* Both authors contribute equally.

36th Conference on Neural Information Processing Systems (NeurIPS 2022).

motivations of our paper. First, [48] transforms the classification problem into a multi-armed bandit problem [55], to minimize a pseudo regret metric. Yet, on the grounds that they seek to minimize the *conditional* population loss on a sequence of given data, it is dubious that the pseudo regret used in [48] can explicitly measure the generalization capability of given algorithms (see Remark 2.1). Second, the training process for NN models is not efficient, as [48] uses vanilla gradient descent and starts from randomly initialized parameters in every round. Third, although [48] removes the curse of input dimensionality $d$, the performance guarantee strongly suffers from another introduced term, the effective dimensionality $\widetilde{d}$, which can be thought of as the non-linear dimensionalities of Hilbert space spanned by NTK. In the worse case, the magnitude of $\widetilde{d}$ can be an unacceptably large number and thus the performance guarantee collapses.

## 1.1 Main contributions

In this paper, we propose a novel algorithm, I-NeurAL (**I**mproved Algorithms for **Neur**al **A**ctive **L**earning), to tackle the above limitations. Our contributions can be summarized as follows: (1) We consider the $k$-class classification problem, and we introduce two new regret metrics to minimize the population loss, which can directly reflect the generalization capability of NN-based algorithms. (2) I-NeurAL has a neural exploration strategy with a novel component to decide whether or not to query the label, coming with the performance guarantee. I-NeurAL exploits the full feedback in active learning which is a subtle but effective idea. (3) I-NeurAL is designed to support mini-batch Stochastic Gradient Descent (SGD). In particular, at every round, I-NeurAL does mini-batch SGD starting with the parameters of the last round, i.e., with warm start, which is more efficient and practical compared to [48]. (4) Without any noise assumption on the data distribution, we provide an instance-dependent performance guarantee of I-NeurAL for over-parameterized neural networks. Compared to [48], we remove the curse of both the input dimensionality $d$ and the effective dimensionality $\widetilde{d}$; Moreover, we roughly improve the regret by a multiplicative factor $\log(T)$, where $T$ is the number of rounds. (5) under a hard-margin assumption on the data distribution, we provide that NN models can achieve the same generalization capability as Bayes-optimal classifier after $\mathcal{O}(\log T)$ number of label queries; (6) we conduct extensive experiments on real-world data sets to demonstrate the improved performance of I-NeurAL over state-of-the-art baselines including the closest work [48] which has not provided empirical validation of their proposed algorithms.

## 1.2 Related Work

Active learning has been extensively studied and applied to many essential applications [44]. Bayesian active learning methods typically use a probabilistic regression model to estimate the improvement of each query [29, 41]. In spite of effectiveness on the small or moderate data sets, the Bayesian-based approaches are difficult to scale to large-scale data sets because of the batch sampling [43]. Another important class, margin algorithms or uncertainty sampling [33], obtains considerate performance improvement over passive learning and is further developed by many practitioners [20, 28, 37, 15]. Margin algorithms are flexible and can be adapted to both streaming and pool settings. In the pool setting, a line of works utilize the neural networks in active learning to improve the empirical performance [36, 42, 5, 17, 30, 45, 47, 54, 4]. However, they do not provide performance guarantee for NN-based active learning algorithms. From the theoretical perspective, [51, 21, 6, 8, 52] provide the performance guarantee with the specific classes of functions and [26, 22] present the theoretical analysis of active learning algorithms with the surrogate loss functions for binary classification. However, their performance guarantee is restricted within hypothesis classes, i.e, the parametric setting. In contrast, our goal is to derive an NN-based algorithm in the non-parametric setting that performs well both empirically and theoretically. Neural contextual bandits[55, 53, 13, 11, 12, 39] provide the principled method to balance between the exploitation and exploration [9, 10]. [48] transforms active learning into neural contextual bandit problem and obtains a performance guarantee, of which limitations are discussed above.

As [48] is the closest related work to our paper, we emphasize the differences of our techniques from [48] throughout the paper. We introduce the problem definition and proposed algorithms in Section 2 and Section 3 respectively. Then, we provide performance guarantees in Section 4 and empirical results in Section 5, ending with the conclusion in Section 6.

## 2   Problem Definition

In this paper, we study the streaming setting of active learning in the $k$-class classification problem. Let $\mathcal{X}$ denote the input space over $\mathbb{R}^d$, $\mathcal{Y} = \{1, 2, \ldots, k\}$ represent the label space, and $\mathcal{D}$ be some unknown distribution over $\mathcal{X} \times \mathcal{Y}$. At round $t \in [T] = \{1, 2, \ldots, T\}$, an instance $\mathbf{x}_t$ is drawn from the marginal distribution $\mathcal{D}_{\mathcal{X}}$ and accordingly $y_t$ is drawn from the conditional distribution $\mathcal{D}_{\mathcal{Y}|\mathbf{x}_t}$. Here, $y_t$ can be thought of as the index of the class that $\mathbf{x}_t$ belongs to. Inspired by [48], we first transform $\mathbf{x}_t$ into $k$ context vectors representing the $k$ classes respectively: $\mathbf{x}_{t,1} = (\mathbf{x}_t^\top, \mathbf{0}^\top, \ldots, \mathbf{0}^\top)^\top, \mathbf{x}_{t,2} = (\mathbf{0}^\top, \mathbf{x}_t^\top, \ldots, \mathbf{0}^\top)^\top, \ldots, \mathbf{x}_{t,k} = (\mathbf{0}^\top, \mathbf{0}^\top, \ldots, \mathbf{x}_t^\top)^\top$ and $\mathbf{x}_{t,i} \in \mathbb{R}^{dk}, \forall i \in [k]$. In accordance with context vectors, we construct the $k$ label vectors representing the $k$ possible prediction: $\mathbf{y}_{t,1} = (1, 0, \ldots, 0)^\top, \mathbf{y}_{t,2} = (0, 1, \ldots, 0)^\top, \ldots, \mathbf{y}_{t,k} = (0, 0, \ldots, 1)^\top$ and $\mathbf{y}_{t,i} \in \mathbb{R}^k, \forall i \in [k]$. Thus, $\mathbf{y}_{t,y_t}$ is the ground-truth label vector for $\mathbf{x}_t$.

Under the non-parametric setting of active learning, we define an unknown function $h$ to formulate the conditional distribution $\mathcal{D}_{\mathcal{Y}|\mathbf{x}_t} \colon \mathcal{X}^k \to [0, 1]$, such that

$$\forall i \in [k], \mathbb{P}(\mathbf{y}_{t,y_t} = \mathbf{y}_{t,i}|\mathbf{x}_t) = h(\mathbf{x}_{t,i}) , \tag{2.1}$$

which is subject to $\sum_{i=1}^k h(\mathbf{x}_{t,i}) = 1$. For simplicity, we consider the $k$-class classification problem with 0-1 loss. Given $\mathbf{x}_t$, i.e., $\mathbf{x}_{t,i}, i \in [k]$, let $\widehat{i}$ be the index of the class predicted by some hypothesis $f$ and thus $\mathbf{y}_{t,\widehat{i}}$ is the prediction. Then, we have the following loss:

$$L(\mathbf{y}_{t,\widehat{i}}, \mathbf{y}_{t,y_t}) = \mathbb{1}\{\mathbf{y}_{t,\widehat{i}} \neq \mathbf{y}_{t,y_t}\} \in \{0, 1\} . \tag{2.2}$$

where $\mathbb{1}$ is the indicator function.

Given the number of rounds $T$, at each round $t \in [T]$, the learner receives an instance $\mathbf{x}_t$ drawn i.i.d. from $\mathcal{D}_{\mathcal{X}}$. Then, the learner needs to make a prediction $\mathbf{y}_{t,\widehat{i}}$, and at the same time, decide on-the-fly whether or not to query the label $\mathbf{y}_{t,y_t}$ where $y_t$ is drawn i.i.d. from $\mathcal{D}_{\mathcal{Y}|\mathbf{x}_t}$. As the goal of active learning tasks is often to minimize the population loss [40], we introduce the following two regret metrics.

**Definition 2.1** (Latest Population Regret). *Given the data distribution $\mathcal{D}$, the number of rounds $T$, the Latest Population Regret is defined as*

$$R_T = \mathop{\mathbb{E}}_{\mathbf{x}_T \sim D_{\mathcal{X}}} \left[ \mathop{\mathbb{E}}_{y_T \sim \mathcal{D}_{\mathcal{Y}|\mathbf{x}_T}} [L(\mathbf{y}_{T,\widehat{i}}, \mathbf{y}_{T,y_T}) \mid \mathbf{x}_T] \right] - \mathop{\mathbb{E}}_{\mathbf{x}_T \sim D_{\mathcal{X}}} \left[ \mathop{\mathbb{E}}_{y_T \sim \mathcal{D}_{\mathcal{Y}|\mathbf{x}_T}} [L(\mathbf{y}_{T,i^*}, \mathbf{y}_{T,y_T}) \mid \mathbf{x}_T] \right] \tag{2.3}$$

*where $\mathbf{y}_{T,i^*}$ is the prediction the Bayes-optimal classifier would make on instance $\mathbf{x}_T$, i.e., $i^* = \arg\max_{i \in [k]} h(\mathbf{x}_{T,i})$ for $\mathbf{y}_{T,i^*}$.*

**Definition 2.2** (Cumulative Population Regret). *Given the data distribution $\mathcal{D}$, the number of rounds $T$, the Cumulative Population Regret is defined as:*

$$\mathbf{R}_T = \sum_{t=1}^T \left( \mathop{\mathbb{E}}_{\mathbf{x}_t \sim D_{\mathcal{X}}} \left[ \mathop{\mathbb{E}}_{y_t \sim \mathcal{D}_{\mathcal{Y}|\mathbf{x}_t}} [L(\mathbf{y}_{t,\widehat{i}}, \mathbf{y}_{t,y_t}) \mid \mathbf{x}_t] \right] - \mathop{\mathbb{E}}_{\mathbf{x}_t \sim \mathcal{D}_{\mathcal{X}}} \left[ \mathop{\mathbb{E}}_{y_t \sim \mathcal{D}_{\mathcal{Y}|\mathbf{x}_t}} [L(\mathbf{y}_{t,i^*}, \mathbf{y}_{t,y_t}) \mid \mathbf{x}_t] \right] \right) \tag{2.4}$$

*where $\mathbf{y}_{t,i^*}$ is the prediction the Bayes-optimal classifier would make on instance $\mathbf{x}_t$, i.e., $i^* = \arg\max_{i \in [k]} h(\mathbf{x}_{t,i})$ for $\mathbf{y}_{t,i^*}$.*

$R_T$ measures the performance at the last round $T$ only, and $\mathbf{R}_T$ measures the overall performance in $T$ rounds combined. Therefore, the goal of this problem is to minimize $R_T$ or $\mathbf{R}_T$, or both. At the same time, we also aim to minimize the following expected query cost:

$$\mathbf{N}_T = \sum_{t=1}^T \mathop{\mathbb{E}}_{\mathbf{x}_t \sim \mathcal{D}_{\mathcal{X}}} [\mathbf{I}_t \mid \mathbf{x}_t], \tag{2.5}$$

where $\mathbf{I}_t$ is the indicator of the query decision in round $t$ such that $\mathbf{I}_t = 1$ if $y_t$ is observed; $\mathbf{I}_t = 0$, otherwise.

**Remark 2.1.** Minimizing $R_T$ or $\mathbf{R}_T$ shows the generalization capability of the learned hypothesis on the distribution $\mathcal{D}$. However, the problem defined in [48] is to minimize the cumulative *conditional*

population regret as follows:

$$\widetilde{\mathbf{R}}_T = \sum_{t=1}^{T} \left( \underset{y_t \sim \mathcal{D}_{\mathcal{Y}|\mathbf{x}_t}}{\mathbb{E}} [L(\mathbf{y}_{t,\widehat{i}}, \mathbf{y}_{t,y_t})|\mathbf{x}_t] - \underset{y_t \sim \mathcal{D}_{\mathcal{Y}|\mathbf{x}_t}}{\mathbb{E}} [L(\mathbf{y}_{t,i^*}, \mathbf{y}_{t,y_t})|\mathbf{x}_t] \right). \qquad (2.6)$$

As $\mathbb{E}_{y_t \sim \mathcal{D}_{\mathcal{Y}|\mathbf{x}_t}}[L(\mathbf{y}_{t,\widehat{i}}, \mathbf{y}_{t,y_t})|\mathbf{x}_t]$ is the population loss conditioned on $\mathbf{x}_t$, unfortunately, $\widetilde{\mathbf{R}}_T$ only measures the performance of the learned hypothesis on the collected data $\{\mathbf{x}_t\}_{t=1}^{T}$, and $\widetilde{\mathbf{R}}_T$ cannot directly measure the accuracy of the hypothesis on unseen data instances. Although $\widetilde{\mathbf{R}}_T$ follows the regret definition in multi-armed bandits [55], it is fair to say that $\widetilde{\mathbf{R}}_T$ may not be a good metric in active learning.

## 3 Proposed Algorithms

In this section, we elaborate on the proposed algorithm I-NeurAL (Algorithm 1). In contrast to the directly comparable work [48], I-NeurAL has the following novel and advantageous aspects: (1) I-NeurAL incorporates a neural-based exploration strategy (Line 6) inspired by recent advances in bandits [13] to solve the exploitation-exploration dilemma in the decision for whether or not to query labels; (2) I-NeurAL includes a novel component (Line 11) to decide whether or not to query labels in the $k$-class classification problem; (3) I-NeurAL infers and exploits the feedback of all the contexts (Lines 12-17), instead of only utilizing the feedback of the chosen context in [48]; (4) I-NeurAL conducts mini-batch SGD based on the parameters of the last round (Algorithm 2), which is more practical, as opposed to conducting vanilla gradient descent from the initialization at every round in [48]. Next, we will present the details of I-NeurAL.

*Exploitation Network $f_1$.* Given $\mathbf{x}_{t,i}, i \in [k]$, to learn the unknown function $h$ (Eq. (2.1)), we use a fully-connected neural network $f_1$ with $L$-depth and $m$-width:

$$f_1(\mathbf{x}_{t,i}; \boldsymbol{\theta}^1) = \mathbf{W}_L^1 \sigma(\mathbf{W}_{L-1}^1 \sigma(\mathbf{W}_{L-2}^1 \ldots \sigma(\mathbf{W}_1^1 \mathbf{x}_{t,i}))), \qquad (3.1)$$

where $\mathbf{W}_1^1 \in \mathbb{R}^{m \times kd}, \mathbf{W}_l^1 \in \mathbb{R}^{m \times m}$, for $2 \leq l \leq L-1$, $\mathbf{W}_L^1 \in \mathbb{R}^{1 \times m}$, $\boldsymbol{\theta}^1 = [\text{vec}(\mathbf{W}_1^1)^\top, \ldots, \text{vec}(\mathbf{W}_L^1)^\top]^\top \in \mathbb{R}^{p_1}$, and $\sigma$ is the ReLU activation function $\sigma(\mathbf{x}) = \max\{0, \mathbf{x}\}$. In round $t$, given $\mathbf{x}_{t,i}, i \in [k]$, $f_1(\mathbf{x}_{t,i}; \boldsymbol{\theta}_{t-1}^1)$ is assigned to learn $h(\mathbf{x}_{t,i})$. Based on the fact $h(\mathbf{x}_{t,i}) = \underset{y_t \sim \mathcal{D}_{\mathcal{Y}|\mathbf{x}_t}}{\mathbb{E}} [1 - L(\mathbf{y}_{t,i}, \mathbf{y}_{t,y_t})]$, it is natural to regard $1 - L(\mathbf{y}_{t,i}, \mathbf{y}_{t,y_t})$ as the label for training $f_1$. Note that we take the basic fully-connected network as an example for the sake of analysis in over-parameterized networks and $f_1$ can be easily replaced with more complicated models depending on the tasks.

*Exploration Network $f_2$.* In addition to the network $f_1$, we assign another network $f_2$ to explore uncertain information contained in incoming instances. First, we carefully design the input of $f_2$ to incorporate the context vectors of the instance and the discrimination-ability of $f_1$, to learn the error between the Bayes-optimal probability $h(\mathbf{x}_{t,i})$ and the prediction $f_1(\mathbf{x}_{t,i}; \boldsymbol{\theta}^1)$.

**Definition 3.1** (Derivative-Context (DC) Embedding). *Given the exploitation network $f_1(\cdot; \boldsymbol{\theta}_{t-1}^1)$ and an input context $\mathbf{x}_{t,i}$, its DC embedding is defined as*

$$\phi(\mathbf{x}_{t,i}) = \left( \frac{vec \left( \nabla_{\mathbf{x}_{t,i}} f_1(\mathbf{x}_{t,i}; \boldsymbol{\theta}_{t-1}^1) \right)^\top}{\sqrt{2} \| \nabla_{\mathbf{x}_{t,i}} f_1(\mathbf{x}_{t,i}; \boldsymbol{\theta}_{t-1}^1) \|_2}, \frac{\mathbf{x}_{t,i}^\top}{\sqrt{2}} \right) \in \mathbb{R}^{2dk}, \qquad (3.2)$$

*where $\nabla_{\mathbf{x}_{t,i}} f_1$ is the partial derivative of $f_1(\mathbf{x}_{t,i}; \boldsymbol{\theta}_{t-1}^1)$ with respect to $\mathbf{x}_{t,i}$.*

$\phi(\mathbf{x}_{t,i})$ is normalized so that $\|\phi(\mathbf{x}_{t,i})\|_2 = 1$. Note that the input for $f_2$ in [13] is the gradient with respect to $\theta_1$, denoted by $\nabla_{\theta_1} f_1(\mathbf{x}_{t,i}; \boldsymbol{\theta}_{t-1}^1) \in \mathbb{R}^{p_1}$. Its dimensionality is much larger than $\nabla_{\mathbf{x}_{t,i}} f_1(\mathbf{x}_{t,i}; \boldsymbol{\theta}_{t-1}^1)$ in Definition 3.1, may causing significant computation cost.

Given the input $\phi(\mathbf{x}_{t,i})$, similarly, we choose the fully-connected network to build $f_2$:

$$f_2(\phi(\mathbf{x}_{t,i}); \boldsymbol{\theta}^2) = \mathbf{W}_L^2 \sigma(\mathbf{W}_{L-1}^2 \sigma(\mathbf{W}_{L-2}^2 \ldots \sigma(\mathbf{W}_1^2 \phi(\mathbf{x}_{t,i})))), \qquad (3.3)$$

where $\mathbf{W}_1^2 \in \mathbb{R}^{m \times 2kd}, \mathbf{W}_l^2 \in \mathbb{R}^{m \times m}$, for $2 \leq l \leq L-1$, $\mathbf{W}_L^2 = \mathbb{R}^{1 \times m}$ and $\boldsymbol{\theta}^2 = [\text{vec}(\mathbf{W}_1^2)^\top, \ldots, \text{vec}(\mathbf{W}_L^2)^\top]^\top \in \mathbb{R}^{p_2}$. In round $t$, given $\mathbf{x}_{t,i}, \forall i \in [k]$, $f_2$ is to predict $h(\mathbf{x}_{t,i}) -$

**Algorithm 1** I-NeurAL

**Input:** $T$ (number of rounds) $f_1, f_2$ (neural networks), $\eta_1, \eta_2$ (learning rate), $\gamma$ (exploration parameter), $b$ (batch size), $\delta$ (confidence level)

1: Initialize $\boldsymbol{\theta}_0^1, \boldsymbol{\theta}_0^2; \widehat{\boldsymbol{\theta}}_0^1 = \boldsymbol{\theta}_0^1; \widehat{\boldsymbol{\theta}}_0^2 = \boldsymbol{\theta}_0^2$
2: $\mathcal{H}_0^1 = \emptyset; \mathcal{H}_0^2 = \emptyset$
3: **for** $t = 1, 2, \ldots, T$ **do**
4:     Observe instance $\mathbf{x}_t \in \mathbb{R}^d$ and build $\mathbf{x}_{t,i}, \forall i \in [k]$
5:     **for** each $i \in [k]$ **do**
6:         $f(\mathbf{x}_{t,i}; \boldsymbol{\theta}_{t-1}) = \Big( \underbrace{f_1(\mathbf{x}_{t,i}; \boldsymbol{\theta}_{t-1}^1)}_{\text{Exploitation Score}} + \underbrace{f_2(\phi(\mathbf{x}_{t,i}); \boldsymbol{\theta}_{t-1}^2)}_{\text{Exploration Score}} \Big)$
7:     **end for**
8:     $\widehat{i} = \arg\max_{i \in [k]} f(\mathbf{x}_{t,i}; \boldsymbol{\theta}_{t-1})$
9:     $i^\circ = \arg\max_{i \in ([k] \setminus \{\widehat{i}\})} f(\mathbf{x}_{t,i}; \boldsymbol{\theta}_{t-1})$
10:     Predict $\mathbf{y}_{t,\widehat{i}}$
11:     $\mathbf{I}_t = \mathbb{1}\{|f(\mathbf{x}_{t,\widehat{i}}; \boldsymbol{\theta}_{t-1}) - f(\mathbf{x}_{t,i^\circ}; \boldsymbol{\theta}_{t-1})| < 2\gamma\beta_t\} \in \{0, 1\}; \beta_t = \sqrt{\frac{2c_1}{t}} + \left(\frac{c_2 3L}{\sqrt{2t}}\right) + \sqrt{\frac{2\log(c_3 Tk)/\delta)}{t}}$
12:     **if** $\mathbf{I}_t = 1$ **then**
13:         Query $\mathbf{x}_t$ and observe $y_t$
14:         **for** $i \in [k]$ **do**
15:             $r_{t,i}^1 = 1 - L(\mathbf{y}_{t,i}, \mathbf{y}_{t,y_t})$ (defined in E.q. (2.2))
16:             $r_{t,i}^2 = r_{t,i}^1 - f_1(\mathbf{x}_{t,i}; \boldsymbol{\theta}_{t-1}^1)$
17:         **end for**
18:     **else**
19:         **for** $i \in [k]$ **do**
20:             $r_{t,i}^1 = 1 - L(\mathbf{y}_{t,i}, \mathbf{y}_{t,\widehat{i}})$
21:             $r_{t,i}^2 = r_{t,i}^1 - f_1(\mathbf{x}_{t,i}; \boldsymbol{\theta}_{t-1}^1)$
22:         **end for**
23:     **end if**
24:     $\mathcal{H}_t^1 = \mathcal{H}_{t-1}^1 \cup \{(\mathbf{x}_{t,i}, r_{t,i}^1), i \in [k]\}$
25:     $\mathcal{H}_t^2 = \mathcal{H}_{t-1}^2 \cup \{(\mathbf{x}_{t,i}, r_{t,i}^2), i \in [k]\}$
26:     $\boldsymbol{\theta}_t^1, \boldsymbol{\theta}_t^2 = $ Mini-Batch-SGD-Warm-Start ( $f_1, f_2, \mathcal{H}_t^1, \mathcal{H}_t^2, b$)
27: **end for**
28: **Return** $(\boldsymbol{\theta}^1, \boldsymbol{\theta}^2)$ uniformly from $((\boldsymbol{\theta}_0^1, \boldsymbol{\theta}_0^2), \ldots, \boldsymbol{\theta}_{T-1}^1, \boldsymbol{\theta}_{T-1}^2)$

---

$f_1(\mathbf{x}_{t,i}; \boldsymbol{\theta}_{t-1}^1)$ for exploration. Because $h(\mathbf{x}_{t,i}) - f_1(\mathbf{x}_{t,i}; \boldsymbol{\theta}_{t-1}^1) = \mathbb{E}_{y_t \sim \mathcal{D}_{\mathcal{Y}|\mathbf{x}_t}}[1 - L(\mathbf{y}_{t,i}, \mathbf{y}_{t,y_t}) - f_1(\mathbf{x}_{t,i}; \boldsymbol{\theta}_{t-1}^1)]$, we regard $1 - L(\mathbf{y}_{t,i}, \mathbf{y}_{t,y_t}) - f_1(\mathbf{x}_{t,i}; \boldsymbol{\theta}_{t-1}^1)$ as the label for training $f_2$.

To sum up, in round $t$, given $\mathbf{x}_{t,i}, \forall i \in [k]$, the prediction $\widehat{i}$ ($\mathbf{y}_{t,\widehat{i}}$) is made based on the sum of exploitation and exploration scores, i.e., $f_1(\mathbf{x}_{t,i}; \boldsymbol{\theta}_{t-1}^1) + f_2(\phi(\mathbf{x}_{t,i}); \boldsymbol{\theta}_{t-1}^2)$ (Lines 5-10).

*Query Decision-maker (Line 11).* A label query is made when I-NeurAL is not confident enough to discriminate the Bayes-optimal class from other classes. $2\gamma\beta_t$ ($\beta_t$ is also defined in Lemma 7.3) can be thought of as a confidence interval for the distance between the optimal class and second optimal class, where $\gamma$ is the hyper-parameter to tune the sensitivity of the decision-maker in practice. Given any $\gamma \geq 1, \delta \in (0, 1)$, with probability at least $1 - \delta$, based on our analysis (Lemma 7.5), $\mathbb{E}_{(\mathbf{x}_t, y_t) \sim \mathcal{D}}[L(\mathbf{y}_{t,\widehat{i}}, \mathbf{y}_{t,y_t})] = \mathbb{E}_{(\mathbf{x}_t, y_t) \sim \mathcal{D}}[L(\mathbf{y}_{t,i^*}, \mathbf{y}_{t,y_t})]$ when $\mathbf{I}_t = 0$, i.e., I-NeurAL suffers no regret. Thus, we use $\mathbf{y}_{t,\widehat{i}}$ as the pseudo-label in this case and we have the following update rules.

*Utilize Full Feedback (Lines 14-25).* Different from the bandit setting where the learner can only observe the reward of the selected context, we can infer the rewards of all contexts in active learning, as we know the specific class of the current instance. Thus, for each $\mathbf{x}_{t,i}, i \in [k], r_{t,i}^1 = 1 - \mathcal{L}(\mathbf{y}_{t,i}, \mathbf{y}_{t,y_t})$ is regarded as the "reward" of $\mathbf{x}_{t,i}$, predicted by $f_1$, and $r_{t,i}^2 = r_{t,i}^1 - f_1(\mathbf{x}_{t,i}; \boldsymbol{\theta}^1)$ is regarded as the "residual reward" of $\mathbf{x}_{t,i}$, predicted by $f_2$. In summary, in round $t$, when $\mathbf{I}_t = 1$, $\mathbf{y}_{t,y_t}$ is observed to

---

**Algorithm 2** Mini-Batch-SGD-Warm-Start ( $f_1, f_2, \mathcal{H}_t^1, \mathcal{H}_t^2, b$ )

---

1: Define $\mathcal{L}_1[(\mathbf{x}, r^1); \boldsymbol{\theta}^1] = (r^1 - f_1(\mathbf{x}; \boldsymbol{\theta}^1))^2/2$
2: Uniformly draw a set $\widehat{\mathcal{H}}_t^1 \subset \mathcal{H}_t^1, s.t., |\widehat{\mathcal{H}}_t^1| = b$
3: $\widehat{\boldsymbol{\theta}}_t^1 = \widehat{\boldsymbol{\theta}}_{t-1}^1 - \frac{\eta_1}{b} \sum\limits_{(\mathbf{x}, r^1) \in \widehat{\mathcal{H}}_t^1} \nabla_{\boldsymbol{\theta}^1} \mathcal{L}_1[(\mathbf{x}, r^1); \widehat{\boldsymbol{\theta}}_{t-1}^1]$
4: Define $\mathcal{L}_2[(\phi(\mathbf{x}), r^2); \boldsymbol{\theta}^2] = (r^2 - f_2(\phi(\mathbf{x}); \boldsymbol{\theta}^2))^2/2$
5: Uniformly draw a set $\widehat{\mathcal{H}}_t^2 \subset \mathcal{H}_t^2, s.t., |\widehat{\mathcal{H}}_t^2| = b$
6: $\widehat{\boldsymbol{\theta}}_t^2 = \widehat{\boldsymbol{\theta}}_{t-1}^2 - \frac{\eta_2}{b} \sum\limits_{(\phi(\mathbf{x}), r^1) \in \widehat{\mathcal{H}}_t^2} \nabla_{\boldsymbol{\theta}^2} \mathcal{L}_2[(\phi(\mathbf{x}), r^2); \widehat{\boldsymbol{\theta}}_{t-1}^2]$
7: $\Omega_t = \Omega_{t-1} \cup \{(\widehat{\boldsymbol{\theta}}_t^1, \widehat{\boldsymbol{\theta}}_t^2)\}$
8: **Return** $(\boldsymbol{\theta}_t^1, \boldsymbol{\theta}_t^2)$ uniformly from $\Omega_t$

---

update $r_{t,i}^1$ and $r_{t,i}^2$; when $\mathbf{I}_t = 0$, $\mathbf{y}_{t,\widehat{i}}$ is regard as the pseudo-label to obtain $r_{t,i}^1$ and $r_{t,i}^2$, $\forall i \in [k]$. Therefore, we have the training data $\mathcal{H}_t^1$ for $f_1$ and $\mathcal{H}_t^2$ for $f_2$.

*Mini-Batch SGD with Warm-Start (Algorithm 3).* Unlike [48] that uses vanilla gradient descent from randomly initialized parameters in each round, causing unnecessarily expensive computation, we extend the training procedure to mini-batch SGD with warm start, i.e., we incrementally train the parameters $\boldsymbol{\theta}_t$ starting from the parameters of the last round $\boldsymbol{\theta}_{t-1}$ in each round $t$.

Algorithm 1 depicts the workflow of I-NeurAL. Lines 1-2 initialize the parameters where each entry of $\mathbf{W}_l$ is drawn from the normal distribution $\mathcal{N}(0, 2/m)$ and each entry of $\mathbf{W}_L$ is drawn from $\mathcal{N}(0, 1/m)$ for both $f_1$ and $f_2$. $\mathcal{H}_0^1, \mathcal{H}_0^2$ store the historical data for $f_1$ and $f_2$ respectively. In each round $t$, Line 4 builds the $k$ contexts for the observed instance $\mathbf{x}_t$, and Lines 5-7 calculate the exploitation-exploration score for each context. $\widehat{i}$ (Line 8) is the index of the optimal-predicted class and thus $\mathbf{y}_{t,\widehat{i}}$ is the prediction. $i^\circ$ (Line 9) is the index of the second optimal-predicted class, which is used to decide whether to make a query. Line 11 is our decision component. When $\mathbf{I_t} = 1$, it shows that we are not confident enough about our prediction, so that we make a query for $\mathbf{x}_t$ and observe the rewards for each context (Lines 12-17). When $\mathbf{I_t} = 0$, based on our analysis, with high confidence, the prediction $\mathbf{y}_{t,\widehat{i}}$ matches the one predicted by the Bayes-optimal classifier. Hence, we consider $\mathbf{y}_{t,\widehat{i}}$ as the label and observe the reward for all contexts (Lines 18-23). In the end, we update the networks $f_1$ and $f_2$, based on the collected data (Lines 24-26).

## 4 Regret Analysis

In this section, we provide the regret analysis of I-NeurAL in the over-parameterized neural networks. First, we need the standard normalization restricted to the input instances.

**Assumption 4.1.** For any $t \in [T]$, $\|\mathbf{x}_t\|_2 = 1$.

Inspired by [16], we define the following function class. Given a constant $\nu > 0$, we define the following $\nu$-ball of $\boldsymbol{\theta}^2$ around the random initialization: $\mathcal{B}(\boldsymbol{\theta}_0^2, \nu) = \{\widetilde{\boldsymbol{\theta}}^2 : \|\widetilde{\boldsymbol{\theta}}^2 - \boldsymbol{\theta}_0^2\|_2 \leq \mathcal{O}(\frac{\nu}{\sqrt{m}})\}$. Recall that $r_{t,\widehat{i}}^2 = r_{t,\widehat{i}}^1 - f_1(\mathbf{x}_{t,\widehat{i}}; \widehat{\boldsymbol{\theta}}_{t-1}^1)$. Let $\widehat{\boldsymbol{\theta}}_{t-1}^{1,*}$ represent the parameters trained on $\mathcal{H}_{t-1}^{1,*}$ using Algorithm 2 with the Bayes-optimal classifier, where $\mathcal{H}_{t-1}^{1,*} = \{\mathbf{x}_{\tau,i^*}, r_{\tau,i^*}^1\}_{\tau=1}^{t-1}$ are the historical Bayes-optimal pairs. We define $r_{t,i^*}^{2,*} = r_{t,i^*}^1 - f_1(\mathbf{x}_{t,i^*}; \widehat{\boldsymbol{\theta}}_{t-1}^{1,*})$. Then, we provide the following regret bound that depends on the classification ability of exploration network class induced by $\mathcal{B}(\boldsymbol{\theta}_0^2, \nu)$.

**Theorem 4.1.** *Given the number of rounds $T$, for any $\delta \in (0,1), \gamma > 1, \nu > 0$, suppose $m \geq \widetilde{\Omega}(poly(T, k, L, \nu)), \eta_1 = \eta_2 = \Theta(\frac{\kappa \nu}{\sqrt{T}m})$,*
$$\inf_{\widetilde{\boldsymbol{\theta}}^2 \in \mathcal{B}(\boldsymbol{\theta}_0^2, \nu)} \sum_{t=1}^T \left( f_2(\phi(\mathbf{x}_{t,\widehat{i}}); \widetilde{\boldsymbol{\theta}}^2) - r_{t,\widehat{i}}^2 \right)^2 \& \inf_{\widetilde{\boldsymbol{\theta}}^2 \in \mathcal{B}(\boldsymbol{\theta}_0^2, \nu)} \sum_{t=1}^T \left( f_2(\phi(\mathbf{x}_{t,i^*}); \widetilde{\boldsymbol{\theta}}^2) - r_{t,i^*}^{2,*} \right)^2 \leq \mu.$$
*Then, with probability at least $1 - \delta$ over the initialization of $\boldsymbol{\theta}_0^1, \boldsymbol{\theta}_0^2$, there exist a small enough*

*constant $\kappa$, such that Algorithm 1 achieves the following regret bound:*

$$\mathbf{R}_T \le \mathcal{O}\left(2\sqrt{T} - 1\right)\left[\frac{6L\nu + 4\sqrt{\mu}}{\sqrt{2}} + 2\sqrt{2\log(\mathcal{O}(Tk)/\delta)} + \mathcal{O}(1)\right] \tag{4.1}$$

*and at the same time $\mathbf{N}_T \le \mathcal{O}(T)$. Suppose the Bayes-optimal classifier has zero classification errors, i.e., $L(\mathbf{y}_{t,i^*}, \mathbf{y}_{t,y_t}) = 0, t \in [T]$. It holds that*

$$R_T \le \mathcal{O}\left(\frac{6L\nu + 4\sqrt{\mu}}{\sqrt{2T}}\right) + 2\sqrt{\frac{2\log(\mathcal{O}(Tk)/\delta)}{T}}. \tag{4.2}$$

Theorem 4.1 provides the regret bound of I-NeurAL for $R_T$ and $\mathbf{R}_T$ respectively, $R_T \le \mathcal{O}(\frac{\sqrt{\log T}}{\sqrt{T}})$ and $\mathbf{R}_T \le \mathcal{O}(\sqrt{T \log T})$. As [48] only provides the regret bound for $\widetilde{R}_T$, to show the advantages of I-NeurAL, we also provide the following lemma for fair comparison.

**Lemma 4.1.** *Given the number of rounds $T$, for any $\delta \in (0,1), \gamma > 1, \nu > 0$, suppose $m \ge \widetilde{\Omega}(poly(T, k, L, \nu)), \eta_1 = \eta_2 = \Theta(\frac{\kappa\nu}{\sqrt{T}m})$, and $\mu$ satisfies the conditions in Theorem 4.1. Then, with probability at least $1 - \delta$ over the initialization of $\boldsymbol{\theta}_0^1, \boldsymbol{\theta}_0^2$, these exists a small enough constant $\kappa$, such that Algorithm 1 can achieve the following regret bound:*

$$\widetilde{\mathbf{R}}_T \le \mathcal{O}\left(\frac{6L\nu + 4\sqrt{\mu}}{\sqrt{2}}\right)\sqrt{T} + 2\sqrt{2T\log(\mathcal{O}(T)/\delta)} + \mathcal{O}(1) \tag{4.3}$$

*and at the same time $\mathbf{N}_T \le \mathcal{O}(T)$.*

**Comparison with [48].** Lemma 4.1 shows that I-NeurAL can achieve the regret bound of same complexity for $\widetilde{\mathbf{R}}_T$ as $\mathbf{R}_T$. Under the same assumption in the over-parameterized neural networks, without any assumption on $\mathcal{D}$, Theorem 1 in [48] (i.e., the lower-noise condition with exponent $\alpha = 0$, and $k = 2$ is ignored in the binary classification) achieves the following regret bound: $\widetilde{\mathbf{R}}_T \le \mathcal{O}(\log\det(I + \mathbf{H})\sqrt{T(\log\det(I + \mathbf{H}) + S^2)})$ where $\mathbf{H}$ is the NTK matrix [27, 3] formed by received instances of all $T$ rounds, $S = \sqrt{\mathbf{h}^\top \mathbf{H}^{-1}\mathbf{h}}$ is a complexity term, and $\mathbf{h} = (h(\mathbf{x}_{1,\widehat{i}}), \dots, h(\mathbf{x}_{T,\widehat{i}}))^\top \in \mathbb{R}^T$. Note that I-NeurAL and [48] have the same trivial label complexity $\mathcal{O}(T)$ in this difficult case. According to the definition of effective dimension $\widetilde{d}$ in [55], the above regret bound obtained by [48] can be represented by:

$$\widetilde{\mathbf{R}}_T \le \mathcal{O}(\widetilde{d}\log(1 + T))\sqrt{T(\widetilde{d}\log(1 + T) + S^2)} \quad \text{and} \quad \widetilde{d} = \frac{\log\det(I + \mathbf{H})}{\log(1 + T)} \tag{4.4}$$

**Remark 4.1.** The instance-dependent complexity term $\mu$ reflects the possible minimal regression error on the data instances caused by the functions induced by $\mathcal{B}(\boldsymbol{\theta}_0^2, \nu)$ controlled by $\nu$. Such complexity term is first introduced in [16]. When $\nu$ is small, the corresponding ball $\mathcal{B}(\boldsymbol{\theta}_0^2, \nu)$ is small, so $\mu$ tends to be large; Otherwise, when $\nu$ is large, $\mu$ tends to be small. In particular, when setting $\nu = \mathcal{O}(1)$, Theorem 4.1 and Lemma 4.1 suggests that if the data can be learned by a function in the function class formed by $\mathcal{B}(\boldsymbol{\theta}_0^2, \mathcal{O}(1))$ with the small training error, then I-NeurAL will have the regret with order $\widetilde{\mathcal{O}}(\sqrt{T})$. Note that [48] has the complexity term $S$ as well, to reflect the boundary of optimal parameters specific to the data.

**Remark 4.2.** Theorem 4.1 and Lemma 4.1 do not depend on $\widetilde{d}$. The effective dimension $\widetilde{d}$ was first introduced in [46] and then used in [55], which can be thought of as the non-linear dimensionalities in the NTK kernel space. However, $\widetilde{d}$ can be $p = m + mkd + m^2(L-1)$ in the worst case, i.e., $\widetilde{d} \gg T$ (see details in Appendix 9). Eq.(4.4) has the term $\mathcal{O}(\widetilde{d})$ and thus the regret bound obtained by [48] can explode due to $\widetilde{d}$. This is because the analysis of [48] closely depends on NTK, i.e., to apply Confidence Ellipsoid bound (Theorem 2 in [1]) to the NTK approximation. This procedure inevitably bind their regret bound to the determinant of NTK that can have a very large magnitude. In contrast, Eq.(4.3) does not have the term $\widetilde{d}$, because our analysis does not depend on the NTK approximation and I-NeurAL directly utilizes the property of over-parameterized neural networks, i.e., the convergence error $\mu$ and the generalization concentration bound (Lemma 7.6). These two terms are independent of $\widetilde{d}$, which paves the way for I-NeurAL to remove the curse of $\widetilde{d}$.

**Remark 4.3.** Theorem 4.1 and Lemma 4.1 improve the regret by a multiplicative factor $\mathcal{O}(\log T)$ over [48]. Note that the analysis of [48] is built for binary classification and thus $k = 2$ in Theorem 4.1 and Lemma 4.1. This improvement stems from the different analysis workflow of I-NeurAL from [48]. Again, our analysis does not rely on NTK approximation and it is built on the convergence and generalization bound of wide neural networks.

**Remark 4.4.** Our proof workflow of Theorem 4.1 and Lemma 4.1 is inspired by [13]. Compared to [13], we provide the first regret bound supporting mini-batch SGD with warm-start and a more generic generalization bound (Lemma 7.3) that holds for every arm (class). Moreover, we carry out the performance analysis of query decision-maker (Lemma 7.5), which is a new addition.

For the label complexity, $\mathbf{N}_T$ has the trivial $\mathcal{O}(T)$ complexity which is the same as Theorem 1 in [48] (with the exponent $\alpha = 0$). Because we have to consider the worst case where the unique Bayes-optimal class does not exist, i.e., given $\mathbf{x}_{t,i}, i \in [k]$, there does not exist $i^*$ such that $h(\mathbf{x}_{t,i^*}) > h(\mathbf{x}_{t,i}), \forall i \in [k] \setminus \{i^*\}$. Therefore, we provide the following analysis and show that $\mathbf{R}_T$ and $\mathbf{N}_T$ can be upper bounded by constants as long as there exists a unique Bayes-optimal class for the input instances, described by the following mild margin assumption.

**Assumption 4.2** ($\epsilon$-margin). In round $t \in [T]$, given an instance $\mathbf{x}_t$ and the label $y_t$, then $\mathbf{x}_t$ has the $\epsilon$-Unique optimal class if there exists $\epsilon > 0$ such that

$$\mathbb{P}(\mathbf{y}_{t,y_t} = \mathbf{y}_{t,i^*}|\mathbf{x}_t) - \mathbb{P}(\mathbf{y}_{t,y_t} = \mathbf{y}_{t,i^\circ}|\mathbf{x}_t) \geq \epsilon, \tag{4.5}$$

where $i^* = \arg\max_{i \in [k]} h(\mathbf{x}_{t,i})$ is the Bayes-optimal class and $i^\circ = \arg\max_{i \in ([k] \setminus \{i^*\})} h(\mathbf{x}_{t,i})$ is the second Bayes-optimal class.

Given any $i \in [k]$, let $i$ be a fixed index, i.e., suppose there exist a policy $\pi_i$ which always select the $i$-th context $(\mathbf{x}_{t,i}, r_{t,i}^1)$ for every round $t \in [T]$. Then, in round $t$, we have the collected data by $\Omega_i$: $\mathcal{H}_{t-1}^{1,i} = \{\mathbf{x}_{\tau,i}, r_{\tau,i}^1\}_{\tau=1}^{t-1}$. Then, let $\widehat{\boldsymbol{\theta}}_{t-1}^{1,i}$ represent the parameters trained only on $\mathcal{H}_{t-1}^{1,i}$ using Algorithm 2 with $\pi_i$ and $r_{t,i}^{2,i} = r_{t,i}^1 - f_1(\mathbf{x}_{t,i}; \widehat{\boldsymbol{\theta}}_{t-1}^{1,i})$.

**Theorem 4.2.** *Suppose the instances that are drawn from $\mathcal{D}$ satisfy Assumption 4.2. Then, given the number of rounds $T$, for any $\delta \in (0,1), \gamma > 1, \epsilon \in (0,1), \nu > 0$, suppose $m \geq \widetilde{\Omega}(poly(T,k,L,\nu)), \eta_1 = \eta_2 = \Theta(\frac{\nu\kappa}{\sqrt{Tm}})$, and $\mu$ satisfies the conditions in Theorem 4.1 and $\max_{i \in [k]} \left\{ \inf_{\widetilde{\boldsymbol{\theta}}^2 \in \mathcal{B}(\boldsymbol{\theta}_0^2, \nu)} \sum_{t=1}^{T} \left( f_2\left(\mathbf{x}_{t,\widehat{i}}; \widetilde{\boldsymbol{\theta}}^2\right) - r_{t,i}^{2,i} \right)^2 \right\} \leq \mu$. Then, with probability at least $1 - \delta$ over the initialization of $\boldsymbol{\theta}_0^1, \boldsymbol{\theta}_0^2$, there exists a small enough constant $\kappa$, such that Algorithm 1 achieves the following regret bound:*

$$\mathbf{R}_T \leq (2\sqrt{\bar{\mathcal{T}}} - 1)\left[ \mathcal{O}\left(\frac{6L\nu + 4\sqrt{\mu}}{\sqrt{2}}\right) + \sqrt{2\log(\mathcal{O}(Tk)/\delta)} \right] \quad \mathbf{N}_T \leq \bar{\mathcal{T}} \tag{4.6}$$

*where $\bar{\mathcal{T}} = \frac{12(\gamma+1)^2 \cdot [2\mu + 9L^2\nu^2 C_1^2 + 2\log(C_2 Tk/\delta)]}{\epsilon^2}$. Suppose the Bayes-optimal classifier has zero classification errors, i.e., $L(\mathbf{y}_{t,i^*}, \mathbf{y}_{t,y_t}) = 0, t \in [T]$. it holds that*

$$\begin{cases} R_T \leq \mathcal{O}\left(\frac{6L\nu + 4\sqrt{\mu}}{\sqrt{2T}}\right) + 2\sqrt{\frac{2\log(\mathcal{O}(Tk)/\delta)}{T}}, & \text{if } T \leq \bar{\mathcal{T}}; \\ R_T = 0, \text{else}. \end{cases} \tag{4.7}$$

**Remark 4.5.** Theorem 4.2 provides the upper bound for $\mathbf{R}_T$ with order of $\mathcal{O}(\log T)$. When other parameters are fixed, this indicates $\mathbf{R}_T$ is upper bounded by $\mathcal{O}(\log T)$. Moreover, the analysis of $R_T$ indicates that I-NeurAL can achieve the same performance as Bayes-optimal classifier with high confidence after $\mathcal{O}(\log T)$ number of rounds (i.e. $T > \bar{\mathcal{T}}$). In Theorem 1 of [48] ( with the exponent $\alpha \to +\infty$ equivalent to Assumption 4.2), $\widetilde{\mathbf{R}}_T \leq \mathcal{O}(\widetilde{d}\log(1 + T))\sqrt{(\widetilde{d}\log(1 + T) + S^2)}$ that still is dependent on $\widetilde{d}$ because NTK depends on $\widetilde{d}$.

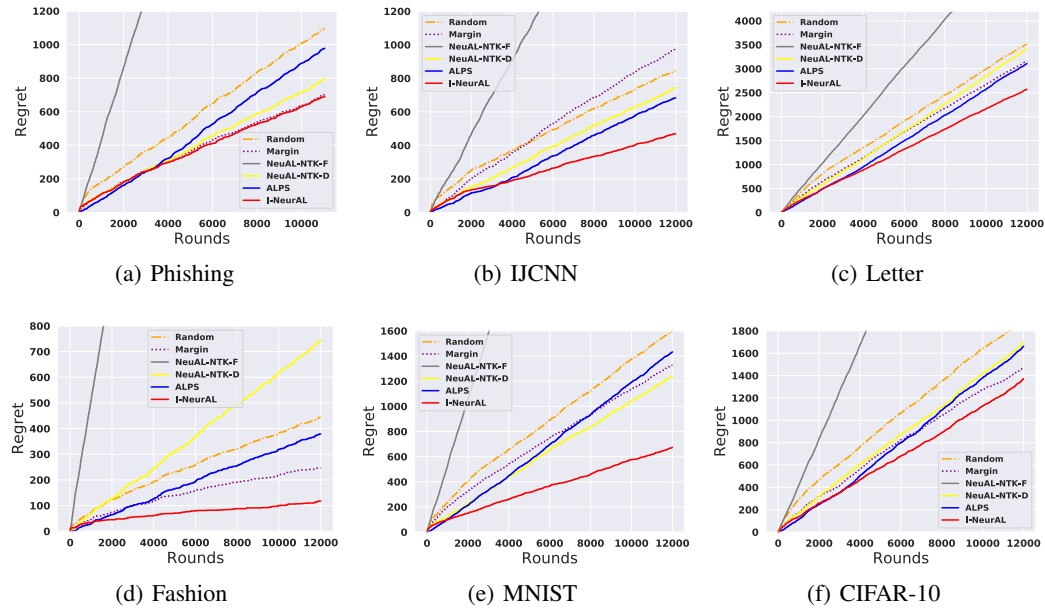

Figure 1: Regret comparison on six data sets. I-NeurAL outperforms all baselines.

## 5 Experiments

In this section, we evaluate I-NeurAL on public classification data sets compared with state-of-the-art (SOTA) baselines. Due to the space limit, we only report the main results here and leave the implementation details and parameter sensitivity in the Appendix 10. Codes are available[1].

We report the experimental results on the following six data sets: Phishing[2], IJCNN [38], Letter [18], Fashion [49], MNIST [32] and CIFAR-10 [31]. In each round, one instance is randomly drawn from the data set and the algorithm is compelled to make prediction on it. Then, the regret is 1 if the prediction does not match the label; the regret is 0, otherwise. At the same time, if the algorithm decides to observe the label, it costs one query budget. As the algorithm may abusively make label queries, we restrict the query budget to 3% of the total number of instances in the data set for fair comparison.

The compared baselines are described as follows. (1) **Random**: The NN classifier queries the label with a fixed probability $p$ until the query budget is exhausted; (2) **Margin**: The NN classifier queries the label when the predicted probability is lower than a threshold. These two baselines are used in [22]. (3) **NeuAL-NTK-F (Algorithm 1 in [48]**: This model makes predictions based on the frozen NTK approximation coming with an Upper-Confidence-Bound(UCB)-based exploration strategy. (4) **NeuAL-NTK-D (Algorithm 3 in [48]**): The prediction is made based on the NN classifier with a UCB while the NTK is updated accordingly. (5) **ALPS [22]**: Given a class of pre-trained hypotheses, the hypothesis minimizing the logistic loss of labeled and pseudo-labeled data is chosen to make predictions and the label query is based on the disagreement of different hypotheses.

**Results**. The regret comparison on six data sets is shown in Table 1 and Figure 1. I-NeurAL consistently outperforms all baselines across all data sets. In particular, I-NeurAL surpasses the best baseline by 31.3%, 45.6%, 52.2% on IJCNN, MNIST, Fashion respectively. Since NeuAL-NTK-F uses frozen NTK approximation, the new knowledge of each round is barely utilized by the neural network and thus it turns into the worst baseline. NeuAL-NTK-D updates the network parameters with gradient descent and queries the label based on the uncertainty estimation. However, its upper confidence bound is still based on the confidence ellipsoid. Instead, I-NeurAL leverages the representation power of neural networks for both exploitation and exploration. ALPS maintains a class of pre-trained hypotheses and tries to make the best decisions based on these hypotheses. Nevertheless, the model parameters are fixed before the online active learning process. Hence, ALPS

---

[1] https://github.com/matouk98/I-NeurAL
[2] https://www.csie.ntu.edu.tw/~cjlin/libsvmtools/datasets/binary.html

| | Phishing | IJCNN | Letter | Fashion | MNIST | CIFAR-10 |
|---|---|---|---|---|---|---|
| Random | 1095 | 845 | 3519 | 444 | 1599 | 1910 |
| Margin | 704 | 974 | 3164 | 247 | 1327 | 1474 |
| NeuAL-NTK-F | 4898 | 2684 | 6066 | 6001 | 6192 | 5007 |
| NeuAL-NTK-D | 796 | 744 | 3410 | 742 | 1239 | 1700 |
| ALPS | 978 | 683 | 3108 | 379 | 1433 | 1662 |
| I-NeurAL | **689**(↑ **2.1%**) | **469**(↑ **31.3%**) | **2571**(↑ **17.3%**) | **118**(↑ **52.2%**) | **674**(↑ **45.6%**) | **1372**(↑ **6.9%**) |

Table 1: Total regret comparison.

is not able to take the new knowledge obtained by queries into account and its performance is highly restricted by the hypothesis class. Although Margin algorithm is simple and straightforward, it exhibits great empirical performance in practice. This observation is consistent with other studies [50] [22]. However, Margin algorithm does not incorporate the exploitation portion and the query criterion is not adaptive to difference instances, thus still outperformed by I-NeurAL.

## 6 Conclusion

In this paper, we introduce two regret metrics and propose a novel neural-based algorithm (I-NeurAL) tailored for the streaming setting of non-parametric active learning. We carefully design its exploration strategy, query decision-maker, update rules, and training procedure, which lead to both the theoretical and empirical improvement compared to SOTA [48]. In the regret analysis, we provide an instance-dependent performance guarantee. On the other hand, we empirically show that I-NeurAL consistently achieves better accuracy under the same query budget than the strong baselines including the SOTA work [48] and [22].

## Acknowledgements

This work is supported by NSF (IIS-1947203, IIS-2117902, IIS-2137468, IIS-2002540, DMS-2134079, IIS-2131335, OAC-2130835, and DBI-2021898), DARPA (HR001121C0165), ARO (W911NF2110088), and C3.ai. The views and conclusions are those of the authors and should not be interpreted as representing the official policies of the funding agencies or the government.

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
