## 7 Proofs of Theorem 4.1 and 4.2

### 7.1 Proof of Theorem 4.1

*Proof.* Let $f(\mathbf{x}; \boldsymbol{\theta}) = f_1(\mathbf{x}; \boldsymbol{\theta}^1) + f_2(\phi(\mathbf{x}); \boldsymbol{\theta}^2)$ and we use $\mathbb{E}_{\mathbf{x}_t, y_t}$ to denote $\mathbb{E}_{(\mathbf{x}_t, y_t) \sim \mathcal{D}}$ for brevity. For any $t \in [T] \wedge (\mathbf{I}_t = 1)$, we have

$$
\begin{aligned}
R_t | (\mathbf{I}_t = 1) &= \mathop{\mathbb{E}}_{\mathbf{x}_t, y_t} \left[ L(\mathbf{y}_{t, \widehat{i}}, \mathbf{y}_{t, y_t}) - L(\mathbf{y}_{t, i^*}, \mathbf{y}_{t, y_t}) \right] \\
&= \mathop{\mathbb{E}}_{\mathbf{x}_t, y_t} \left[ 1 - L(\mathbf{y}_{t, i^*}, \mathbf{y}_{t, y_t}) - \left( 1 - L(\mathbf{y}_{t, \widehat{i}}, \mathbf{y}_{t, y_t}) \right) \right] \\
&= \mathop{\mathbb{E}}_{\mathbf{x}_t, y_t} \left[ r_{t, i^*}^1 - r_{t, \widehat{i}}^1 \right] \\
&\overset{E_1}{=} \mathop{\mathbb{E}}_{\mathbf{x}_t, y_t} \left[ \min\{ r_{t, i^*}^1 - r_{t, \widehat{i}}^1, 1 \} \right] \\
&= \mathop{\mathbb{E}}_{\mathbf{x}_t, y_t} \left[ \min\{ r_{t, i^*}^1 - f(\mathbf{x}_{t, i_t}; \boldsymbol{\theta}_{t-1}) + f(\mathbf{x}_{t, i_t}; \boldsymbol{\theta}_{t-1}) - r_{t, \widehat{i}}^1, 1 \} \right] \\
&\overset{E_2}{\leq} \mathop{\mathbb{E}}_{\mathbf{x}_t, y_t} \left[ \min\{ r_{t, i^*}^1 - f(\mathbf{x}_{t, i^*}; \boldsymbol{\theta}_{t-1}) + f(\mathbf{x}_{t, i_t}; \boldsymbol{\theta}_{t-1}) - r_{t, \widehat{i}}^1, 1 \} \right] \\
&\overset{E_3}{=} \mathop{\mathbb{E}}_{\mathbf{x}_t, y_t} \Big[ \min\{ r_{t, i^*}^1 - f(\mathbf{x}_{t, i^*}; \boldsymbol{\theta}_{t-1}^*) + f(\mathbf{x}_{t, i^*}; \boldsymbol{\theta}_{t-1}^*) - f(\mathbf{x}_{t, i^*}; \boldsymbol{\theta}_{t-1}) \\
&\qquad\qquad + f(\mathbf{x}_{t, i_t}; \boldsymbol{\theta}_{t-1}) - r_{t, \widehat{i}}^1, 1 \} \Big] \\
&\leq \mathop{\mathbb{E}}_{\mathbf{x}_t, y_t} \left[ \min\{ r_{t, i^*}^1 - f(\mathbf{x}_{t, i^*}; \boldsymbol{\theta}_{t-1}^*), 1 \} \right] + \mathop{\mathbb{E}}_{\mathbf{x}_t} \left[ \min\{ f(\mathbf{x}_{t, i^*}; \boldsymbol{\theta}_{t-1}^*) - f(\mathbf{x}_{t, i^*}; \boldsymbol{\theta}_{t-1}), 1 \} \right] \\
&\qquad + \mathop{\mathbb{E}}_{\mathbf{x}_t, y_t} \left[ \min\{ f(\mathbf{x}_{t, i_t}; \boldsymbol{\theta}_{t-1}) - r_{t, \widehat{i}}^1, 1 \} \right] \\
&\leq \mathop{\mathbb{E}}_{\mathbf{x}_t, y_t} \left[ \min\{ |r_{t, i^*}^1 - f(\mathbf{x}_{t, i^*}; \boldsymbol{\theta}_{t-1}^*)|, 1 \} \right] + \mathop{\mathbb{E}}_{\mathbf{x}_t} \left[ \min\{ |f(\mathbf{x}_{t, i^*}; \boldsymbol{\theta}_{t-1}^*) - f(\mathbf{x}_{t, i^*}; \boldsymbol{\theta}_{t-1})|, 1 \} \right] \\
&\qquad + \mathop{\mathbb{E}}_{\mathbf{x}_t, y_t} \left[ \min\{ |f(\mathbf{x}_{t, i_t}; \boldsymbol{\theta}_{t-1}) - r_{t, \widehat{i}}^1|, 1 \} \right]
\end{aligned}
\tag{7.1}
$$

where $E_1$ is based on the fact $r_{t, i} \in [0, 1], \forall i \in [k]$, $E_2$ is due to $f(\mathbf{x}_{t, i^*}; \boldsymbol{\theta}_{t-1}) \leq f(\mathbf{x}_{t, i_t}; \boldsymbol{\theta}_{t-1})$ according to our selection criterion, and $\boldsymbol{\theta}_{t-1}^*$ in $E_3$ are intermediate parameters to bound errors.

For any $t \in [T] \wedge (\mathbf{I}_t = 0)$, we have $R_t | (\mathbf{I}_t = 0) = \mathop{\mathbb{E}}_{\mathbf{x}_t, y_t} \left[ L(\mathbf{y}_{t, \widehat{i}}, \mathbf{y}_{t, y_t}) - L(\mathbf{y}_{t, i^*}, \mathbf{y}_{t, y_t}) \right] = 0$ based on Lemma 7.5.

Therefore, for any $t \in [T]$, we have

$$
\begin{aligned}
R_t &\leq \mathop{\mathbb{E}}_{\mathbf{x}_t, y_t} \left[ \min\{ |r_{t, i^*}^1 - f(\mathbf{x}_{t, i^*}; \boldsymbol{\theta}_{t-1}^*)|, 1 \} \right] + \mathop{\mathbb{E}}_{\mathbf{x}_t} \left[ \min\{ |f(\mathbf{x}_{t, i^*}; \boldsymbol{\theta}_{t-1}^*) - f(\mathbf{x}_{t, i^*}; \boldsymbol{\theta}_{t-1})|, 1 \} \right] \\
&\qquad + \mathop{\mathbb{E}}_{\mathbf{x}_t, y_t} \left[ \min\{ |f(\mathbf{x}_{t, i_t}; \boldsymbol{\theta}_{t-1}) - r_{t, \widehat{i}}^1|, 1 \} \right]
\end{aligned}
\tag{7.2}
$$

Based on Lemma 7.6, Lemma 7.7, and Lemma 7.14, with probability at least $1 - \delta$, we have

$$
R_t \leq 2 \left( \mathcal{O} \left( \frac{3L\nu + 2\sqrt{\mu}}{\sqrt{2t}} \right) + \mathcal{O} \left( \sqrt{ \frac{2 \log(\mathcal{O}(k)/\delta)}{t} } \right) + \xi_1 \right),
\tag{7.3}
$$

where $\xi_1 = \mathcal{O} \left( \frac{\nu L}{\sqrt{m}} \right) + \mathcal{O} \left( \frac{L^2 \sqrt{\log m} \nu^{4/3}}{m^{1/6}} \right)$.

Applying the union bound over all the rounds, with probability at least $1 - \delta$, we have

$$
\forall t \in [T], \quad R_t \leq 2 \left( \mathcal{O} \left( \frac{3L\nu + 2\sqrt{\mu}}{\sqrt{2t}} \right) + \mathcal{O} \left( \sqrt{ \frac{2 \log(\mathcal{O}(Tk)/\delta)}{t} } \right) + \xi_1. \right)
\tag{7.4}
$$

When $m$ is large enough, we have $\xi_1 = \mathcal{O}(\frac{1}{\sqrt{T}})$. Therefore, in round $T$, we have

$$R_T \leq \mathcal{O}\left(\frac{6L\nu + 4\sqrt{\mu}}{\sqrt{2T}}\right) + \mathcal{O}\left(\sqrt{\frac{2\log(\mathcal{O}(Tk)/\delta)}{T}}\right). \tag{7.5}$$

Finally, the regret of $T$ rounds is

$$\mathbf{R}_T = \sum_{t=1}^{T} R_t$$

$$\leq \sum_{t=1}^{T} 2\left(\underbrace{\mathcal{O}\left(\frac{3L\nu + 2\sqrt{\mu}}{\sqrt{2t}}\right) + \sqrt{\frac{2\log(\mathcal{O}(Tk)/\delta)}{t}}}_{I_1} + \underbrace{\xi_1}_{I_2}\right) \tag{7.6}$$

$$\leq 2\left(\underbrace{(2\sqrt{T}-1)\left[\mathcal{O}\left(\frac{3L\nu + 2\sqrt{\mu}}{\sqrt{2}}\right) + \sqrt{2\log(\mathcal{O}(Tk)/\delta)} + \underbrace{\mathcal{O}(1)}_{I_2}\right]}_{I_1}\right)$$

where $I_1$ is due to $\sum_{t=1}^{T} \frac{1}{\sqrt{t}} \leq \int_1^T \frac{1}{\sqrt{t}}\, dx + 1 = 2\sqrt{T} - 1$ and $I_2$ is because of the choice of $m$. The proof is complete. $\qquad \square$

## 7.2  Proof of Theorem 4.2

*Proof.* Given $\bar{\mathcal{T}}$, suppose $T > \bar{\mathcal{T}}$, we have

$$\mathbf{R}_T = \sum_{t=1}^{T} R_t$$

$$= \sum_{t=1}^{\bar{\mathcal{T}}} R_t + \sum_{t=\bar{\mathcal{T}}+1}^{T} R_t$$

$$= \underbrace{\sum_{t=1}^{\bar{\mathcal{T}}} \mathbb{E}_{\mathbf{x}_t, y_t}\left[L(\mathbf{y}_{t,\hat{i}}, \mathbf{y}_{t,y_t}) - L(\mathbf{y}_{t,i^*}, \mathbf{y}_{t,y_t})\right]}_{I_1} + \underbrace{\sum_{t=\bar{\mathcal{T}}+1}^{T} \mathbb{E}_{\mathbf{x}_t, y_t}\left[L(\mathbf{y}_{t,\hat{i}}, \mathbf{y}_{t,y_t}) - L(\mathbf{y}_{t,i^*}, \mathbf{y}_{t,y_t})\right]}_{I_2} \tag{7.7}$$

For $I_1$, based on Eq.(7.4), for any $\mu \in (0,1), t \in [\bar{\mathcal{T}}]$, we have

$$I_1 \leq \sum_{t=1}^{\bar{\mathcal{T}}} 2\left(\mathcal{O}\left(\frac{3L\nu + 2\sqrt{\mu}}{\sqrt{2t}}\right) + \sqrt{\frac{2\log(\mathcal{O}(Tk)/\delta)}{t}} + \xi_1.\right) \tag{7.8}$$

$$\overset{E_1}{\leq} (2\sqrt{\bar{\mathcal{T}}} - 1)\left[\mathcal{O}\left(\frac{6L\nu + 4\sqrt{\mu}}{\sqrt{2}}\right) + \sqrt{2\log(\mathcal{O}(Tk)/\delta)}\right]$$

where $E_1$ is because of $\sum_{t=1}^{T} \frac{1}{\sqrt{t}} \leq \int_1^T \frac{1}{\sqrt{t}}\, dx + 1 = 2\sqrt{T} - 1$ and the choice of $m$. It is straight forward to show that $R_T$ also satisfies this upper bound when $T \leq \bar{\mathcal{T}}$. For $I_2$, based on the Lemma 7.2, we have $\mathbb{E}_{\mathbf{x}_t \sim \mathcal{D}_{\mathcal{X}}}[h(\mathbf{x}_{t,\hat{i}}) - h(\mathbf{x}_{t,i^*})] = 0$ when $t \geq \bar{\mathcal{T}}$. This implies

$$\mathbb{E}_{(\mathbf{x}_t, y_t)\sim\mathcal{D}}[L(\mathbf{y}_{t,i^*}, \mathbf{y}_{t,y_t}) - L(\mathbf{y}_{t,\hat{i}}, \mathbf{y}_{t,y_t})] = 0. \tag{7.9}$$

Therefore, we have $I_2 = 0$. Putting them together, we have

$$\mathbf{R}_T \leq (2\sqrt{\bar{\mathcal{T}}} - 1)\left[\mathcal{O}\left(\frac{6L\nu + 4\sqrt{\mu}}{\sqrt{2}}\right) + \sqrt{2\log(\mathcal{O}(Tk)/\delta)}\right]. \tag{7.10}$$

According to Eq.(7.4) and Eq.(7.9), we have

$$\begin{cases} R_T \leq \mathcal{O}\left(\frac{6L\nu + 4\sqrt{\mu}}{\sqrt{2T}}\right) + 2\sqrt{\frac{2\log(\mathcal{O}(Tk)/\delta)}{T}}, & \text{if } T \leq \bar{\mathcal{T}}; \\ R_T = 0, \text{else}. \end{cases} \tag{7.11}$$

Then, replace $\bar{\mathcal{T}}$ and the proof is complete. $\qquad\qquad\square$

## 7.3  Main Lemmas

**Lemma 7.1.** *When* $t > \bar{\mathcal{T}} = \frac{12(\gamma+1)^2 \cdot \left[2\mu + 9L^2\nu^2 C_1^2 + 2\log(C_2 Tk/\delta)\right]}{\epsilon^2}$, *it has* $2(\gamma+1)\boldsymbol{\beta}_t \leq \epsilon$.

*Proof.* To achieve $2(\gamma+1)\boldsymbol{\beta}_t \leq \epsilon$, there exist constants $C_1, C_2$, such that

$$\sqrt{\frac{2\mu}{t}} + \left(\frac{3C_1 L\nu}{\sqrt{2t}}\right) + \sqrt{\frac{2\log(C_2 Tk/\delta)}{t}} \leq \frac{\epsilon}{2(\gamma+1)}$$

$$\left(\sqrt{\frac{2\mu}{t}} + \left(\frac{3C_1 L\nu}{\sqrt{2t}}\right) + \sqrt{\frac{2\log(C_2 Tk/\delta)}{t}}\right)^2 \leq \left(\frac{\epsilon}{2(\gamma+1)}\right)^2$$

$$3\left(\left(\sqrt{\frac{2\mu}{t}}\right)^2 + \left(\frac{3C_1 L\nu}{\sqrt{2t}}\right)^2 + \left(\sqrt{\frac{2\log(C_2 Tk/\delta)}{t}}\right)^2\right) \leq \left(\frac{\epsilon}{2(\gamma+1)}\right)^2$$

By calculations, we have

$$t \geq \frac{12(\gamma+1)^2 \cdot \left[2\mu + 9L^2\nu^2 C_1^2 + 2\log(C_2 Tk/\delta)\right]}{\epsilon^2}.$$

The proof is completed.

$\qquad\qquad\square$

**Lemma 7.2.** *For any* $\delta \in (0,1)$, $\gamma \geq 1$, *suppose* $T \geq \bar{\mathcal{T}}$. *Then, with probability at least* $1 - \delta$, *these exist constants* $C_1, C_2$, *such that the following two event* $\mathcal{E}_1, \mathcal{E}_2$ *happens*

$$\mathcal{E}_1 = \left\{ t \geq \bar{\mathcal{T}}, \underset{\mathbf{x}_t \sim \mathcal{D}_{\mathcal{X}}}{\mathbb{E}}[h(\mathbf{x}_{t,i^*}) - h(\mathbf{x}_{t,\widehat{i}})] = 0 \right\}, \tag{7.12}$$

$$\mathcal{E}_2 = \left\{ t \geq \bar{\mathcal{T}}, \underset{\mathbf{x}_t \sim \mathcal{D}_{\mathcal{X}}}{\mathbb{E}}[f(\mathbf{x}_{t,i^*}; \boldsymbol{\theta}_{t-1}) - f(\mathbf{x}_{t,\widehat{i}}; \boldsymbol{\theta}_{t-1})] = 0 \right\}. \tag{7.13}$$

*Proof.* According to Lemma 7.3 and Jensen's inequality, for any $i \in [k]$, with probability at least $1 - \delta$, we have

$$\underset{\mathbf{x}_t \sim \mathcal{D}_{\mathcal{X}}}{\mathbb{E}}\left[\min\left\{|f(\mathbf{x}_{t,i}; \boldsymbol{\theta}_{t-1}) - h(\mathbf{x}_{t,i})|, 1\right\}\right] \leq \underset{(\mathbf{x}_t, y_t) \sim \mathcal{D}}{\mathbb{E}}\left[\min\left\{|f(\mathbf{x}_{t,i}; \boldsymbol{\theta}_{t-1}) - r_{t,j}|, 1\right\}\right]$$

$$\leq \sqrt{\frac{2\mu}{t}} + \mathcal{O}\left(\frac{3L\nu}{\sqrt{2t}}\right) + \sqrt{\frac{2\log(\mathcal{O}(1)/\delta)}{t}} + 2\xi_1 \tag{7.14}$$

In round $t$, define the event

$$\widehat{\mathcal{E}}_0 = \left\{ \tau \in [t], i \in [k], \underset{\mathbf{x}_\tau \sim \mathcal{D}_{\mathcal{X}}}{\mathbb{E}}\left[\min\{|f(\mathbf{x}_{\tau,i}; \boldsymbol{\theta}_{\tau-1}) - h(\mathbf{x}_{\tau,i})|, 1\}\right] \leq \boldsymbol{\beta}_\tau \right\} \tag{7.15}$$

Then, applying the union bound over $T$ and $k$, then, with probability at least $1 - \delta$, $\mathcal{E}$ happens, where

$$\boldsymbol{\beta}_\tau = \sqrt{\frac{2\mu}{\tau}} + \mathcal{O}\left(\frac{3L\nu}{\sqrt{2\tau}}\right) + \sqrt{\frac{2\log(\mathcal{O}(Tk)/\delta)}{\tau}}. \tag{7.16}$$

where we merge $\xi_1$ into $\mathcal{O}\left(\frac{3L}{\sqrt{2\tau}}\right)$ as a result of choice of $m$. Next, define the event

$$\widehat{\mathcal{E}}_1 = \left\{ t \geq \bar{\mathcal{T}}, \underset{\mathbf{x}_t \sim \mathcal{D}_{\mathcal{X}}}{\mathbb{E}}[f(\mathbf{x}_{t,i^*}; \boldsymbol{\theta}_{t-1}) - f(\mathbf{x}_{t,i^\circ}; \boldsymbol{\theta}_{t-1})] < 2\gamma\boldsymbol{\beta}_t \right\}. \tag{7.17}$$

When $\widehat{\mathcal{E}}_0$ happens with probability at least $1 - \delta$, based on the fact $h(\cdot) \in [0, 1]$, we have

$$\begin{cases} \mathbb{E}_{\mathbf{x}_t \sim \mathcal{D}_{\mathcal{X}}}[f(\mathbf{x}_{t,i^*}; \boldsymbol{\theta}_{t-1})] - \min\{\boldsymbol{\beta}_t, 1\} \leq \mathbb{E}_{\mathbf{x}_t \sim \mathcal{D}_{\mathcal{X}}}[h(\mathbf{x}_{t,i^*})] \leq \mathbb{E}_{\mathbf{x}_t \sim \mathcal{D}_{\mathcal{X}}}[f(\mathbf{x}_{t,i^*}; \boldsymbol{\theta}_{t-1})] + \min\{\boldsymbol{\beta}_t, 1\} \\ \mathbb{E}_{\mathbf{x}_t \sim \mathcal{D}_{\mathcal{X}}}[f(\mathbf{x}_{t,i^\circ}; \boldsymbol{\theta}_{t-1})] - \min\{\boldsymbol{\beta}_t, 1\} \leq \mathbb{E}_{\mathbf{x}_t \sim \mathcal{D}_{\mathcal{X}}}[h(\mathbf{x}_{t,i^\circ})] \leq \mathbb{E}_{\mathbf{x}_t \sim \mathcal{D}_{\mathcal{X}}}[f(\mathbf{x}_{t,i^\circ}; \boldsymbol{\theta}_{t-1})] + \min\{\boldsymbol{\beta}_t, 1\} \end{cases}$$
$$(7.18)$$

Then, based on Lemma 7.1, with probability at least $1 - \delta$, when $t > \bar{\mathcal{T}}$, $2(\gamma + 1)\boldsymbol{\beta}_t \leq \epsilon \Rightarrow \boldsymbol{\beta}_t < 1$. This implies

$$\begin{cases} \mathbb{E}_{\mathbf{x}_t \sim \mathcal{D}_{\mathcal{X}}}[f(\mathbf{x}_{t,i^*}; \boldsymbol{\theta}_{t-1})] - \boldsymbol{\beta}_t \leq \mathbb{E}_{\mathbf{x}_t \sim \mathcal{D}_{\mathcal{X}}}[h(\mathbf{x}_{t,i^*})] \leq \mathbb{E}_{\mathbf{x}_t \sim \mathcal{D}_{\mathcal{X}}}[f(\mathbf{x}_{t,i^*}; \boldsymbol{\theta}_{t-1})] + \boldsymbol{\beta}_t \\ \mathbb{E}_{\mathbf{x}_t \sim \mathcal{D}_{\mathcal{X}}}[f(\mathbf{x}_{t,i^\circ}; \boldsymbol{\theta}_{t-1})] - \boldsymbol{\beta}_t \leq \mathbb{E}_{\mathbf{x}_t \sim \mathcal{D}_{\mathcal{X}}}[h(\mathbf{x}_{t,i^\circ})] \leq \mathbb{E}_{\mathbf{x}_t \sim \mathcal{D}_{\mathcal{X}}}[f(\mathbf{x}_{t,i^\circ}; \boldsymbol{\theta}_{t-1})] + \boldsymbol{\beta}_t \end{cases}$$
$$(7.19)$$

Therefore, we have

$$\mathbb{E}_{\mathbf{x}_t \sim \mathcal{D}_{\mathcal{X}}}[h(\mathbf{x}_{t,i^*}) - h(\mathbf{x}_{t,i^\circ})] \leq \mathbb{E}_{\mathbf{x}_t \sim \mathcal{D}_{\mathcal{X}}}[f(\mathbf{x}_{t,i^*}; \boldsymbol{\theta}_{t-1})] + \boldsymbol{\beta}_t - \left( \mathbb{E}_{\mathbf{x}_t \sim \mathcal{D}_{\mathcal{X}}}[f(\mathbf{x}_{t,i^\circ}; \boldsymbol{\theta}_{t-1})] - \boldsymbol{\beta}_t \right)$$
$$\leq \mathbb{E}_{\mathbf{x}_t \sim \mathcal{D}_{\mathcal{X}}}[f(\mathbf{x}_{t,i^*}; \boldsymbol{\theta}_{t-1}) - f(\mathbf{x}_{t,i^\circ}; \boldsymbol{\theta}_{t-1})] + 2\boldsymbol{\beta}_t.$$
$$(7.20)$$

Suppose $\widehat{\mathcal{E}}_1$ happens, we have

$$\mathbb{E}_{\mathbf{x}_t \sim \mathcal{D}_{\mathcal{X}}}[h(\mathbf{x}_{t,i^*}) - h(\mathbf{x}_{t,i^\circ})] \leq 2(\gamma + 1)\boldsymbol{\beta}_t.$$
$$(7.21)$$

Then, based on Lemma 7.1, when $t > \bar{\mathcal{T}}$, $2(\gamma + 1)\boldsymbol{\beta}_t \leq \epsilon$. Therefore, we have

$$\mathbb{E}_{\mathbf{x}_t \sim \mathcal{D}_{\mathcal{X}}}[h(\mathbf{x}_{t,i^*}) - h(\mathbf{x}_{t,i^\circ})] \leq 2(\gamma + 1)\boldsymbol{\beta}_t \leq \epsilon.$$
$$(7.22)$$

This contradicts Assumption 4.2, i.e., $h(\mathbf{x}_{t,i^*}) - h(\mathbf{x}_{t,i^\circ}) \geq \epsilon$. Hence, $\widehat{\mathcal{E}}_1$ will not happen. Accordingly, with probability at least $1 - \delta$, the following event will happen

$$\widehat{\mathcal{E}}_2 = \left\{ t \geq \bar{\mathcal{T}}, \mathbb{E}_{\mathbf{x}_t \sim \mathcal{D}_{\mathcal{X}}}[f(\mathbf{x}_{t,i^*}; \boldsymbol{\theta}_{t-1}) - f(\mathbf{x}_{t,i^\circ}; \boldsymbol{\theta}_{t-1})] \geq 2\gamma \boldsymbol{\beta}_t \right\}.$$
$$(7.23)$$

Therefore, we have $\mathbb{E}[f(\mathbf{x}_{t,i^*}; \boldsymbol{\theta}_{t-1})] > \mathbb{E}[f(\mathbf{x}_{t,i^\circ}; \boldsymbol{\theta}_{t-1})]$. Recall that $i^* = \arg\max_{i \in [k]} h(\mathbf{x}_{t,i})$ and $\widehat{i} = \arg\max_{i \in [k]} f(\mathbf{x}_{t,i}; \boldsymbol{\theta}_{t-1})$. As

$$\forall i \in ([k] \setminus \{\widehat{i}\}), f(\mathbf{x}_{t,i}; \boldsymbol{\theta}_{t-1}) \leq f(\mathbf{x}_{t,i^\circ}; \boldsymbol{\theta}_{t-1})$$
$$\Rightarrow \forall i \in ([k] \setminus \{\widehat{i}\}), \mathbb{E}_{\mathbf{x}_t \sim \mathcal{D}_{\mathcal{X}}}[f(\mathbf{x}_{t,i}; \boldsymbol{\theta}_{t-1})] \leq \mathbb{E}_{\mathbf{x}_t \sim \mathcal{D}_{\mathcal{X}}}[f(\mathbf{x}_{t,i^\circ}; \boldsymbol{\theta}_{t-1})]$$
$$(7.24)$$

we have

$$\forall i \in ([k] \setminus \{\widehat{i}\}), \mathbb{E}_{\mathbf{x}_t \sim \mathcal{D}_{\mathcal{X}}}[f(\mathbf{x}_{t,i^*}; \boldsymbol{\theta}_{t-1})] > \mathbb{E}_{\mathbf{x}_t \sim \mathcal{D}_{\mathcal{X}}}[f(\mathbf{x}_{t,i}; \boldsymbol{\theta}_{t-1})].$$
$$(7.25)$$

Based on the definition of $\widehat{i}$, we have

$$\mathbb{E}_{\mathbf{x}_t \sim \mathcal{D}_{\mathcal{X}}}[f(\mathbf{x}_{t,i^*}; \boldsymbol{\theta}_{t-1})] = \mathbb{E}_{\mathbf{x}_t \sim \mathcal{D}_{\mathcal{X}}}[f(\mathbf{x}_{t,\widehat{i}}; \boldsymbol{\theta}_{t-1})] = \mathbb{E}_{\mathbf{x}_t \sim \mathcal{D}_{\mathcal{X}}}[\max_{i \in [k]} f(\mathbf{x}_{t,i}; \boldsymbol{\theta}_{t-1})].$$
$$(7.26)$$

This indicates $\mathcal{E}_2$ happens with probability at least $1 - \delta$.

Therefore, based on $\widehat{\mathcal{E}}_2$, the following inferred event $\widehat{\mathcal{E}}_3$ happens with probability at least $1 - \delta$:

$$\widehat{\mathcal{E}}_3 = \left\{ t \geq \bar{\mathcal{T}}, \mathbb{E}_{\mathbf{x}_t \sim \mathcal{D}_{\mathcal{X}}}[f(\mathbf{x}_{t,\widehat{i}}; \boldsymbol{\theta}_{t-1}) - f(\mathbf{x}_{t,i^\circ}; \boldsymbol{\theta}_{t-1})] \geq 2\gamma \boldsymbol{\beta}_t \right\}.$$
$$(7.27)$$

Then, based on Eq. 7.19, we have

$$\mathbb{E}[h(\mathbf{x}_{t,\widehat{i}}) - h(\mathbf{x}_{t,i^\circ})] \geq \mathbb{E}[f(\mathbf{x}_{t,\widehat{i}}; \boldsymbol{\theta}_{t-1})] - \boldsymbol{\beta}_t - (\mathbb{E}[f(\mathbf{x}_{t,i^\circ}; \boldsymbol{\theta}_{t-1})] + \boldsymbol{\beta}_t)$$
$$= \mathbb{E}[f(\mathbf{x}_{t,\widehat{i}}; \boldsymbol{\theta}_{t-1}) - f(\mathbf{x}_{t,i^\circ}; \boldsymbol{\theta}_{t-1})] - 2\boldsymbol{\beta}_t$$
$$\overset{E_1}{\geq} 2(\gamma - 1)\boldsymbol{\beta}_t$$
$$\geq 0$$
$$(7.28)$$

where $E_1$ is because $\widehat{\mathcal{E}}_3$ happened with probability at least $1 - \delta$. Therefore, we have

$$\mathop{\mathbb{E}}_{\mathbf{x}_t \sim \mathcal{D}_{\mathcal{X}}}[h(\mathbf{x}_{t,\widehat{i}})] - \mathop{\mathbb{E}}_{\mathbf{x}_t \sim \mathcal{D}_{\mathcal{X}}}[h(\mathbf{x}_{t,i^\circ})] > 0. \tag{7.29}$$

Similarly, we can prove that

$$\Rightarrow \forall i \in ([k] \setminus \{\widehat{i}\}), \mathop{\mathbb{E}}_{\mathbf{x}_t \sim \mathcal{D}_{\mathcal{X}}}[h(\mathbf{x}_{t,\widehat{i}})] - \mathop{\mathbb{E}}_{\mathbf{x}_t \sim \mathcal{D}_{\mathcal{X}}}[h(\mathbf{x}_{t,i})] > 0. \tag{7.30}$$

Then, based on the definition of $\mathbf{x}_{t,i^*}$, we have

$$\mathop{\mathbb{E}}_{\mathbf{x}_t \sim \mathcal{D}_{\mathcal{X}}}[h(\mathbf{x}_{t,\widehat{i}})] = \mathop{\mathbb{E}}_{\mathbf{x}_t \sim \mathcal{D}_{\mathcal{X}}}[h(\mathbf{x}_{t,i^*})] = \mathop{\mathbb{E}}_{\mathbf{x}_t \sim \mathcal{D}_{\mathcal{X}}}[\max_{i \in [k]} h(\mathbf{x}_{t,i})]. \tag{7.31}$$

Thus, the event $\mathcal{E}_1$ happens with probability at least $1 - \delta$. $\qquad \square$

**Lemma 7.3.** *For any $\delta \in (0,1), \nu > 0$, suppose $m$ satisfies the conditions in Theorem 4.1. Then, with probability at least $1 - \delta$, given any fixed index $i \in [k]$, it holds that*

$$\mathop{\mathbb{E}}_{(\mathbf{x}_t, y_t) \sim \mathcal{D}}\left[\min\left\{\left|f_1(\mathbf{x}_{t,i}; \boldsymbol{\theta}_{t-1}^1) + f_2(\phi(\mathbf{x}_{t,i}); \boldsymbol{\theta}_{t-1}^2) - r_{t,i}^1\right|, 1\right\}\right]$$

$$\leq \sqrt{\frac{2\mu}{t}} + \mathcal{O}\left(\frac{3L\nu}{\sqrt{2t}}\right) + \sqrt{\frac{2\log(\mathcal{O}(1)/\delta)}{t}} + 2\xi_1. \tag{7.32}$$

*Proof.* Given any $i \in [k]$, let $i$ be a fixed index, i.e., suppose there exist a policy $\Omega_i$ which always select the $i$-th context $(\mathbf{x}_{t,i}, r_{t,i}^1)$ for every round $t \in [T]$. Then, in round $t$, we have the collected data by $\Omega_i$: $\mathcal{H}_{t-1}^{1,i} = \{\mathbf{x}_{\tau,i}, r_{\tau,i}^1\}_{\tau=1}^{t-1}$. Then, let $\boldsymbol{\theta}_{t-1}^{1,i}, \boldsymbol{\theta}_{t-1}^{2,i}$ represent the parameters trained only on $\mathcal{H}_{t-1}^i$ using Algorithm 2, satisfying $\|\boldsymbol{\theta}_{t-1}^{1,i} - \boldsymbol{\theta}_0^1\|_2 \leq \mathcal{O}(\frac{\nu}{\sqrt{m}})$ and $\|\boldsymbol{\theta}_{t-1}^{2,i} - \boldsymbol{\theta}_0^2\|_2 \leq \mathcal{O}(\frac{\nu}{\sqrt{m}})$. Note that $\boldsymbol{\theta}_{t-1}^{1,i}, \boldsymbol{\theta}_{t-1}^{2,i}$ are uniformly drawn from $\{\widehat{\boldsymbol{\theta}}_{\tau-1}^{1,i}, \widehat{\boldsymbol{\theta}}_{\tau-1}^{2,i}\}_{\tau=0}^{t-1}$ and these parameters are unknown but introduced for the sake of analysis. Then, for $\tau \in [t]$, we define

$$V_\tau = \mathop{\mathbb{E}}_{(\mathbf{x}_\tau, y_\tau) \sim \mathcal{D}}\left[\min\{|f_1(\mathbf{x}_{\tau,i}; \widehat{\boldsymbol{\theta}}_{\tau-1}^{1,i}) + f_2(\phi(\mathbf{x}_{\tau,i}); \widehat{\boldsymbol{\theta}}_{\tau-1}^{2,i}) - r_{\tau,i}^1|, 1\}\right]$$

$$- \min\{|f_1(\mathbf{x}_{\tau,i}; \widehat{\boldsymbol{\theta}}_{\tau-1}^{1,i}) + f_2(\phi(\mathbf{x}_{\tau,i}); \widehat{\boldsymbol{\theta}}_{\tau-1}^{2,i}) - r_{\tau,i}^1|, 1\} \tag{7.33}$$

Then, we have

$$\mathbb{E}[V_\tau | F_{\tau-1}] = \mathop{\mathbb{E}}_{(\mathbf{x}_\tau, y_\tau) \sim \mathcal{D}}\left[\min\{|f_1(\mathbf{x}_{\tau,i}; \widehat{\boldsymbol{\theta}}_{\tau-1}^{1,i}) + f_2(\phi(\mathbf{x}_{\tau,i}); \widehat{\boldsymbol{\theta}}_{\tau-1}^{2,i}) - r_{\tau,i}^1|, 1\}\right]$$

$$- \mathop{\mathbb{E}}_{(\mathbf{x}_\tau, y_\tau) \sim \mathcal{D}}\left[\min\{|f_1(\mathbf{x}_{\tau,i}; \widehat{\boldsymbol{\theta}}_{\tau-1}^{1,i}) + f_2(\phi(\mathbf{x}_{\tau,i}); \widehat{\boldsymbol{\theta}}_{\tau-1}^{2,i}) - r_{\tau,i}^1|, 1\}\right] \tag{7.34}$$

$$= 0$$

where $F_{\tau-1}$ denotes the $\sigma$-algebra generated by the history $\mathcal{H}_{\tau-1}^i$. Therefore, $\{V_\tau\}_{\tau=1}^t$ are the martingale difference sequence.

Then, using the similar proof method in Lemma 7.6, we have

$$\mathop{\mathbb{E}}_{(\mathbf{x}_t, y_t) \sim \mathcal{D}}\left[\min\left\{\left|f_1(\mathbf{x}_{t,i}; \boldsymbol{\theta}_{t-1}^{1,i}) + f_2(\phi(\mathbf{x}_{t,i}); \boldsymbol{\theta}_{t-1}^{2,i}) - r_{t,i}^1\right|, 1\right\} \mid \mathcal{H}_{t-1}^i\right]$$

$$\leq \sqrt{\frac{2\mu}{t}} + \mathcal{O}\left(\frac{3L\nu}{\sqrt{2t}}\right) + \sqrt{\frac{2\log(\mathcal{O}(1)/\delta)}{t}}. \tag{7.35}$$

Let $f(\mathbf{x}; \boldsymbol{\theta}_{t-1}) = f_1(\mathbf{x}; \boldsymbol{\theta}_{t-1}^1) + f_2(\phi(\mathbf{x}); \boldsymbol{\theta}_{t-1}^2)$ and $f(\mathbf{x}; \boldsymbol{\theta}_{t-1}^i) = f_1(\mathbf{x}; \boldsymbol{\theta}_{t-1}^{1,i}) + f_2(\phi(\mathbf{x}); \boldsymbol{\theta}_{t-1}^{2,i})$.

Then, given $i \in [k]$, we have

$$\mathop{\mathbb{E}}_{(\mathbf{x}_t, y_t) \sim \mathcal{D}}\left[\min\left\{|f(\mathbf{x}_{t,i}; \boldsymbol{\theta}_{t-1}) - r_{t,i}^1|, 1\right\}\right]$$

$$= \mathop{\mathbb{E}}_{(\mathbf{x}_t, y_t) \sim \mathcal{D}}\left[\min\left\{|f(\mathbf{x}_{t,i}; \boldsymbol{\theta}_{t-1}) - f(\mathbf{x}_{t,i}; \boldsymbol{\theta}_{t-1}^i) + f(\mathbf{x}_{t,i}; \boldsymbol{\theta}_{t-1}^i) - r_{t,i}^1|, 1\right\}\right]$$

$$\leq \mathop{\mathbb{E}}_{(\mathbf{x}_t, y_t) \sim \mathcal{D}}\left[|f(\mathbf{x}_{t,i}; \boldsymbol{\theta}_{t-1}) - f(\mathbf{x}_{t,i}; \boldsymbol{\theta}_{t-1}^i)|\right] + \mathop{\mathbb{E}}_{(\mathbf{x}_t, y_t) \sim \mathcal{D}}\left[\min\left\{|f(\mathbf{x}_{t,i}; \boldsymbol{\theta}_{t-1}^i) - r_{t,i}^1|, 1\right\}\right] \tag{7.36}$$

$$\leq \sqrt{\frac{2\mu}{t}} + \mathcal{O}\left(\frac{3L\nu}{\sqrt{2t}}\right) + \sqrt{\frac{2\log(\mathcal{O}(1)/\delta)}{t}} + 2\xi_1$$

where the last inequality is the application of Lemma 7.14 and Eq. (7.35). The proof is complete.

**Lemma 7.4** (Label Complexity Analysis). *For any $\delta \in (0,1), \gamma \geq 1$, suppose $m$ satisfies the conditions in Theorem 4.1. Then, with probability at least $1 - \delta$, we have*

$$\mathbf{N}_T \leq \frac{12(\gamma+1)^2 \cdot \left[2\mu + 9L^2\nu^2 C_1^2 + 2\log(C_2 Tk/\delta)\right]}{\epsilon^2}. \tag{7.37}$$

*Proof.* Recall that $\mathbf{x}_{t,\widehat{i}} = \max_{\mathbf{x}_{t,i}, i \in [k]} f(\mathbf{x}_{t,i}; \boldsymbol{\theta}_{t-1})$, and $\mathbf{x}_{t,i^\circ} = \max_{\mathbf{x}_{t,i}, i \in ([k]/\{\mathbf{x}_{t,\widehat{i}}\})} f(\mathbf{x}_{t,i}; \boldsymbol{\theta}_{t-1})$. With probability at least $1 - \delta$, according to Eq. (7.15) the event

$$\widehat{\mathcal{E}}_0 = \left\{ \tau \in [t], i \in [k], \underset{\mathbf{x}_\tau \sim \mathcal{D}_{\mathcal{X}}}{\mathbb{E}} \left[ \min\{|f(\mathbf{x}_{\tau,i}; \boldsymbol{\theta}_{\tau-1}) - h(\mathbf{x}_{\tau,i})|, 1\} \right] \leq \boldsymbol{\beta}_\tau \right\}$$

happens. Therefore, we have

$$\begin{cases} \underset{\mathbf{x}_t \sim \mathcal{D}_{\mathcal{X}}}{\mathbb{E}}[h(\mathbf{x}_{t,\widehat{i}})] - \min\{\boldsymbol{\beta}_t, 1\} \leq \underset{\mathbf{x}_t \sim \mathcal{D}_{\mathcal{X}}}{\mathbb{E}}[f(\mathbf{x}_{t,\widehat{i}}; \boldsymbol{\theta}_{t-1})] \leq \underset{\mathbf{x}_t \sim \mathcal{D}_{\mathcal{X}}}{\mathbb{E}}[h(\mathbf{x}_{t,\widehat{i}})] + \min\{\boldsymbol{\beta}_t, 1\} \\ \underset{\mathbf{x}_t \sim \mathcal{D}_{\mathcal{X}}}{\mathbb{E}}[h(\mathbf{x}_{t,i^\circ})] - \min\{\boldsymbol{\beta}_t, 1\} \leq \underset{\mathbf{x}_t \sim \mathcal{D}_{\mathcal{X}}}{\mathbb{E}}[f(\mathbf{x}_{t,i^\circ}; \boldsymbol{\theta}_{t-1})] \leq \underset{\mathbf{x}_t \sim \mathcal{D}_{\mathcal{X}}}{\mathbb{E}}[h(\mathbf{x}_{t,i^\circ})] + \min\{\boldsymbol{\beta}_t, 1\}. \end{cases} \tag{7.38}$$

Then, we have

$$\begin{cases} \underset{\mathbf{x}_t \sim \mathcal{D}_{\mathcal{X}}}{\mathbb{E}}[f(\mathbf{x}_{t,\widehat{i}}; \boldsymbol{\theta}_{t-1}) - f(\mathbf{x}_{t,i^\circ}; \boldsymbol{\theta}_{t-1})] \leq \underset{\mathbf{x}_t \sim \mathcal{D}_{\mathcal{X}}}{\mathbb{E}}[h(\mathbf{x}_{t,\widehat{i}})] - \underset{\mathbf{x}_t \sim \mathcal{D}_{\mathcal{X}}}{\mathbb{E}}[h(\mathbf{x}_{t,i^\circ})] + 2\min\{\boldsymbol{\beta}_t, 1\} \\ \underset{\mathbf{x}_t \sim \mathcal{D}_{\mathcal{X}}}{\mathbb{E}}[f(\mathbf{x}_{t,\widehat{i}}; \boldsymbol{\theta}_{t-1}) - f(\mathbf{x}_{t,i^\circ}; \boldsymbol{\theta}_{t-1})] \geq \underset{\mathbf{x}_t \sim \mathcal{D}_{\mathcal{X}}}{\mathbb{E}}[h(\mathbf{x}_{t,\widehat{i}})] - \underset{\mathbf{x}_t \sim \mathcal{D}_{\mathcal{X}}}{\mathbb{E}}[h(\mathbf{x}_{t,i^\circ})] - 2\min\{\boldsymbol{\beta}_t, 1\}. \end{cases} \tag{7.39}$$

Let $\epsilon_t = |\underset{\mathbf{x} \sim \mathcal{D}_{\mathcal{X}}}{\mathbb{E}}[h(\mathbf{x}_{t,\widehat{i}})] - \underset{\mathbf{x} \sim \mathcal{D}_{\mathcal{X}}}{\mathbb{E}}[h(\mathbf{x}_{t,i^\circ})]|$. Then, based on Lemma 7.1, when $t \geq \bar{\mathcal{T}}$, we have

$$2(\gamma+1)\boldsymbol{\beta}_t \leq \epsilon_t \leq 1. \tag{7.40}$$

For any $t \in [T]$ and $t < \bar{\mathcal{T}}$, we have $\underset{\mathbf{x}_t \sim \mathcal{D}_{\mathcal{X}}}{\mathbb{E}}[\mathbf{I}_t] \leq 1$. For the round $t > \bar{\mathcal{T}}$, suppose $\underset{\mathbf{x}_t \sim \mathcal{D}_{\mathcal{X}}}{\mathbb{E}}[h(\mathbf{x}_{t,\widehat{i}})] - \underset{\mathbf{x}_t \sim \mathcal{D}_{\mathcal{X}}}{\mathbb{E}}[h(\mathbf{x}_{t,i^\circ})] = -\epsilon_t$, then, we have

$$\underset{\mathbf{x}_t \sim \mathcal{D}_{\mathcal{X}}}{\mathbb{E}}[f(\mathbf{x}_{t,\widehat{i}}; \boldsymbol{\theta}_{t-1}) - f(\mathbf{x}_{t,i^\circ}; \boldsymbol{\theta}_{t-1})] \leq -\epsilon_t + 2\boldsymbol{\beta}_t \overset{E_1}{\leq} -\epsilon_t + \frac{\epsilon_t}{2} \leq 0, \tag{7.41}$$

where $E_1$ is because of Eq. (7.40) since $\gamma \geq 1$. This contradicts the fact $\underset{\mathbf{x}_t \sim \mathcal{D}_{\mathcal{X}}}{\mathbb{E}}[f(\mathbf{x}_{t,\widehat{i}}; \boldsymbol{\theta}_{t-1}) - f(\mathbf{x}_{t,i^\circ}; \boldsymbol{\theta}_{t-1})] \geq 0$. Therefore, $\underset{\mathbf{x}_t \sim \mathcal{D}_{\mathcal{X}}}{\mathbb{E}}[h(\mathbf{x}_{t,\widehat{i}})] - \underset{\mathbf{x}_t \sim \mathcal{D}_{\mathcal{X}}}{\mathbb{E}}[h(\mathbf{x}_{t,i^\circ})] = \epsilon_t$. Then, based on Eq.(7.39), we have

$$\underset{\mathbf{x}_t \sim \mathcal{D}_{\mathcal{X}}}{\mathbb{E}}[f(\mathbf{x}_{t,\widehat{i}}; \boldsymbol{\theta}_{t-1}) - f(\mathbf{x}_{t,i^\circ}; \boldsymbol{\theta}_{t-1})] \geq \epsilon_t - 2\boldsymbol{\beta}_t \overset{E_2}{\geq} 2\gamma\boldsymbol{\beta}_t, \tag{7.42}$$

where $E_2$ is because of Eq. (7.40).

According to Lemma 7.2, when $t > \bar{\mathcal{T}}$, $\underset{\mathbf{x}_t \sim \mathcal{D}_{\mathcal{X}}}{\mathbb{E}}[f(\mathbf{x}_{t,i^*}; \boldsymbol{\theta}_{t-1})] = \underset{\mathbf{x}_t \sim \mathcal{D}_{\mathcal{X}}}{\mathbb{E}}[f(\mathbf{x}_{t,\widehat{i}}; \boldsymbol{\theta}_{t-1})]$. Then, applying Eq. (7.42), for the round $t > \bar{\mathcal{T}}$, we have $\underset{\mathbf{x}_t \sim \mathcal{D}_{\mathcal{X}}}{\mathbb{E}}[\mathbf{I}_t] = 0$.

Then, assume $T > \bar{\mathcal{T}}$, we have

$$\begin{aligned} \mathbf{N}_T &= \sum_{t=1}^{T} \underset{\mathbf{x}_t \sim \mathcal{D}_{\mathcal{X}}}{\mathbb{E}} \left[ \mathbb{1}\{f(\mathbf{x}_{t,\widehat{i}}; \boldsymbol{\theta}_{t-1}) - f(\mathbf{x}_{t,i^\circ}; \boldsymbol{\theta}_{t-1}) < 2\gamma\boldsymbol{\beta}_t\} \right] \\ &\leq \sum_{t=1}^{\bar{\mathcal{T}}} 1 + \sum_{t=\bar{\mathcal{T}}+1}^{T} \underset{\mathbf{x}_t \sim \mathcal{D}_{\mathcal{X}}}{\mathbb{E}} \left[ \mathbb{1}\{f(\mathbf{x}_{t,\widehat{i}}; \boldsymbol{\theta}_{t-1}) - f(\mathbf{x}_{t,i^\circ}; \boldsymbol{\theta}_{t-1}) < 2\gamma\boldsymbol{\beta}_t\} \right] \\ &= \bar{\mathcal{T}} + 0. \end{aligned} \tag{7.43}$$

Therefore, we have $\mathbf{N}_T \leq \bar{\mathcal{T}}$. $\qquad\square$

**Lemma 7.5.** *For any $\delta \in (0,1), \gamma \geq 1$, suppose $m$ satisfies the conditions in Theorem 4.1. Then, with probability at least $1 - \delta$, when $\mathbf{I}_t = 0$, we have*

$$\mathop{\mathbb{E}}_{\mathbf{x}_t \sim \mathcal{D}_{\mathcal{X}}}[h(\mathbf{x}_{t,\widehat{i}})] = \mathop{\mathbb{E}}_{\mathbf{x}_t \sim \mathcal{D}_{\mathcal{X}}}[h(\mathbf{x}_{t,i^*})],$$

$$\mathop{\mathbb{E}}_{(\mathbf{x}_t,y_t) \sim \mathcal{D}}[L(\mathbf{y}_{t,\widehat{i}}, \mathbf{y}_{t,y_t})] = \mathop{\mathbb{E}}_{(\mathbf{x}_t,y_t) \sim \mathcal{D}}[L(\mathbf{y}_{t,i^*}, \mathbf{y}_{t,y_t})].$$

*Proof.* As $\mathbf{I}_t = 0$, we have

$$|f(\mathbf{x}_{t,\widehat{i}}; \boldsymbol{\theta}_{t-1}) - f(\mathbf{x}_{t,i^\circ}; \boldsymbol{\theta}_{t-1})| = f(\mathbf{x}_{t,\widehat{i}}; \boldsymbol{\theta}_{t-1}) - f(\mathbf{x}_{t,i^\circ}; \boldsymbol{\theta}_{t-1}) \geq 2\gamma\boldsymbol{\beta}_t$$

When $\widehat{\mathcal{E}}_0$ (Eq. (7.15)) happens with probability at least $1 - \delta$, based on the fact $h(\cdot) \in [0,1]$, we have

$$\begin{cases} \mathop{\mathbb{E}}_{\mathbf{x}_t \sim \mathcal{D}_{\mathcal{X}}}[f(\mathbf{x}_{t,\widehat{i}}; \boldsymbol{\theta}_{t-1})] - \min\{\boldsymbol{\beta}_t, 1\} \leq \mathop{\mathbb{E}}_{\mathbf{x}_t \sim \mathcal{D}_{\mathcal{X}}}[h(\mathbf{x}_{t,\widehat{i}})] \leq \mathop{\mathbb{E}}_{\mathbf{x}_t \sim \mathcal{D}_{\mathcal{X}}}[f(\mathbf{x}_{t,\widehat{i}}; \boldsymbol{\theta}_{t-1})] + \min\{\boldsymbol{\beta}_t, 1\} \\ \mathop{\mathbb{E}}_{\mathbf{x}_t \sim \mathcal{D}_{\mathcal{X}}}[f(\mathbf{x}_{t,i^\circ}; \boldsymbol{\theta}_{t-1})] - \min\{\boldsymbol{\beta}_t, 1\} \leq \mathop{\mathbb{E}}_{\mathbf{x}_t \sim \mathcal{D}_{\mathcal{X}}}[h(\mathbf{x}_{t,i^\circ})] \leq \mathop{\mathbb{E}}_{\mathbf{x}_t \sim \mathcal{D}_{\mathcal{X}}}[f(\mathbf{x}_{t,i^\circ}; \boldsymbol{\theta}_{t-1})] + \min\{\boldsymbol{\beta}_t, 1\} \end{cases}$$

(7.44)

Then, with probability at least $1 - \delta$, we have

$$\begin{aligned} \mathop{\mathbb{E}}_{\mathbf{x}_t \sim \mathcal{D}_{\mathcal{X}}}[h(\mathbf{x}_{t,\widehat{i}}) - h(\mathbf{x}_{t,i^\circ})] &\geq \mathop{\mathbb{E}}_{\mathbf{x}_t \sim \mathcal{D}_{\mathcal{X}}}[f(\mathbf{x}_{t,\widehat{i}}; \boldsymbol{\theta}_{t-1}) - f(\mathbf{x}_{t,i^\circ}; \boldsymbol{\theta}_{t-1})] - 2\min\{\boldsymbol{\beta}_t, 1\} \\ &\geq 2\gamma\boldsymbol{\beta}_t - 2\min\{\boldsymbol{\beta}_t, 1\} \\ &\geq 0 \end{aligned}$$

(7.45)

where the last inequality is because of $\gamma \geq 1$. Then, similarly, for any $i' \in ([k] \setminus \{\widehat{i}, i^\circ\})$, we have $\mathop{\mathbb{E}}_{\mathbf{x}_t \sim \mathcal{D}_{\mathcal{X}}}[h(\mathbf{x}_{t,\widehat{i}}) - h(\mathbf{x}_{t,i'})] \geq 0$. Thus, based on the definition of $h(\mathbf{x}_{t,i^*})$, we have $\mathop{\mathbb{E}}_{\mathbf{x}_t \sim \mathcal{D}_{\mathcal{X}}}[h(\mathbf{x}_{t,\widehat{i}})] = \mathop{\mathbb{E}}_{\mathbf{x}_t \sim \mathcal{D}_{\mathcal{X}}}[h(\mathbf{x}_{t,i^*})]$. Because $\mathop{\mathbb{E}}_{\mathbf{x}_t \sim \mathcal{D}_{\mathcal{X}}}[h(\mathbf{x}_{t,\widehat{i}})] = \mathop{\mathbb{E}}_{(\mathbf{x}_t,y_t) \sim \mathcal{D}}[1 - L(\mathbf{y}_{t,\widehat{i}}, \mathbf{y}_{t,y_t})]$, we have

$$\mathop{\mathbb{E}}_{(\mathbf{x}_t,y_t) \sim \mathcal{D}}[1 - L(\mathbf{y}_{t,\widehat{i}}, \mathbf{y}_{t,y_t})] = \mathop{\mathbb{E}}_{(\mathbf{x}_t,y_t) \sim \mathcal{D}}[1 - L(\mathbf{y}_{t,i^*}, \mathbf{y}_{t,y_t})]$$

$$\Rightarrow \mathop{\mathbb{E}}_{(\mathbf{x}_t,y_t) \sim \mathcal{D}}[L(\mathbf{y}_{t,\widehat{i}}, \mathbf{y}_{t,y_t})] = \mathop{\mathbb{E}}_{(\mathbf{x}_t,y_t) \sim \mathcal{D}}[L(\mathbf{y}_{t,i^*}, \mathbf{y}_{t,y_t})].$$

The proof is complete. $\qquad\square$

**Lemma 7.6.** *For any $\delta \in (0,1), \nu > 0$, suppose $m$ satisfies the conditions in Theorem 4.1. In round $t \in [T]$, given $(\mathbf{x}_t, y_t) \sim \mathcal{D}$, let*

$$\widehat{i} = \arg\max_{i \in [k]} \left( f_1(\mathbf{x}_{t,\widehat{i}}; \boldsymbol{\theta}_{t-1}^1) + f_2(\phi(\mathbf{x}_{t,\widehat{i}}); \boldsymbol{\theta}_{t-1}^2) \right).$$

*Then, with probability at least $1 - \delta$, we have*

$$\mathop{\mathbb{E}}_{(\mathbf{x}_t,y_t) \sim \mathcal{D}}\left[ \min\left\{ \left| f_1(\mathbf{x}_{t,\widehat{i}}; \boldsymbol{\theta}_{t-1}^1) + f_2(\phi(\mathbf{x}_{t,\widehat{i}}); \boldsymbol{\theta}_{t-1}^2) - r_{t,\widehat{i}}^1 \right|, 1 \right\} |\mathcal{H}_{t-1}^1 \right]$$

$$\leq \mathcal{O}\left( \frac{3L\nu + 2\sqrt{\mu}}{\sqrt{2t}} \right) + 2\sqrt{\frac{2\log(\mathcal{O}(1)/\delta)}{t}},$$

(7.46)

*where $\mathcal{H}_{t-1}^1 = \{\mathbf{x}_{\tau,\widehat{i}}, r_{\tau,\widehat{i}}^1\}_{\tau=1}^{t-1}$ is historical data and the expectation is taken over $(\boldsymbol{\theta}_{t-1}^1, \boldsymbol{\theta}_{t-1}^2)$.*

*Proof.* This lemma is inspired by Lemma 5.1 in [13]. For any round $\tau \in [t]$, define

$$\begin{aligned} V_\tau = \mathop{\mathbb{E}}_{(\mathbf{x}_\tau,y_\tau) \sim \mathcal{D}} &\left[ \min\{|f_1(\mathbf{x}_{\tau,\widehat{i}}; \widehat{\boldsymbol{\theta}}_{\tau-1}^1) + f_2(\phi(\mathbf{x}_{\tau,\widehat{i}}); \widehat{\boldsymbol{\theta}}_{\tau-1}^2) - r_{\tau,\widehat{i}}^1|, 1\} \right] \\ &- \min\{|f_1(\mathbf{x}_{\tau,\widehat{i}}; \widehat{\boldsymbol{\theta}}_{\tau-1}^1) + f_2(\phi(\mathbf{x}_{\tau,\widehat{i}}); \widehat{\boldsymbol{\theta}}_{\tau-1}^2) - r_{\tau,\widehat{i}}^1|, 1\} \end{aligned}$$

(7.47)

Then, we have

$$\begin{aligned} \mathbb{E}[V_\tau | F_{\tau-1}] = \mathop{\mathbb{E}}_{(\mathbf{x}_\tau,y_\tau) \sim \mathcal{D}} &\left[ \min\{|f_1(\mathbf{x}_{\tau,\widehat{i}}; \widehat{\boldsymbol{\theta}}_{\tau-1}^1) + f_2(\phi(\mathbf{x}_{\tau,\widehat{i}}); \widehat{\boldsymbol{\theta}}_{\tau-1}^2) - r_{\tau,\widehat{i}}^1|, 1\} \right] \\ &- \mathop{\mathbb{E}}_{(\mathbf{x}_\tau,y_\tau) \sim \mathcal{D}} \left[ \min\{|f_1(\mathbf{x}_{\tau,\widehat{i}}; \widehat{\boldsymbol{\theta}}_{\tau-1}^1) + f_2(\phi(\mathbf{x}_{\tau,\widehat{i}}); \widehat{\boldsymbol{\theta}}_{\tau-1}^2) - r_{\tau,\widehat{i}}^1|, 1\} \right] \\ &= 0 \end{aligned}$$

(7.48)

where $F_{\tau-1}$ denotes the $\sigma$-algebra generated by the history $\mathcal{H}^1_{\tau-1}$. Therefore, $\{V_\tau\}^t_{\tau=1}$ are the martingale difference sequence.

Then, applying the Hoeffding-Azuma inequality, with probability at least $1 - \delta$, we have

$$\mathbb{P}\left[\frac{1}{t}\sum_{\tau=1}^{t} V_\tau - \underbrace{\frac{1}{t}\sum_{\tau=1}^{t}\mathbb{E}[V_\tau|\mathbf{F}_{\tau-1}]}_{I_1} > \sqrt{\frac{2\log(1/\delta)}{t}}\right] \leq \delta \tag{7.49}$$

As $I_1$ is equal to 0, we have

$$\frac{1}{t}\sum_{\tau=1}^{t}\underset{(\mathbf{x}_\tau,y_\tau)\sim\mathcal{D}}{\mathbb{E}}\left[\min\{|f_1(\mathbf{x}_{\tau,\widehat{i}};\widehat{\boldsymbol{\theta}}^1_{\tau-1}) + f_2(\phi(\mathbf{x}_{\tau,\widehat{i}});\widehat{\boldsymbol{\theta}}^2_{\tau-1}) - r^1_{\tau,\widehat{i}}|, 1\}\right]$$

$$\leq \frac{1}{t}\sum_{\tau=1}^{t}\min\{|f_1(\mathbf{x}_{\tau,\widehat{i}};\widehat{\boldsymbol{\theta}}^1_{\tau-1}) + f_2(\phi(\mathbf{x}_{\tau,\widehat{i}});\widehat{\boldsymbol{\theta}}^2_{\tau-1}) - r^1_{\tau,\widehat{i}}|, 1\} + \sqrt{\frac{2\log(1/\delta)}{t}} \tag{7.50}$$

$$\leq \frac{1}{t}\sum_{\tau=1}^{t}\left|f_2\left(\mathbf{x}_{\tau,\widehat{i}};\widehat{\boldsymbol{\theta}}^2_{\tau-1}\right) - \left(r^1_{\tau,\widehat{i}} - f_1(\mathbf{x}_{\tau,\widehat{i}};\widehat{\boldsymbol{\theta}}^1_{\tau-1})\right)\right| + \sqrt{\frac{2\log(1/\delta)}{t}}.$$

Based on the the definition of $\boldsymbol{\theta}^1_{t-1}, \boldsymbol{\theta}^2_{t-1}$, we have

$$\underset{(\mathbf{x}_t,y_t)\sim\mathcal{D}(\boldsymbol{\theta}^1,\boldsymbol{\theta}^2)}{\mathbb{E}}\underset{}{\mathbb{E}}\left[\min\{|f_1(\mathbf{x}_{t,\widehat{i}};\boldsymbol{\theta}^1_{t-1}) + f_2(\mathbf{x}_{t,\widehat{i}};\boldsymbol{\theta}^2_{t-1}) - r^1_{t,\widehat{i}}|, 1\}\right]$$

$$= \frac{1}{t}\sum_{\tau=1}^{t}\underset{(\mathbf{x}_\tau,y_\tau)\sim\mathcal{D}}{\mathbb{E}}\left[\min\{\left|f_1(\mathbf{x}_{\tau,\widehat{i}};\widehat{\boldsymbol{\theta}}^1_{\tau-1}) + f_2(\mathbf{x}_{\tau,\widehat{i}};\widehat{\boldsymbol{\theta}}^2_{\tau-1}) - r^1_{\tau,\widehat{i}}\right|, 1\}\right]. \tag{7.51}$$

Therefore, putting them together, we have

$$\underset{(\mathbf{x}_t,y_t)\sim\mathcal{D}(\boldsymbol{\theta}^1,\boldsymbol{\theta}^2)}{\mathbb{E}}\underset{}{\mathbb{E}}\left[|f_1(\mathbf{x}_{t,\widehat{i}};\boldsymbol{\theta}^1_{t-1}) + f_2(\mathbf{x}_{t,\widehat{i}};\boldsymbol{\theta}^2_{t-1}) - r^1_{t,\widehat{i}}|\right]$$

$$\leq \underbrace{\frac{1}{t}\sum_{\tau=1}^{t}\left|f_2\left(\mathbf{x}_{\tau,\widehat{i}};\widehat{\boldsymbol{\theta}}^2_{\tau-1}\right) - \left(r^1_{\tau,\widehat{i}} - f_1(\mathbf{x}_{\tau,\widehat{i}};\widehat{\boldsymbol{\theta}}^1_{\tau-1})\right)\right|}_{I_2} + \sqrt{\frac{2\log(1/\delta)}{t}}. \tag{7.52}$$

For $I_2$, based on Lemma 7.8, we have

$$I_2 \leq \frac{1}{t}\sum_{\tau=1}^{t}\left|f_2\left(\mathbf{x}_{\tau,\widehat{i}};\widetilde{\boldsymbol{\theta}}^2\right) - \left(r^1_{\tau,\widehat{i}} - f_1(\mathbf{x}_{\tau,\widehat{i}};\widehat{\boldsymbol{\theta}}^1_{\tau-1})\right)\right| + \mathcal{O}\left(\frac{3L\nu}{\sqrt{2t}}\right) + \sqrt{\frac{2\log(1/\delta)}{t}}$$

$$\leq \frac{1}{t}\sqrt{t}\sqrt{\underbrace{\sum_{\tau=1}^{t}\left(f_2\left(\mathbf{x}_{\tau,\widehat{i}};\widetilde{\boldsymbol{\theta}}^2\right) - \left(r^1_{\tau,\widehat{i}} - f_1(\mathbf{x}_{\tau,\widehat{i}};\widehat{\boldsymbol{\theta}}^1_{\tau-1})\right)\right)^2}_{I_3}} + \mathcal{O}\left(\frac{3L\nu}{\sqrt{2t}}\right) + \sqrt{\frac{2\log(1/\delta)}{t}}$$

$$\leq \sqrt{\frac{2\mu}{t}} + \mathcal{O}\left(\frac{3L\nu}{\sqrt{2t}}\right) + \sqrt{\frac{2\log(1/\delta)}{t}} \tag{7.53}$$

where $I_3$ is based on the assumption of $\mu$.

Combining above Eq. (7.52) and (7.53) together, with probability at least $1 - \delta$, we have

$$\underset{(\mathbf{x}_t,y_t)\sim\mathcal{D}}{\mathbb{E}}\left[\min\left\{\left|f_1(\mathbf{x}_{t,\widehat{i}};\boldsymbol{\theta}^1_{t-1}) + f_2(\phi(\mathbf{x}_{t,\widehat{i}});\boldsymbol{\theta}^2_{t-1}) - r^1_{t,\widehat{i}}\right|, 1\right\}\right]$$

$$\leq \mathcal{O}\left(\frac{3L\nu + 2\sqrt{\mu}}{\sqrt{2t}}\right) + 2\sqrt{\frac{2\log(\mathcal{O}(1)/\delta)}{t}}. \tag{7.54}$$

where we apply union bound over $\delta$ to make above events occur concurrently.

Then, based on Lemma 7.13 (2), it is sufficient to show that $\boldsymbol{\theta}_{t-1}^1, \boldsymbol{\theta}_{t-1}^2$ are close to initialization for any $t \in [T]$. The proof is complete. $\qquad\square$

**Lemma 7.7.** *In round* $t \in [T]$, *given* $(\mathbf{x}_t, y_t) \sim \mathcal{D}$, *let* $i^* = \arg\max_{i \in [k]} h(\mathbf{x}_{t,i})$. *Let* $\boldsymbol{\theta}_{t-1}^{1,*}, \boldsymbol{\theta}_{t-1}^{2,*}$ *are the parameters trained on* $\mathcal{H}_{t-1}^*$ *using Algorithm 2. For any* $\nu > 0$, *suppose* $\inf_{\widetilde{\boldsymbol{\theta}}^{2,*} \in \mathcal{B}(\theta_0^2, \nu)} \frac{1}{2} \sum_{\tau=1}^t \left( f_1(\mathbf{x}_{\tau,i^*}; \widehat{\boldsymbol{\theta}}_{\tau-1}^{1,*}) + f_2\left(\phi(\mathbf{x}_{\tau,i^*}); \widetilde{\boldsymbol{\theta}}^{2,*}\right) - r_{\tau,i^*}^1 \right)^2 \leq \mu$. *Then, with probability at least* $1 - \delta$, *we have*

$$
\underset{(\mathbf{x}_t, y_t) \sim \mathcal{D}}{\mathbb{E}} \left[ \min \left\{ \left| f_1(\mathbf{x}_{t,i^*}; \boldsymbol{\theta}_{t-1}^{1,*}) + f_2(\phi(\mathbf{x}_{t,i^*}); \boldsymbol{\theta}_{t-1}^{2,*}) - r_{t,i^*}^1 \right|, 1 \right\} | \mathcal{H}_{t-1}^* \right]
$$
$$
\leq \mathcal{O}\left( \frac{3L\nu + 2\sqrt{\mu}}{\sqrt{2t}} \right) + 2\sqrt{\frac{2\log(\mathcal{O}(1)/\delta)}{t}}, \tag{7.55}
$$

*where* $\mathcal{H}_{t-1}^* = \{\mathbf{x}_{\tau,i^*}, r_{\tau,i^*}^1\}_{\tau=1}^{t-1}$ *is optimal data of past rounds the expectation is taken over* $\boldsymbol{\theta}_{t-1}^{1,*}$, $\boldsymbol{\theta}_{t-1}^{2,*}$.

*Proof.* This lemma is a direct corollary of Lemma 7.6. For any $\tau \in [t]$, define

$$
V_\tau = \underset{(\mathbf{x}_\tau, y_\tau) \sim \mathcal{D}}{\mathbb{E}} \left[ \min\{|f_1(\mathbf{x}_{\tau,i^*}; \widehat{\boldsymbol{\theta}}_{\tau-1}^{1,*}) + f_2(\phi(\mathbf{x}_{\tau,i^*}); \widehat{\boldsymbol{\theta}}_{\tau-1}^{2,*}) - r_{\tau,i^*}^1|, 1\} \right]
$$
$$
- \min\{|f_1(\mathbf{x}_{\tau,i^*}; \widehat{\boldsymbol{\theta}}_{\tau-1}^{1,*}) + f_2(\phi(\mathbf{x}_{\tau,i^*}); \widehat{\boldsymbol{\theta}}_{\tau-1}^{2,*}) - r_{\tau,i^*}^1|, 1\} \tag{7.56}
$$

Then, we have

$$
\mathbb{E}[V_\tau | F_{\tau-1}] = \underset{(\mathbf{x}_\tau, y_\tau) \sim \mathcal{D}}{\mathbb{E}} \left[ \min\{|f_1(\mathbf{x}_{\tau,i^*}; \widehat{\boldsymbol{\theta}}_{\tau-1}^{1,*}) + f_2(\phi(\mathbf{x}_{\tau,i^*}); \widehat{\boldsymbol{\theta}}_{\tau-1}^{2,*}) - r_{\tau,i^*}^1|, 1\} \right]
$$
$$
- \underset{(\mathbf{x}_\tau, y_\tau) \sim \mathcal{D}}{\mathbb{E}} \left[ \min\{|f_1(\mathbf{x}_{\tau,i^*}; \widehat{\boldsymbol{\theta}}_{\tau-1}^{1,*}) + f_2(\phi(\mathbf{x}_{\tau,i^*}); \widehat{\boldsymbol{\theta}}_{\tau-1}^{2,*}) - r_{\tau,i^*}^1|, 1\} \right] \tag{7.57}
$$
$$
= 0
$$

where $F_{\tau-1}$ denotes the $\sigma$-algebra generated by the history $\mathcal{H}_{\tau-1}$. Therefore, $\{V_\tau\}_{\tau=1}^t$ are the martingale difference sequence.

Then, applying the Hoeffding-Azuma inequality, with probability at least $1 - \delta$, we have

$$
\mathbb{P}\left[ \frac{1}{t} \sum_{\tau=1}^t V_\tau - \underbrace{\frac{1}{t} \sum_{\tau=1}^t \mathbb{E}[V_\tau | \mathbf{F}_\tau]}_{I_1} > \sqrt{\frac{2\log(1/\delta)}{t}} \right] \leq \delta \tag{7.58}
$$

As $I_1$ is equal to 0, we have

$$
\underset{(\mathbf{x}_t, y_t) \sim \mathcal{D}(\boldsymbol{\theta}^1, \boldsymbol{\theta}^2)}{\mathbb{E}} \underset{}{\mathbb{E}} \left[ \min\{|f_1(\mathbf{x}_{t,i^*}; \boldsymbol{\theta}_{t-1}^1) + f_2(\phi(\mathbf{x}_{t,i^*}); \boldsymbol{\theta}_{t-1}^2) - r_{t,i^*}^1|, 1\} \right]
$$
$$
= \frac{1}{t} \sum_{\tau=1}^t \underset{(\mathbf{x}_\tau, y_\tau) \sim \mathcal{D}}{\mathbb{E}} \left[ \min\{|f_1(\mathbf{x}_{\tau,i^*}; \widehat{\boldsymbol{\theta}}_{\tau-1}^{1,*}) + f_2(\phi(\mathbf{x}_{\tau,i^*}); \widehat{\boldsymbol{\theta}}_{\tau-1}^{2,*}) - r_{\tau,i^*}^1, 1\}| \right]
$$
$$
\leq \underbrace{\frac{1}{t} \sum_{\tau=1}^t \left| f_2\left(\phi(\mathbf{x}_{\tau,i^*}); \widehat{\boldsymbol{\theta}}_{\tau-1}^{2,*}\right) - \left(r_{\tau,i^*}^1 - f_1(\mathbf{x}_{\tau,i^*}; \widehat{\boldsymbol{\theta}}_{\tau-1}^{1,*})\right) \right|}_{I_2} + \sqrt{\frac{2\log(1/\delta)}{t}}. \tag{7.59}
$$

For $I_2$, applying Lemma 7.8, for any $\widetilde{\boldsymbol{\theta}}^{2,*}$ satisfying $\|\widetilde{\boldsymbol{\theta}}^{2,*} - \boldsymbol{\theta}_0^2\|_2 \leq \mathcal{O}(\frac{\nu}{\sqrt{m}})$, with probability at least $1 - 3\delta$, we have

$$
\begin{aligned}
I_2 &\leq \frac{1}{t}\sum_{\tau=1}^{t} |f_1(\mathbf{x}_{\tau,i^*}; \widehat{\boldsymbol{\theta}}_{\tau-1}^{1,*}) + f_2\left(\phi(\mathbf{x}_{\tau,i^*}); \widetilde{\boldsymbol{\theta}}^{2,*}\right) - r_{\tau,i^*}^1| + \mathcal{O}\left(\frac{3L\nu}{\sqrt{2t}}\right) + \sqrt{\frac{2\log(1/\delta)}{t}} \\
&\leq \frac{1}{t}\sqrt{t}\sqrt{\underbrace{\sum_{\tau=1}^{t}\left(f_1(\mathbf{x}_{\tau,i^*}; \widehat{\boldsymbol{\theta}}_{\tau-1}^{1,*}) + f_2\left(\phi(\mathbf{x}_{\tau,i^*}); \widetilde{\boldsymbol{\theta}}^{2,*}\right) - r_{\tau,i^*}^1\right)^2}_{I_3}} + \mathcal{O}\left(\frac{3L\nu}{\sqrt{2t}}\right) + \sqrt{\frac{2\log(1/\delta)}{t}} \\
&\leq \sqrt{\frac{2\mu}{t}} + \mathcal{O}\left(\frac{3L\nu}{\sqrt{2t}}\right) + \sqrt{\frac{2\log(1/\delta)}{t}}
\end{aligned}
$$
(7.60)

where $I_3$ is because of the assumption of $\mu$.

Combining above inequalities together, as $\mu \in (0,1]$, with probability at least $1 - \delta$, we have

$$
\mathop{\mathbb{E}}_{(\mathbf{x}_t, y_t) \sim \mathcal{D}}\left[\min\left\{\left|f_1(\mathbf{x}_{t,i^*}; \boldsymbol{\theta}_{t-1}^{1,*}) + f_2(\phi(\mathbf{x}_{t,i^*}); \boldsymbol{\theta}_{t-1}^{2,*}) - r_{t,i^*}^1\right|, 1\right\}\right]
$$
$$
\leq \mathcal{O}\left(\frac{3L\nu + 2\sqrt{\mu}}{\sqrt{2t}}\right) + 2\sqrt{\frac{2\log(\mathcal{O}(1)/\delta)}{t}},
$$
(7.61)

where we apply union bound over $\delta$ to make above events occur concurrently. Then, based on Lemma 7.13 (2), it is sufficient to show that $\boldsymbol{\theta}_{t-1}^{1,*}, \boldsymbol{\theta}_{t-1}^{2,*}$ are close to initialization for any $t \in [T]$. $\qquad \square$

**Lemma 7.8.** *For any $\delta \in (0,1)$, suppose $m$ satisfies the condition in Theorem 4.1. Then, with probability at least $1 - \delta$, setting $\eta_2 = \frac{\kappa\nu}{m\sqrt{t}}$ for algorithm 1, for $\nu > 0$ and any $\widetilde{\boldsymbol{\theta}}^2$ satisfying $\|\widetilde{\boldsymbol{\theta}}^2 - \boldsymbol{\theta}_0^2\|_2 \leq \mathcal{O}(\frac{\nu}{\sqrt{m}})$, and $i \in [k]$, there exists a small enough constant $\kappa$, such that*

$$
\sum_{\tau=1}^{t}\left|f_2\left(\phi(\mathbf{x}_{\tau,i}); \widehat{\boldsymbol{\theta}}_{\tau-1}^2\right) - \left(r_{\tau,i}^1 - f_1(\mathbf{x}_{\tau,i}; \widehat{\boldsymbol{\theta}}_{\tau-1}^1)\right)\right|
$$
$$
\leq \sum_{\tau=1}^{t}\left|f_2\left(\phi(\mathbf{x}_{\tau,i}); \widetilde{\boldsymbol{\theta}}^2\right) - \left(r_{\tau,i}^1 - f_1(\mathbf{x}_{\tau,i}; \widehat{\boldsymbol{\theta}}_{\tau-1}^1)\right)\right| + \mathcal{O}\left(\frac{3L\nu\sqrt{t}}{\sqrt{2}}\right) + \sqrt{2t\log(1/\delta)}.
$$

*Proof.* This is a direct application of Lemma 7.9 by setting $\hat{\epsilon} = \frac{L\nu}{\sqrt{2\kappa t}}$, and, where $\kappa$ is some small enough absolute constant. We set $L_\tau(\widehat{\boldsymbol{\theta}}_{\tau-1}^2) = \left|f_2(\phi(\mathbf{x}_{\tau,i}); \widehat{\boldsymbol{\theta}}_{\tau-1}^2) - \left(r_{\tau,i}^1 - f_1(\mathbf{x}_{\tau,i}; \widehat{\boldsymbol{\theta}}_{\tau-1}^1)\right)\right|$. Then, for any $\widetilde{\boldsymbol{\theta}}^2$ satisfying $\|\widetilde{\boldsymbol{\theta}}^2 - \boldsymbol{\theta}_0^2\|_2 \leq \mathcal{O}(\frac{\nu}{\sqrt{m}})$, there exist a small enough absolute constant $\kappa$, such that

$$
\sum_{\tau=1}^{t} L_\tau(\widehat{\boldsymbol{\theta}}_{\tau-1}^2) \leq \sum_{\tau=1}^{t} L_\tau(\widetilde{\boldsymbol{\theta}}^2) + \mathcal{O}(3t\hat{\epsilon}) + \sqrt{2t\log(1/\delta)}.
$$
(7.62)

Then, replacing $\hat{\epsilon}$ completes the proof. $\qquad \square$

**Lemma 7.9.** *With probability at least $1 - \delta$ over the randomness of $\boldsymbol{\theta}_0$, given the convex loss $L$ satisfying $L' \leq \mathcal{O}(1)$, for any $\hat{\epsilon}, \nu > 0$ and $\widetilde{\boldsymbol{\theta}}$ satisfying $\|\widehat{\boldsymbol{\theta}} - \boldsymbol{\theta}_0\|_2 \leq \mathcal{O}(\frac{\nu}{\sqrt{m}})$, Algorithm 1 with $\eta = \frac{\kappa\hat{\epsilon}}{Lm}$ and $t = \frac{L^2\nu^2}{2\kappa\hat{\epsilon}^2}$ for some small enough constant $\kappa$ has the following bound:*

$$
\sum_{\tau=1}^{t} \min\{L_{(\mathbf{x}_\tau, r_\tau)}(\widehat{\boldsymbol{\theta}}_{\tau-1}) - L_{(\mathbf{x}_\tau, r)}(\widetilde{\boldsymbol{\theta}}), 1\} \leq \mathcal{O}(3t\hat{\epsilon}) + \sqrt{2t\log(1/\delta)}.
$$
(7.63)

*Proof.* Define $\mathcal{B}(\boldsymbol{\theta}_0, \nu) = \{\boldsymbol{\theta} \in \mathbb{R}^p : \|\boldsymbol{\theta} - \boldsymbol{\theta}_0\|_2 \leq \mathcal{O}(\nu/\sqrt{m})\}$ and $L_{(\mathbf{x};r)}(\boldsymbol{\theta}) = |r - f(\mathbf{x}; \boldsymbol{\theta})|$. First, we need to show $\widehat{\boldsymbol{\theta}}_1, \ldots, \widehat{\boldsymbol{\theta}}_t$ also are in $\mathcal{B}(\boldsymbol{\theta}_0, w)$, where. According to Lemma 7.14, when

$\boldsymbol{\theta} \in \mathcal{B}(\boldsymbol{\theta}_0, w)$, we have

$$\|\nabla_{\boldsymbol{\theta}} f(\mathbf{x}; \boldsymbol{\theta})\|_2 \leq \mathcal{O}(L), \ \|\nabla_{\boldsymbol{\theta}} L_{(\mathbf{x};r)}(\boldsymbol{\theta})\|_2 \leq \sqrt{\sum_{l=1}^{L} \|\mathcal{O}(\nabla_{\mathbf{W}_l} f(\mathbf{x}; \boldsymbol{\theta}))\|_2^2} \leq \mathcal{O}(L). \quad (7.64)$$

The proof follows a simple induction. Suppose that $\boldsymbol{\theta}_0, \widehat{\boldsymbol{\theta}}_1, \ldots, \widehat{\boldsymbol{\theta}}_t \in \mathcal{B}(\boldsymbol{\theta}_0, w)$, by triangle inequality, we have

$$\|\widehat{\boldsymbol{\theta}}_t - \boldsymbol{\theta}_0\|_2 \leq \sum_{\tau=0}^{t-1} \|\widehat{\boldsymbol{\theta}}_{\tau+1} - \widehat{\boldsymbol{\theta}}_\tau\|_2 \leq \frac{1}{|\widehat{\mathcal{H}}_t|} \sum_{\tau=0}^{t-1} \sum_{(\mathbf{x},r) \in \widehat{\mathcal{H}}_t} \|\nabla_{\boldsymbol{\theta}} L(\mathbf{x}; r)\|_2 \leq \mathcal{O}(L\eta t). \quad (7.65)$$

Because $\eta = \mathcal{O}(\frac{1}{m})$, we have $\|\widehat{\boldsymbol{\theta}}_t - \boldsymbol{\theta}_0\|_2 \leq \mathcal{O}(\nu/\sqrt{m})$. In round $\tau \in [t]$, recall that $|\widehat{\mathcal{H}}_\tau| = b$ and $\widehat{\mathcal{H}}_\tau \subset \mathcal{H}_\tau$. Given the context $\mathbf{x}$ and its reward $r$, we have the fact

$$\tau \mathop{\mathbb{E}}_{\widehat{\mathcal{H}}_\tau} \left[ \frac{1}{b} \sum_{(\mathbf{x},r) \in \widehat{\mathcal{H}}_\tau} \nabla_{\widehat{\boldsymbol{\theta}}} L_{(\mathbf{x},r)}(\widehat{\boldsymbol{\theta}}^{t-1}) \right] \overset{E_1}{=} \tau \mathop{\mathbb{E}}_{(\mathbf{x},r) \sim \mathcal{H}_\tau} \left[ \nabla_{\widehat{\boldsymbol{\theta}}} L_{(\mathbf{x},r)}(\widehat{\boldsymbol{\theta}}^{t-1}) \right]$$

$$= \tau \sum_{(\mathbf{x},r) \in \mathcal{H}_\tau} \frac{1}{|\mathcal{H}_\tau|} \nabla_{\widehat{\boldsymbol{\theta}}} L_{(\mathbf{x},r)}(\widehat{\boldsymbol{\theta}}^{t-1}) \quad (7.66)$$

$$\overset{E_2}{=} \sum_{(\mathbf{x},r) \in \mathcal{H}_\tau} \nabla_{\widehat{\boldsymbol{\theta}}} L_{(\mathbf{x},r)}(\widehat{\boldsymbol{\theta}}^{t-1}),$$

where $E_1$ is because $\widehat{\mathcal{H}}_\tau$ is uniformly drawn from $\mathcal{H}_\tau$ and $E_2$ is duo to $|\mathcal{H}_\tau| = \tau$. Then, based on Lemma 7.12, for any $\epsilon > 0$, we have

$$\mathop{\mathbb{E}}_{(\mathbf{x},r) \sim \mathcal{H}_\tau} [L_{(\mathbf{x},r)}(\widehat{\boldsymbol{\theta}}_{\tau-1}) - L_{(\mathbf{x},r)}(\widetilde{\boldsymbol{\theta}})] \leq \langle \mathop{\mathbb{E}}_{(\mathbf{x},r) \sim \mathcal{H}_\tau} [\nabla_{\widehat{\boldsymbol{\theta}}} L_\tau(\widehat{\boldsymbol{\theta}}_{\tau-1})], \widehat{\boldsymbol{\theta}}_{\tau-1} - \widetilde{\boldsymbol{\theta}} \rangle + \hat{\epsilon}$$

$$\leq \left\langle \mathop{\mathbb{E}}_{\widehat{\mathcal{H}}_\tau} \left[ \frac{1}{b} \sum_{(\mathbf{x},r) \in \widehat{\mathcal{H}}_\tau} \nabla_{\widehat{\boldsymbol{\theta}}} L_{(\mathbf{x},r)}(\widehat{\boldsymbol{\theta}}^{t-1}) \right], \widehat{\boldsymbol{\theta}}_{\tau-1} - \widetilde{\boldsymbol{\theta}} \right\rangle + \hat{\epsilon}$$

$$= \frac{\langle \widehat{\boldsymbol{\theta}}_{\tau-1} - \mathop{\mathbb{E}}_{\widehat{\mathcal{H}}_\tau} [\widehat{\boldsymbol{\theta}}_\tau], \widehat{\boldsymbol{\theta}}_{\tau-1} - \widetilde{\boldsymbol{\theta}} \rangle}{\eta} + \hat{\epsilon}. \quad (7.67)$$

Based on the fact $2 \langle \mathbf{A}, \mathbf{B} \rangle = \|\mathbf{A}\|_2^2 + \|\mathbf{B}\|_2^2 - \|\mathbf{A}, \mathbf{B}\|_2^2$, we have

$$\mathop{\mathbb{E}}_{(\mathbf{x},r) \sim \mathcal{H}_\tau} [L_{(\mathbf{x},r)}(\widehat{\boldsymbol{\theta}}_{\tau-1}) - L_{(\mathbf{x},r)}(\widetilde{\boldsymbol{\theta}})] \leq \frac{\|\widehat{\boldsymbol{\theta}}_{\tau-1} - \mathop{\mathbb{E}}_{\widehat{\mathcal{H}}_\tau} [\widehat{\boldsymbol{\theta}}_\tau]\|_2^2 + \|\widehat{\boldsymbol{\theta}}_{\tau-1} - \widetilde{\boldsymbol{\theta}}\|_2^2 - \|\mathop{\mathbb{E}}_{\widehat{\mathcal{H}}_\tau} [\widehat{\boldsymbol{\theta}}_\tau] - \widetilde{\boldsymbol{\theta}}\|_2^2}{2\eta} + \hat{\epsilon}$$

$$\overset{E_3}{\leq} \frac{\|\widehat{\boldsymbol{\theta}}_{\tau-1} - \widetilde{\boldsymbol{\theta}}\|_2^2 - \|\mathop{\mathbb{E}}_{\widehat{\mathcal{H}}_\tau} [\widehat{\boldsymbol{\theta}}_\tau] - \widetilde{\boldsymbol{\theta}}\|_2^2}{2\eta} + \mathcal{O}(L^2 \eta) + \hat{\epsilon} \quad (7.68)$$

where $E_3$ is because of 7.66:

$$\|\widehat{\boldsymbol{\theta}}_{\tau-1} - \mathop{\mathbb{E}}_{\widehat{\mathcal{H}}_\tau} [\widehat{\boldsymbol{\theta}}_\tau]\|_2 = \eta \left\| \mathop{\mathbb{E}}_{\widehat{\mathcal{H}}_\tau} \left[ \frac{1}{b} \sum_{(\mathbf{x},r) \in \widehat{\mathcal{H}}_\tau} \nabla_{\widehat{\boldsymbol{\theta}}} L_{(\mathbf{x},r)}(\widehat{\boldsymbol{\theta}}^{t-1}) \right] \right\|_2$$

$$= \eta \frac{1}{\tau} \| \sum_{(\mathbf{x},r) \in \mathcal{H}_\tau} \nabla_{\widehat{\boldsymbol{\theta}}} L_{(\mathbf{x},r)}(\widehat{\boldsymbol{\theta}}^{t-1})\|_2 \leq \mathcal{O}(\eta L). \quad (7.69)$$

Therefore, we have

$$\sum_{\tau=1}^{t} \mathop{\mathbb{E}}_{(\mathbf{x},r)\sim\mathcal{H}_\tau}[L_{(\mathbf{x},r)}(\widehat{\boldsymbol{\theta}}_{\tau-1}) - L_{(\mathbf{x},r)}(\widetilde{\boldsymbol{\theta}})] \leq \frac{\|\widehat{\boldsymbol{\theta}}_0 - \widetilde{\boldsymbol{\theta}}\|_2^2 - \|\underset{\mathcal{H}_\tau}{\mathbb{E}}[\widehat{\boldsymbol{\theta}}_t] - \widetilde{\boldsymbol{\theta}}\|_2^2}{2\eta} + \mathcal{O}(tL^2\eta) + t\hat{\epsilon}$$

$$\leq \frac{\|\widehat{\boldsymbol{\theta}}_0 - \widetilde{\boldsymbol{\theta}}\|_2^2}{2\eta} + \mathcal{O}(tL^2\eta) + t\hat{\epsilon} \tag{7.70}$$

$$\leq \frac{LR^2}{2\eta m} + \mathcal{O}(tL^2\eta) + t\hat{\epsilon}.$$

Then, for $\tau \in [t]$, define

$$V_\tau = \min\{L_{(\mathbf{x},r)}(\widehat{\boldsymbol{\theta}}_{\tau-1}) - L_{(\mathbf{x},r)}(\widetilde{\boldsymbol{\theta}}), 1\} - \mathop{\mathbb{E}}_{(\mathbf{x},r)\sim\mathcal{H}_\tau}[\min\{L_{(\mathbf{x},r)}(\widehat{\boldsymbol{\theta}}_{\tau-1}) - L_{(\mathbf{x},r)}(\widetilde{\boldsymbol{\theta}}), 1\}]. \tag{7.71}$$

Then, we have

$$\mathbb{E}[V_\tau|\mathcal{F}_{\tau-1}] = \mathop{\mathbb{E}}_{(\mathbf{x},r)\sim\mathcal{H}_\tau}[\min\{L_{(\mathbf{x},r)}(\widehat{\boldsymbol{\theta}}_{\tau-1}), -L_{(\mathbf{x},r)}(\widetilde{\boldsymbol{\theta}}), 1\}]$$

$$- \mathop{\mathbb{E}}_{(\mathbf{x},r)\sim\mathcal{H}_\tau}[\min\{L_{(\mathbf{x},r)}(\widehat{\boldsymbol{\theta}}_{\tau-1}) - L_{(\mathbf{x},r)}(\widetilde{\boldsymbol{\theta}}), 1\}] = 0, \tag{7.72}$$

where where $F_{\tau-1}$ denotes the $\sigma$-algebra generated by the history $\mathcal{H}_{\tau-1}$. Therefore, $\{V_0, \ldots, V_t\}$ is the martingale difference sequence. Then, applying the Hoeffding-Azuma inequality, with probability at least $1 - \delta$, we have

$$\sum_{\tau=1}^{t} \min\{L_{(\mathbf{x},r)}(\widehat{\boldsymbol{\theta}}_{\tau-1}) - L_{(\mathbf{x}_\tau,r)}(\widetilde{\boldsymbol{\theta}}), 1\}$$

$$\leq \sum_{\tau=1}^{t} \mathop{\mathbb{E}}_{(\mathbf{x},r)\sim\mathcal{H}_\tau}[\min\{L_{(\mathbf{x},r)}(\widehat{\boldsymbol{\theta}}_{\tau-1}) - L_{(\mathbf{x},r)}(\widetilde{\boldsymbol{\theta}}), 1\}] + t\sqrt{\frac{2\log(1/\delta)}{t}} \tag{7.73}$$

$$\stackrel{E_4}{\leq} \frac{LR^2}{2\eta m} + \mathcal{O}(tL^2\eta) + t\hat{\epsilon} + \sqrt{2t\log(1/\delta)}$$

$$\stackrel{E_5}{\leq} \mathcal{O}(3t\hat{\epsilon}) + \sqrt{2t\log(1/\delta)}$$

where $E_4$ be because of 7.70 and $E_5$ is by placing the parameter choice $\eta = \frac{\kappa\hat{\epsilon}}{Lm}$ and $t = \frac{L^2\nu^2}{2\kappa\hat{\epsilon}}$. The proof is completed. $\qquad\square$

## 7.4 Ancillary Lemmas

**Lemma 7.10** (Theorem 5, [2]). *For any $\delta \in (0,1)$, if $w$ satisfies that*

$$\mathcal{O}(m^{-3/2}L^{-3/2}\max\{\log^{-3/2}m, \log^{3/2}(Tn/\delta)\}) \leq w \leq \mathcal{O}(L^{-9/2}\log^{-3}m), \tag{7.74}$$

*then, with probability at least $1 - \delta$, for all $\|\boldsymbol{\theta} - \boldsymbol{\theta}_0\|_2 \leq w$, we have*

$$\|\nabla_{\boldsymbol{\theta}} f(\mathbf{x}; \boldsymbol{\theta}) - \nabla_{\boldsymbol{\theta}_0} f(\mathbf{x}; \boldsymbol{\theta}_0)\|_2 \leq \mathcal{O}(\sqrt{\log m}w^{1/3}L^3)\|\nabla_{\boldsymbol{\theta}_0} f(\mathbf{x}; \boldsymbol{\theta}_0)\|_2. \tag{7.75}$$

**Lemma 7.11** (Lemma 4.1, [16]). *For any $\delta \in (0,1)$, if $w$ satisfies*

$$\mathcal{O}(m^{-3/2}L^{-3/2}[\log(tnL^2/\delta)]^{3/2}) \leq w \leq \mathcal{O}(L^{-6}[\log m]^{-3/2}),$$

*then, with probability at least $1 - \delta$ over randomness of $\boldsymbol{\theta}_0$, for any $t \in [T]$, $\|\mathbf{x}\|_2 = 1$, and $\boldsymbol{\theta}, \boldsymbol{\theta}'$ satisfying $\|\boldsymbol{\theta} - \boldsymbol{\theta}_0\|_2 \leq w$ and $\|\boldsymbol{\theta}' - \boldsymbol{\theta}_0\|_2 \leq w$, it holds uniformly that*

$$|f(\mathbf{x}_i; \boldsymbol{\theta}) - f(\mathbf{x}_i; \boldsymbol{\theta}') - \langle\nabla_{\boldsymbol{\theta}'} f(\mathbf{x}_i; \boldsymbol{\theta}'), \boldsymbol{\theta} - \boldsymbol{\theta}'\rangle| \leq \mathcal{O}(w^{1/3}L^2\sqrt{m\log(m)})\|\boldsymbol{\theta} - \boldsymbol{\theta}'\|_2. \tag{7.76}$$

**Lemma 7.12** (Lemma 4.2, [16]). *For any $\delta \in (0,1), \hat{\epsilon} > 0$, if $w$ satisfies*

$$\mathcal{O}(m^{-3/2}L^{-3/2}[\log(tnL^2/\delta)]^{3/2}) \leq w \leq \kappa L^{-6}m^{-3/8}[\log m]^{-3/2}\hat{\epsilon}^{3/4},$$

*then, with probability at least $1 - \delta$ over randomness of $\boldsymbol{\theta}^{(0)}$, for any $\hat{\epsilon} > 0, i \in [n]$, and $\boldsymbol{\theta}, \widetilde{\boldsymbol{\theta}}$ satisfying $\|\boldsymbol{\theta} - \boldsymbol{\theta}^{(0)}\|_2 \leq w$ and $\|\widetilde{\boldsymbol{\theta}} - \boldsymbol{\theta}^{(0)}\|_2 \leq w$, it holds uniformly that*

$$L_{(\mathbf{x},r)}(\widehat{\boldsymbol{\theta}}_{\tau-1}) - L_{(\mathbf{x},r)}(\widetilde{\boldsymbol{\theta}}) \leq \langle\nabla_{\widehat{\boldsymbol{\theta}}} L_{\mathbf{x},r}(\widehat{\boldsymbol{\theta}}_{\tau-1}), \widehat{\boldsymbol{\theta}}_{\tau-1} - \widetilde{\boldsymbol{\theta}}\rangle + \hat{\epsilon} \tag{7.77}$$

**Lemma 7.13.** *Given a constant $0 < \hat{\epsilon} < 1$, suppose $m$ satisfies the conditions in Lemma 4.1, the learning rate $\eta = \Omega(\frac{\rho}{poly(t,n,L)m})$, the number of iterations $K = \Omega(\frac{poly(t,n,L)}{\rho^2} \cdot \log \hat{\epsilon}^{-1})$. Then, with probability at least $1 - \delta$, starting from random initialization $\boldsymbol{\theta}_0$,*

(1) *(Theorem 1 in [2]) In round $t \in [T]$, given the collected data $\{\mathbf{x}_\tau, r_\tau\}_{i=\tau}^t$, the loss function is defined as: $\mathcal{L}(\boldsymbol{\theta}) = \frac{1}{2} \sum_{\tau=1}^t (f(\mathbf{x}_\tau; \boldsymbol{\theta}) - r_\tau)^2$. Then, there exists $\widetilde{\boldsymbol{\theta}}$ satisfying $\|\widetilde{\boldsymbol{\theta}} - \boldsymbol{\theta}_0\|_2 \leq \mathcal{O}\left(\frac{t^3}{\rho\sqrt{m}} \log m\right)$, such that $\mathcal{L}(\widetilde{\boldsymbol{\theta}}) \leq \hat{\epsilon}$ in $K = \Omega(\frac{poly(t,n,L)}{\rho^2} \cdot \log \hat{\epsilon}^{-1})$ iterations;*

(2) *For any $t \in [T]$, it holds uniformly that $\|\boldsymbol{\theta}_{t-1} - \boldsymbol{\theta}_0\|_2 \leq \mathcal{O}\left(\frac{t^3}{\rho\sqrt{m}} \log m\right)$;*

(3) *(Lemma C.4 in [13]) Following the initialization, given $\|\mathbf{x}\|_2 = 1$, it holds that*

$$\|\nabla_{\boldsymbol{\theta}_0} f(\mathbf{x}; \boldsymbol{\theta}_0)\|_2 \leq \mathcal{O}(L), \quad |f(\mathbf{x}; \boldsymbol{\theta}_0)| \leq \mathcal{O}(1).$$

*Proof.* (2) is a corollary of Theorem 1 in [2]. Suppose $\boldsymbol{\theta}_\tau = \boldsymbol{\theta}_{\tau-1} - \frac{\eta}{b} \sum_{(\mathbf{x},r) \in \widehat{\mathcal{H}}_\tau} \nabla_{\boldsymbol{\theta}} \mathcal{L}[(\mathbf{x}, r); \boldsymbol{\theta}_{\tau-1}]$. The proof is based on the following induction. Let $w = \mathcal{O}\left(\frac{t^3}{\delta\sqrt{m}} \log m\right)$. Then, based on the Theorem 1 in [2], we have

$$\mathcal{L}[(\mathbf{x}, r); \boldsymbol{\theta}_\tau] = \left(1 - \Omega\left(\frac{\eta\rho m}{t^2}\right)\right) \mathcal{L}[(\mathbf{x}, r); \boldsymbol{\theta}_{\tau-1}].$$

Then, we have

$$\|\boldsymbol{\theta}_t - \boldsymbol{\theta}_{t-1}\|_2 \leq \sum_{\tau=1} \|\frac{\eta}{b} \sum_{(\mathbf{x},r) \in \widehat{\mathcal{H}}_\tau} \nabla_{\boldsymbol{\theta}} \mathcal{L}[(\mathbf{x}, r); \boldsymbol{\theta}_{\tau-1}]\| \leq \mathcal{O}(\eta\sqrt{tm}) \sum_{\tau=1}^t \sqrt{\mathcal{L}[(\mathbf{x}, r); \boldsymbol{\theta}_{\tau-1}]}$$

$$\leq \mathcal{O}(\eta\sqrt{tm}) \cdot \Omega(\frac{t^2}{\eta\rho m}) \cdot \mathcal{O}(\sqrt{t \log^2 m}) \leq \mathcal{O}\left(\frac{t^3}{\rho\sqrt{m}} \log m\right).$$

$\square$

**Lemma 7.14** (Lemma C.2 [13]). *For any $\delta \in (0, 1), \nu > 0$, suppose $m$ satisfies the conditions in Theorem 4.1. Then, with probability at least $1 - \delta$, in each round $t \in [T]$, for any $\mathbf{x}$ satisfying $\|\mathbf{x}\|_2 = 1$, $\boldsymbol{\theta}_{t-1}^{1,*}, \boldsymbol{\theta}_{t-1}^1$ satisfying $\|\boldsymbol{\theta}_{t-1}^{1,*} - \boldsymbol{\theta}_{t-1}^1\|_2 \leq \mathcal{O}\left(\frac{\nu}{\sqrt{m}}\right)$, and $\boldsymbol{\theta}_{t-1}^{2,*}, \boldsymbol{\theta}_{t-1}^2$ satisfying $\|\boldsymbol{\theta}_{t-1}^{2,*} - \boldsymbol{\theta}_{t-1}^2\|_2 \leq \mathcal{O}\left(\frac{\nu}{\sqrt{m}}\right)$, we have*

$$
\begin{align}
(1) \quad & |f_1(\mathbf{x}; \boldsymbol{\theta}_{t-1}^{1,*}) - f_1(\mathbf{x}; \boldsymbol{\theta}_{t-1}^1)| \notag \\
& \leq \mathcal{O}\left(\frac{\nu L}{\sqrt{m}}\right) + \mathcal{O}\left(\frac{L^2 \sqrt{\log m} \nu^{4/3}}{m^{1/6}}\right) := \xi_1; \tag{7.78}
\end{align}
$$

$$
\begin{align}
(2) \quad & \left|f_2\left(\phi(\mathbf{x}); \boldsymbol{\theta}_{t-1}^{2,*}\right) - f_2\left(\phi(\mathbf{x}); \boldsymbol{\theta}_{t-1}^2\right)\right| \notag \\
& \leq \left(\frac{\nu L}{\sqrt{m}}\right) + \mathcal{O}\left(\frac{L^2 \sqrt{\log m} \nu^{4/3}}{m^{1/6}}\right); \tag{7.79}
\end{align}
$$

$$
\begin{align}
(3) \quad & \|\nabla_{\boldsymbol{\theta}_{t-1}^1} f_1(\mathbf{x}; \boldsymbol{\theta}_{t-1}^1)\|_2, \|\nabla_{\boldsymbol{\theta}_{t-1}^2} f_2\left(\phi(\mathbf{x}); \boldsymbol{\theta}_{t-1}^2\right)\|_2 \notag \\
& \leq \mathcal{O}(L). \tag{7.80}
\end{align}
$$

**Algorithm 3** Batch-GD-Warm-Start ( $f_1$, $f_2$, $\mathcal{H}_t^1, \mathcal{H}_t^2$)

---

1: Define $\mathcal{L}_1[(\mathbf{x}, r^1); \boldsymbol{\theta}^1] = (r^1 - f_1(\mathbf{x}; \boldsymbol{\theta}^1))^2/2$
2: Uniformly draw a set $\widehat{\mathcal{H}}_t^1 \subset \mathcal{H}_t^1, s.t., |\widehat{\mathcal{H}}_t^1| = b$
3: $\widehat{\boldsymbol{\theta}}_t^1 = \widehat{\boldsymbol{\theta}}_{t-1}^1 - \frac{\eta_1}{b} \sum_{(\mathbf{x}, r^1) \in \widehat{\mathcal{H}}_t^1} \nabla_{\boldsymbol{\theta}^1} \mathcal{L}_1[(\mathbf{x}, r^1); \widehat{\boldsymbol{\theta}}_{t-1}^1]$
4: Define $\mathcal{L}_2[(\phi(\mathbf{x}), r^2); \boldsymbol{\theta}^2] = (r^2 - f_2(\phi(\mathbf{x}); \boldsymbol{\theta}^2))^2/2$
5: Uniformly draw a set $\widehat{\mathcal{H}}_t^2 \subset \mathcal{H}_t^2, s.t., |\widehat{\mathcal{H}}_t^2| = b$
6: $\widehat{\boldsymbol{\theta}}_t^2 = \widehat{\boldsymbol{\theta}}_{t-1}^2 - \frac{\eta_2}{b} \sum_{(\phi(\mathbf{x}), r^1) \in \widehat{\mathcal{H}}_t^2} \nabla_{\boldsymbol{\theta}^2} \mathcal{L}_2[(\phi(\mathbf{x}), r^2); \widehat{\boldsymbol{\theta}}_{t-1}^2]$
7: **Return** $(\widehat{\boldsymbol{\theta}}_t^1, \widehat{\boldsymbol{\theta}}_t^2)$

---

# 8 Proof of Lemma 4.1

*Proof.* For any $t \in [T] \wedge (\mathbf{I}_t = 1)$, the regret of one round can be bounded as:

$$
\begin{aligned}
&R_t | (\mathbf{I}_t = 1) \\
&= \mathop{\mathbb{E}}_{y_t \sim \mathcal{D}_{\mathcal{Y}|\mathbf{x}_t}} [\mathcal{L}(\mathbf{y}_{t,\widehat{i}}, \mathbf{y}_{t,y_t}) | \mathbf{x}_t] - \mathop{\mathbb{E}}_{y_t \sim \mathcal{D}_{\mathcal{Y}|\mathbf{x}_t}} [\mathcal{L}(\mathbf{y}_{t,i^*}, \mathbf{y}_{t,y_t}) | \mathbf{x}_t] \\
&= 1 - \mathop{\mathbb{E}}_{y_t \sim \mathcal{D}_{\mathcal{Y}|\mathbf{x}_t}} [\mathcal{L}(\mathbf{y}_{t,i^*}, \mathbf{y}_{t,y_t}) | \mathbf{x}_t] - (1 - \mathop{\mathbb{E}}_{y_t \sim \mathcal{D}_{\mathcal{Y}|\mathbf{x}_t}} [\mathcal{L}(\mathbf{y}_{t,\widehat{i}}, \mathbf{y}_{t,y_t}) | \mathbf{x}_t]) \\
&= \mathop{\mathbb{E}}_{y_t \sim \mathcal{D}_{\mathcal{Y}|\mathbf{x}_t}} [r_{t,i^*}^1 - r_{t,\widehat{i}}^1] \\
&= \mathop{\mathbb{E}}_{y_t \sim \mathcal{D}_{\mathcal{Y}|\mathbf{x}_t}} [\min\{r_{t,i^*}^1 - r_{t,\widehat{i}}^1, 1\}] \\
&= \mathop{\mathbb{E}}_{y_t \sim \mathcal{D}_{\mathcal{Y}|\mathbf{x}_t}} [\min\{r_{t,i^*}^1 - f(\mathbf{x}_{t,\widehat{i}}; \boldsymbol{\theta}_{t-1}) + f(\mathbf{x}_{t,\widehat{i}}; \boldsymbol{\theta}_{t-1}) - r_{t,\widehat{i}}^1, 1\}] \\
&\overset{E_1}{\leq} \mathop{\mathbb{E}}_{y_t \sim \mathcal{D}_{\mathcal{Y}|\mathbf{x}_t}} [\min\{r_{t,i^*}^1 - f(\mathbf{x}_{t,i^*}; \boldsymbol{\theta}_{t-1}) + f(\mathbf{x}_{t,\widehat{i}}; \boldsymbol{\theta}_{t-1}) - r_{t,\widehat{i}}^1, 1\}] \\
&= \mathop{\mathbb{E}}_{y_t \sim \mathcal{D}_{\mathcal{Y}|\mathbf{x}_t}} [\min\{r_{t,i^*}^1 - f(\mathbf{x}_{t,i^*}; \boldsymbol{\theta}_{t-1}) + f(\mathbf{x}_{t,\widehat{i}}; \boldsymbol{\theta}_{t-1}) - r_{t,\widehat{i}}^1, 1\}] \\
&= \mathop{\mathbb{E}}_{y_t \sim \mathcal{D}_{\mathcal{Y}|\mathbf{x}_t}} [\min\{r_{t,i^*}^1 - f(\mathbf{x}_{t,i^*}; \boldsymbol{\theta}_{t-1}^*) + f(\mathbf{x}_{t,i^*}; \boldsymbol{\theta}_{t-1}^*) - f(\mathbf{x}_{t,i^*}; \boldsymbol{\theta}_{t-1}) + f(\mathbf{x}_{t,\widehat{i}}; \boldsymbol{\theta}_{t-1}) - r_{t,\widehat{i}}^1, 1\}] \\
&\leq \mathop{\mathbb{E}}_{y_t \sim \mathcal{D}_{\mathcal{Y}|\mathbf{x}_t}} [\min\{r_{t,i^*}^1 - f(\mathbf{x}_{t,i^*}; \boldsymbol{\theta}_{t-1}^*), 1\}] + f(\mathbf{x}_{t,i^*}; \boldsymbol{\theta}_{t-1}^*) - f(\mathbf{x}_{t,i^*}; \boldsymbol{\theta}_{t-1}) \\
&\quad + \mathop{\mathbb{E}}_{y_t \sim \mathcal{D}_{\mathcal{Y}|\mathbf{x}_t}} [\min\{f(\mathbf{x}_{t,\widehat{i}}; \boldsymbol{\theta}_{t-1}) - r_{t,\widehat{i}}^1, 1\}] \\
&\leq \mathop{\mathbb{E}}_{y_t \sim \mathcal{D}_{\mathcal{Y}|\mathbf{x}_t}} [\min\{|r_{t,i^*}^1 - f(\mathbf{x}_{t,i^*}; \boldsymbol{\theta}_{t-1}^*)|, 1\}] + |f(\mathbf{x}_{t,i^*}; \boldsymbol{\theta}_{t-1}^*) - f(\mathbf{x}_{t,i^*}; \boldsymbol{\theta}_{t-1})| \\
&\quad + \mathop{\mathbb{E}}_{y_t \sim \mathcal{D}_{\mathcal{Y}|\mathbf{x}_t}} [\min\{|f(\mathbf{x}_{t,\widehat{i}}; \boldsymbol{\theta}_{t-1}) - r_{t,\widehat{i}}^1|, 1\}]
\end{aligned}
$$

(8.1)

where $E_1$ is because of $f(\mathbf{x}_{t,i^*}; \boldsymbol{\theta}_{t-1}) \leq f(\mathbf{x}_{t,\widehat{i}}; \boldsymbol{\theta}_{t-1})$. For any $t \in [T] \wedge (\mathbf{I}_t = 0)$, we have $R_t | (\mathbf{I}_t = 0) = \underset{\mathbf{x}_t, y_t}{\mathbb{E}} [L(\mathbf{y}_{t,\widehat{i}}, \mathbf{y}_{t,y_t}) - L(\mathbf{y}_{t,i^*}, \mathbf{y}_{t,y_t})] = 0$ based on Lemma 7.5. Therefore, we have

$$
\begin{aligned}
\mathbf{R}_T &= \sum_{t=1}^{T} R_t \\
&\leq \underbrace{\sum_{t=1}^{T} \underset{y_t \sim \mathcal{D}_{\mathcal{Y}|\mathbf{x}_t}}{\mathbb{E}} [\min\{|r_{t,i^*}^1 - f(\mathbf{x}_{t,i^*}; \boldsymbol{\theta}_{t-1}^*)|, 1\}]}_{I_1} + \underbrace{\sum_{t=1}^{T} |f(\mathbf{x}_{t,i^*}; \boldsymbol{\theta}_{t-1}^*) - f(\mathbf{x}_{t,i^*}; \boldsymbol{\theta}_{t-1})|}_{I_2} \\
&\quad + \underbrace{\sum_{t=1}^{T} \underset{y_t \sim \mathcal{D}_{\mathcal{Y}|\mathbf{x}_t}}{\mathbb{E}} [\min\{|f(\mathbf{x}_{t,\widehat{i}}; \boldsymbol{\theta}_{t-1}) - r_{t,\widehat{i}}^1|, 1\}]}_{I_3} \\
&\leq 2\left(2\sqrt{t\mu} + \mathcal{O}\left(\frac{3}{\sqrt{2}} L\nu\sqrt{T}\right) + \sqrt{2T\log(\mathcal{O}(T)/\delta)}\right) + 2T\xi_1 \\
&\overset{E_2}{\leq} \mathcal{O}(1) + \mathcal{O}\left(\frac{6L\nu + 4\sqrt{\mu}}{\sqrt{2}}\right)\sqrt{T} + 2\sqrt{2T\log(\mathcal{O}(T)/\delta)}
\end{aligned}
$$

(8.2)

where $I_1$ is because of Lemma 8.2, $I_3$ is due to Lemma 8.1, $I_2$ is the application of Lemma 7.14 and $E_2$ is the result of choice of $m$. $\qquad\square$

**Lemma 8.1.** *For any $\delta \in (0,1), \nu > 0, \gamma \geq 1$, suppose $m$ satisfies the conditions in Lemma 4.1. In round $t \in [T]$, given $(\mathbf{x}_t, y_t) \sim \mathcal{D}$, let*

$$
\widehat{i} = \arg\max_{i \in [k]} \left(f_1(\mathbf{x}_{t,\widehat{i}}; \boldsymbol{\theta}_{t-1}^1) + f_2(\phi(\mathbf{x}_{t,\widehat{i}}); \boldsymbol{\theta}_{t-1}^2)\right).
$$

*Then, with probability at least $1 - \delta$, we have*

$$
\begin{aligned}
\frac{1}{t} \sum_{\tau=1}^{t} \underset{y_\tau \sim \mathcal{D}_{\mathcal{Y}|\mathbf{x}_\tau}}{\mathbb{E}} &\left[\min\left\{\left|f_1(\mathbf{x}_{\tau,\widehat{i}}; \boldsymbol{\theta}_{\tau-1}^1) + f_2(\phi(\mathbf{x}_{\tau,\widehat{i}}); \boldsymbol{\theta}_{\tau-1}^2) - r_{\tau,\widehat{i}}^1\right|, 1\right\}\right] \\
&\leq \sqrt{\frac{2\mu}{t}} + \mathcal{O}\left(\frac{3L\nu}{\sqrt{2t}}\right) + \sqrt{\frac{2\log(\mathcal{O}(1)/\delta)}{t}}.
\end{aligned}
$$

(8.3)

*Proof.* For any $\tau \in [t]$, define

$$
\begin{aligned}
V_\tau = \underset{y_\tau \sim \mathcal{D}_{\mathcal{Y}|\mathbf{x}_\tau}}{\mathbb{E}} &\left[\min\{|f_1(\mathbf{x}_{\tau,\widehat{i}}; \boldsymbol{\theta}_{\tau-1}^1) + f_2(\mathbf{x}_{\tau,\widehat{i}}; \boldsymbol{\theta}_{\tau-1}^2) - r_{\tau,\widehat{i}}^1|, 1\}\right] \\
&- \min\{|f_1(\mathbf{x}_{\tau,\widehat{i}}; \boldsymbol{\theta}_{\tau-1}^1) + f_2(\mathbf{x}_{\tau,\widehat{i}}; \boldsymbol{\theta}_{\tau-1}^2) - r_{\tau,\widehat{i}}^1|, 1\}
\end{aligned}
$$

(8.4)

Then, we have

$$
\begin{aligned}
\mathbb{E}[V_\tau | F_{\tau-1}] = \underset{y_\tau \sim \mathcal{D}_{\mathcal{Y}|\mathbf{x}_\tau}}{\mathbb{E}} &\left[\min\{|f_1(\mathbf{x}_{\tau,\widehat{i}}; \boldsymbol{\theta}_{\tau-1}^1) + f_2(\mathbf{x}_{\tau,\widehat{i}}; \boldsymbol{\theta}_{\tau-1}^2) - r_{\tau,\widehat{i}}^1|, 1\}\right] \\
&- \underset{y_\tau \sim \mathcal{D}_{\mathcal{Y}|\mathbf{x}_\tau}}{\mathbb{E}} \left[\min\{|f_1(\mathbf{x}_{\tau,\widehat{i}}; \boldsymbol{\theta}_{\tau-1}^1) + f_2(\mathbf{x}_{\tau,\widehat{i}}; \boldsymbol{\theta}_{\tau-1}^2) - r_{\tau,\widehat{i}}^1|, 1\}\right] \\
&= 0
\end{aligned}
$$

(8.5)

where $F_{\tau-1}$ denotes the $\sigma$-algebra generated by the history $\mathcal{H}_{\tau-1}$. Therefore, $\{V_\tau\}_{\tau=1}^{t}$ are the martingale difference sequence.

Applying the Hoeffding-Azuma inequality, with probability at least $1 - \delta$, we have

$$
\mathbb{P}\left[\frac{1}{t} \sum_{\tau=1}^{t} V_\tau - \underbrace{\frac{1}{t} \sum_{\tau=1}^{t} \underset{y_\tau \sim \mathcal{D}_{\mathcal{Y}|\mathbf{x}_\tau}}{\mathbb{E}} [V_\tau | \mathbf{F}_\tau]}_{I_1} > \sqrt{\frac{2\log(1/\delta)}{t}}\right] \leq \delta
$$

(8.6)

As $I_1$ is equal to 0, we have

$$\frac{1}{t}\sum_{\tau=1}^{t}\mathop{\mathbb{E}}_{y_\tau \sim \mathcal{D}_{\mathcal{Y}|\mathbf{x}_\tau}}\left[\min\{\left|f_1(\mathbf{x}_{\tau,\widehat{i}};\boldsymbol{\theta}^1_{\tau-1}) + f_2(\mathbf{x}_{\tau,\widehat{i}};\boldsymbol{\theta}^2_{\tau-1}) - r^1_{\tau,\widehat{i}}\right|,1\}\right]$$

$$\leq \underbrace{\frac{1}{t}\sum_{\tau=1}^{t}\min\{\left|f_2\left(\mathbf{x}_{\tau,\widehat{i}};\boldsymbol{\theta}^2_{\tau-1}\right) - \left(r_{\tau,\widehat{i}} - f_1(\mathbf{x}_{\tau,\widehat{i}};\boldsymbol{\theta}^1_{\tau-1})\right)\right|,1\}}_{I_3} + \sqrt{\frac{2\log(1/\delta)}{t}}. \quad (8.7)$$

For $I_3$, based on Lemma 7.8, for any $\widetilde{\boldsymbol{\theta}}^2$ satisfying $\|\widetilde{\boldsymbol{\theta}}^2 - \boldsymbol{\theta}^2_0\|_2 \leq \mathcal{O}(\frac{\nu}{\sqrt{m}})$, with probability at least $1-\delta$, we have

$$I_3 \leq \frac{1}{t}\sum_{\tau=1}^{t}\min\{|f_1(\mathbf{x}_{\tau,\widehat{i}};\boldsymbol{\theta}^1_{\tau-1}) + f_2\left(\mathbf{x}_{\tau,\widehat{i}};\widetilde{\boldsymbol{\theta}}^2\right) - r^1_{\tau,\widehat{i}}|,1\} + \mathcal{O}\left(\frac{3L\nu}{\sqrt{2t}}\right)$$

$$\leq \frac{1}{t}\sqrt{t}\sqrt{\underbrace{\sum_{\tau=1}^{t}\left(f_1(\mathbf{x}_{\tau,\widehat{i}};\boldsymbol{\theta}^1_{\tau-1}) + f_2\left(\mathbf{x}_{\tau,\widehat{i}};\widetilde{\boldsymbol{\theta}}^2\right) - r^1_{\tau,\widehat{i}}\right)^2}_{I_4}} + \mathcal{O}\left(\frac{3L\nu}{\sqrt{2t}}\right) + \sqrt{\frac{2\log(1/\delta)}{t}} \quad (8.8)$$

$$\leq \sqrt{\frac{2\mu}{t}} + \mathcal{O}\left(\frac{3L\nu}{\sqrt{2t}}\right) + \sqrt{\frac{2\log(1/\delta)}{t}}.$$

where $I_4$ is by the definition of $\mu$.

Combining above inequalities together, with probability at least $1-\delta$, we have

$$\frac{1}{t}\sum_{\tau=1}^{t}\mathop{\mathbb{E}}_{y_\tau \sim \mathcal{D}_{\mathcal{Y}|\mathbf{x}_\tau}}\left[\min\left\{\left|f_1(\mathbf{x}_{\tau,\widehat{i}};\boldsymbol{\theta}^1_{\tau-1}) + f_2(\phi(\mathbf{x}_{\tau,\widehat{i}});\boldsymbol{\theta}^2_{\tau-1}) - r^1_{\tau,\widehat{i}}\right|,1\right\}\right]$$

$$\leq \sqrt{\frac{2\mu}{t}} + \mathcal{O}\left(\frac{3L\nu}{\sqrt{2t}}\right) + 2\sqrt{\frac{2\log(\mathcal{O}(1)/\delta)}{t}}, \quad (8.9)$$

where we applied union bound over $\delta$ to make above events occur concurrently. $\qquad\square$

**Lemma 8.2.** *For any $\delta \in (0,1), \nu > 0, \gamma \geq 1$, suppose $m$ satisfies the conditions in Lemma 4.1. In round $t \in [T]$, given $(\mathbf{x}_t, y_t) \sim \mathcal{D}$, let $i^* = \arg\max_{i\in[k]} h(\mathbf{x}_{t,i})$. Then, with probability at least $1-\delta$, there exists $\boldsymbol{\theta}^{1,*}_{t-1}, \boldsymbol{\theta}^{2,*}_{t-1}$, such that*

$$\frac{1}{t}\sum_{\tau=1}^{t}\mathop{\mathbb{E}}_{y_\tau \sim \mathcal{D}_{\mathcal{Y}|\mathbf{x}_\tau}}\left[\min\left\{\left|f_1(\mathbf{x}_{\tau,i^*};\boldsymbol{\theta}^1_{\tau-1}) + f_2(\phi(\mathbf{x}_{\tau,i^*});\boldsymbol{\theta}^2_{\tau-1}) - r^1_{\tau,i^*}\right|,1\right\}\right]$$

$$\leq \sqrt{\frac{2\mu}{t}} + \mathcal{O}\left(\frac{3L\nu}{\sqrt{2t}}\right) + 2\sqrt{\frac{2\log(\mathcal{O}(1)/\delta)}{t}}, \quad (8.10)$$

*where $\mathcal{H}_{t-1} = \{\mathbf{x}_{\tau,i^*}, r^1_{\tau,i^*}\}_{\tau=1}^{t-1}$ is historical data.*

*Proof.* For any $\tau \in [t]$, define

$$V_\tau = \mathop{\mathbb{E}}_{y_\tau \sim \mathcal{D}_{\mathcal{Y}|\mathbf{x}_\tau}}\left[\min\{|f_1(\mathbf{x}_{\tau,i^*};\boldsymbol{\theta}^1_{\tau-1}) + f_2(\mathbf{x}_{\tau,i^*};\boldsymbol{\theta}^2_{\tau-1}) - r^1_{\tau,i^*})|,1\}\right]$$

$$- \min\{|f_1(\mathbf{x}_{\tau,i^*};\boldsymbol{\theta}^1_{\tau-1}) + f_2(\mathbf{x}_{\tau,i^*};\boldsymbol{\theta}^2_{\tau-1}) - r^1_{\tau,i^*}|,1\} \quad (8.11)$$

Then, we have

$$\mathbb{E}[V_\tau|F_{\tau-1}] = \mathop{\mathbb{E}}_{y_\tau \sim \mathcal{D}_{\mathcal{Y}|\mathbf{x}_\tau}}\left[\min\{|f_1(\mathbf{x}_{\tau,i^*};\boldsymbol{\theta}^1_{\tau-1}) + f_2(\mathbf{x}_{\tau,i^*};\boldsymbol{\theta}^2_{\tau-1}) - r^1_{\tau,i^*})|,1\}\right]$$

$$- \mathop{\mathbb{E}}_{y_\tau \sim \mathcal{D}_{\mathcal{Y}|\mathbf{x}_\tau}}\left[\min\{|f_1(\mathbf{x}_{\tau,i^*};\boldsymbol{\theta}^1_{\tau-1}) + f_2(\mathbf{x}_{\tau,i^*};\boldsymbol{\theta}^2_{\tau-1}) - r^1_{\tau,i^*}|,1\}\right] \quad (8.12)$$

$$= 0$$

where $F_{\tau-1}$ denotes the $\sigma$-algebra generated by the history $\mathcal{H}_{\tau-1}$. Therefore, $\{V_\tau\}_{\tau=1}^t$ are the martingale difference sequence.

Applying the Hoeffding-Azuma inequality, with probability at least $1 - \delta$, we have

$$\mathbb{P}\left[\frac{1}{t}\sum_{\tau=1}^t V_\tau - \underbrace{\frac{1}{t}\sum_{\tau=1}^t \mathop{\mathbb{E}}_{y_\tau \sim \mathcal{D}_{\mathcal{Y}|\mathbf{x}_\tau}}[V_\tau|\mathbf{F}_\tau]}_{I_1} > \sqrt{\frac{2\log(1/\delta)}{t}}\right] \le \delta \tag{8.13}$$

As $I_1$ is equal to 0, we have

$$\frac{1}{t}\sum_{\tau=1}^t \mathop{\mathbb{E}}_{y_\tau \sim \mathcal{D}_{\mathcal{Y}|\mathbf{x}_\tau}}\left[\min\{\left|f_1(\mathbf{x}_{\tau,i^*};\boldsymbol{\theta}_{\tau-1}^1) + f_2(\mathbf{x}_{\tau,i^*};\boldsymbol{\theta}_{\tau-1}^2) - r_{\tau,i^*}^1)\right|, 1\}\right]$$
$$\le \underbrace{\frac{1}{t}\sum_{\tau=1}^t \min\{\left|f_2\left(\mathbf{x}_{\tau,i^*};\boldsymbol{\theta}_{\tau-1}^2\right) - \left(r_{\tau,i^*} - f_1(\mathbf{x}_{\tau,i^*};\boldsymbol{\theta}_{\tau-1}^1)\right)\right|, 1\}}_{I_3} + \sqrt{\frac{2\log(1/\delta)}{t}}. \tag{8.14}$$

For $I_3$, based on Lemma 7.8, for any $\widetilde{\boldsymbol{\theta}}^2$ satisfying $\|\widetilde{\boldsymbol{\theta}}^2 - \boldsymbol{\theta}_0^2\|_2 \le \mathcal{O}(\frac{\nu}{\sqrt{m}})$, with probability at least $1 - 3\delta$, we have

$$I_3 \le \frac{1}{t}\sum_{\tau=1}^t \min\{|f_1(\mathbf{x}_{\tau,i^*};\boldsymbol{\theta}_{\tau-1}^1) + f_2\left(\mathbf{x}_{\tau,i^*};\widetilde{\boldsymbol{\theta}}^2\right) - r_{\tau,i^*}^1|, 1\} + \mathcal{O}\left(\frac{3L\nu}{\sqrt{2t}}\right)$$
$$\le \frac{1}{t}\sqrt{t}\sqrt{\underbrace{\sum_{\tau=1}^t \left(f_1(\mathbf{x}_{\tau,i^*};\boldsymbol{\theta}_{\tau-1}^1) + f_2\left(\mathbf{x}_{\tau,i^*};\widetilde{\boldsymbol{\theta}}^2\right) - r_{\tau,i^*}^1\right)^2}_{I_4}} + \mathcal{O}\left(\frac{3L\nu}{\sqrt{2t}}\right) + \sqrt{\frac{2\log(1/\delta)}{t}}$$
$$\le \sqrt{\frac{2\mu}{t}} + \mathcal{O}\left(\frac{3L\nu}{\sqrt{2t}}\right) + \sqrt{\frac{2\log(1/\delta)}{t}}. \tag{8.15}$$

where $I_4$ is by the assumption of $\mu$.

Combining above inequalities together, with probability at least $1 - \delta$, we have

$$\frac{1}{t}\sum_{\tau=1}^t \mathop{\mathbb{E}}_{y_\tau \sim \mathcal{D}_{\mathcal{Y}|\mathbf{x}_\tau}}\left[\min\left\{\left|f_1(\mathbf{x}_{\tau,i^*};\boldsymbol{\theta}_{\tau-1}^1) + f_2(\phi(\mathbf{x}_{\tau,i^*});\boldsymbol{\theta}_{\tau-1}^2) - r_{\tau,i^*}^1\right|, 1\right\}\right]$$
$$\le \sqrt{\frac{2\mu}{t}} + \mathcal{O}\left(\frac{3L\nu}{\sqrt{2t}}\right) + 2\sqrt{\frac{2\log(\mathcal{O}(1)/\delta)}{t}}, \tag{8.16}$$

where we applied union bound over $\delta$ to make above events occur concurrently. □

# 9 Bounds for Effective Dimension $\widetilde{d}$

Let $\{\mathbf{x}_{t,\widehat{i}}\}_{t=1}^T$ be the selected contexts in $T$ rounds, then we have the following definition of NTK.

**Definition 9.1** ( NTK [27, 3])**.** *Let $\mathcal{N}$ denote the normal distribution. Define*

$$\mathbf{H}_{i,j}^0 = \boldsymbol{\Sigma}_{i,j}^0 = \langle \mathbf{x}_i, \mathbf{x}_j \rangle, \ \ \mathbf{N}_{i,j}^l = \begin{pmatrix} \boldsymbol{\Sigma}_{i,i}^l & \boldsymbol{\Sigma}_{i,j}^l \\ \boldsymbol{\Sigma}_{j,i}^l & \boldsymbol{\Sigma}_{j,j}^l \end{pmatrix}$$

$$\boldsymbol{\Sigma}_{i,j}^l = 2\mathbb{E}_{a,b\sim\mathcal{N}(\mathbf{0},\mathbf{N}_{i,j}^{l-1})}[\sigma(a)\sigma(b)]$$

$$\mathbf{H}_{i,j}^l = 2\mathbf{H}_{i,j}^{l-1}\mathbb{E}_{a,b\sim\mathcal{N}(\mathbf{0},\mathbf{N}_{i,j}^{l-1})}[\sigma'(a)\sigma'(b)] + \boldsymbol{\Sigma}_{i,j}^l.$$

*Then, over the contexts $\{\mathbf{x}_{t,\widehat{i}}\}_{t=1}^T$, the Neural Tangent Kernel (NTK) is defined as $\mathbf{H} = (\mathbf{H}^L + \boldsymbol{\Sigma}^L)/2$.*

Then, we define the following gram matrix $\mathbf{G}$. Let $g(x; \boldsymbol{\theta}_0) = \nabla_{\boldsymbol{\theta}} f(x; \boldsymbol{\theta}_0) \in \mathbb{R}^p$ and $G = [g(\mathbf{x}_{1,\widehat{i}}; \boldsymbol{\theta}_0)/\sqrt{m}, \ldots, g(\mathbf{x}_{T,\widehat{i}}; \boldsymbol{\theta}_0)/\sqrt{m}] \in \mathbb{R}^{p \times T}$ where $p = m + mkd + m^2(L-1)$. Therefore, we have $\mathbf{G} = G^\top G$. Based on Theorem 3.1 in [3], when $m \geq \mathcal{O}(T^4 k^6 \log(2Tk/\delta)/\lambda_0^4)$ where $\lambda_0$ is the smallest eigenvalue of $\mathbf{H}$, with probability at least $1 - \delta$, we have

$$\|\mathbf{G} - \mathbf{H}\| \leq \frac{\lambda_0}{2}. \tag{9.1}$$

Then, we have the following bound:

$$
\begin{aligned}
\log \det(\mathbf{I} + \mathbf{H}) &= \log \det\left(\mathbf{I} + \mathbf{G} + (\mathbf{H} - \mathbf{G})\right) \\
&\leq \log \det(\mathbf{I} + \mathbf{G}) + \langle (\mathbf{I} + \mathbf{G})^{-1}, (\mathbf{H} - \mathbf{G}) \rangle \\
&\leq \log \det(\mathbf{I} + \mathbf{G}) + \|(\mathbf{I} + \mathbf{G})^{-1}\|_F \|\mathbf{H} - \mathbf{G}\|_F \\
&\leq \log \det(\mathbf{I} + \mathbf{G}) + \sqrt{T} \|\mathbf{H} - \mathbf{G}\|_F \\
&\leq \log \det(\mathbf{I} + \mathbf{G}) + 1
\end{aligned}
\tag{9.2}
$$

where the first inequality is because of the concavity of $\log \det(\cdot)$ and the third inequality is by Lemma B.1 in [55] with the choice of $m$. Then, the effective dimension $\widetilde{d}$ can be bounded by:

$$
\begin{aligned}
\widetilde{d} &= \frac{\log \det(\mathbf{I} + \mathbf{H})}{\log(1 + T)} \\
&\leq \frac{\log \det(\mathbf{I} + \mathbf{G}) + 1}{\log(1 + T)} \\
&\stackrel{E_1}{=} \frac{\log \det(\mathbf{I} + GG^\top) + 1}{\log(1 + T)} \\
&\stackrel{E_2}{\leq} p \cdot \frac{\log \|\mathbf{I} + GG^\top\|_2}{\log(1 + T)} + \frac{1}{\log(1 + T)} \\
&\stackrel{E_3}{\leq} p + \frac{1}{\log(1 + T)}
\end{aligned}
\tag{9.3}
$$

where $E_1$ is because of $\det(\mathbf{I} + G^\top G) = \det(\mathbf{I} + GG^\top)$ and $E_2$ is due to $\det(GG^\top) = \|GG^\top\|_2^p$ ($GG^\top \in \mathbb{R}^{p \times p}$) and $E_3$ is according to

$$\|\mathbf{I} + GG^\top\|_2 \leq 1 + \|GG^\top\|_2 \leq 1 + \sum_{t=1}^{T} \|g(\mathbf{x}_{t,\widehat{i}}; \boldsymbol{\theta}_0) g(\mathbf{x}_{t,\widehat{i}}; \boldsymbol{\theta}_0)^\top / m\|_2 \leq 1 + T,$$

where the last inequality is as the result of $\|g(\mathbf{x}_{t,\widehat{i}}; \boldsymbol{\theta}_0)/\sqrt{m}\|_2 \leq 1$ (Lemma B.3 in [16]). Therefore, we have

$$\widetilde{d} \leq p + \frac{1}{\log(1 + T)} \quad \text{and} \quad p = m + mkd + m^2(L-1). \tag{9.4}$$

## 10 Further Details in Experiments

In this section, we report the specific configurations in the experiments, the sensitivity study of the core hyperparameter $\gamma$ for I-NeurAL, and the ablation study for label budget. Table 2 exhibits the details of using datasets.

| Dataset | Features | Samples | Classes |
|---------|----------|---------|---------|
| Phishing | 68 | 11,055 | 2 |
| IJCNN | 22 | 12,000 | 2 |
| Letter | 784 | 12,000 | 26 |
| Fashion | 784 | 12,000 | 10 |
| MNIST | 784 | 12,000 | 10 |
| CIFAR-10 | 3,072 | 12,000 | 10 |

Table 2: Statistics of the datasets used in our experiments. We conduct experiments on binary classification tasks. For Letter, the binary task is to separate 'A-M' versus 'N-Z'. For Fashion, the binary task is to separate 'T-shirt' versus 'Trouser' images. For MNIST, the binary task is to separate odd and even digits. For CIFAR-10, the binary task is to separate 'horse' and 'ship' images.

**Implementation Details**. We use PyTorch as our backend, and all experiments were conducted on a server with NVIDIA Tesla V100 SXM2 GPU. The classification model in all methods is the same 2-layer fully-connected network with 100-width for the fair comparison. We use Adam optimizer to train the classification model with the fixed learning rate is $0.001$, and the batch size is $64$, since these are model-agnostic hyperparameters. As NeuAL-NTK-F and NeuAL-NTK-D only work on the binary classification problem, we transformed the $k$-class classification problem into the binary classification problem when $k > 2$. In detail, given $k$ class, we regard the $k/2$ classes as one class and remaining classes as another class. For Random algorithm, the query probability $p$ is set as $0.1$. To find the best performance of each method, we conduct the grid search over all hyperparameters. In Margin algorithm, the query threshold is searched over $\{0.3, 0.5, 0.7, 0.9, 0.95\}$ for all datasets. For NeuAL-NTK-F and NeuAL-NTK-D, there is also an exploration parameter $\gamma$ to determine the query aggressiveness and we conduct the grid search over $\{0.1, 0.3, 0.5, 0.8, 1.0\}$ for it. For ALPS, following the method in [22], we form the hypothesis class by generating 20 hypotheses on the $3\%$ of total data samples (as same as the query budget) with different random seeds, and we conduct the grid search $\{0.1, 0.25, 0.5, 0.75, 0.9\}$ over the two slack terms in ALPS. We have tried to generate more hypotheses in the experiments, but the performance of ALPS does not improve accordingly. For I-NeurAL, the only hyperparameter $\gamma$ is searched over $\{1, 2, 5, 6, 7, 10\}$ for all datasets ($c_1$ and $c_2$ in $\boldsymbol{\beta}_t$ is set as 1). The confidence level $\delta$ is set as $0.1$ for all the needed methods. In the end, we report the average results of 5 runs for all methods.

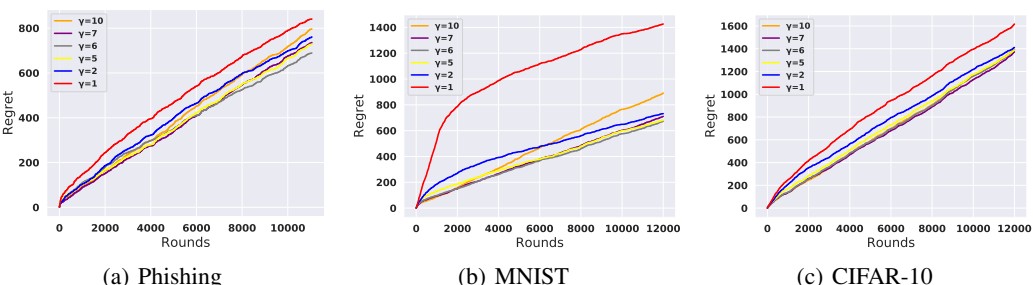

Figure 2: Parameter sensitivity on three datasets.

**Sensitivity study for** $\gamma$. As $\gamma$ is closely related to the query threshold of I-NeurAL, we test the sensitivity of I-NeurAL with regard to $\gamma$. Based on our analysis, it is required that $\gamma \geq 1$. When $\gamma$ is the smallest number (e.g., $\gamma = 1$), I-NeurAL queries the labels only if the difference between the top two classes is very small (i.e., the confidence level is very low). In this manner, I-NeurAL will save more query budget but take more risks on many instances, incurring more regret. This explains why the red line ($\gamma = 1$) is above all other lines. In contrast, if $\gamma$ is a large number, I-NeurAL will be more aggressive in making queries and thus obtain satisfactory performance. However, if $\gamma$ is too large,

I-NeurAL tends to query on these instances even when our model is very confident to the predictions, wasting the query budget that could have been used on these uncertain instances. Therefore, we expect that $\gamma$ is neither too small nor too large. The experiments verify our assumption, when $\gamma$ is 6 or 7, I-NeurAL almost achieves the best performance throughout all datasets and configurations.

**Ablation study for label budget**. To examine the final performance of each algorithm, we conduct new experiments with different percentages of label budget: 3%, 10%, 20%, 50%. After $T$ rounds, we evaluate the latest model on the test (unseen) data to calculate the accuracy, which evaluates the population accuracy. For all the datasets, $T$ is set as 10000, except that $T = 2000$ for Phishing because Phishing has fewer data instances. Table 3 - 6 reports the results. To sum up, I-NeurAL still achieves the best accuracy with different label budget. With a small amount of label budget (3%, 10%), I-NeurAL can make smart decisions to query labels on these instances with big uncertainty and leverage the full feedback to exploit the past knowledge, which enable I-NeurAL to outperform all the baselines. With the larger label budget (20%, 50 %), all methods have enough labels to train. Thus, the advantages of I-NeurAL is less significant and the gap between I-NeurAL and baselines is decreasing. Nevertheless, I-NeurAl still has the best performance benefiting from smart query choices.

|  | Phishing | IJCNN | Letter | Fashion | MNIST | CIFAR-10 |
|---|---|---|---|---|---|---|
| Random | 91.75% | 93.80% | 71.60% | 95.70% | 87.90% | 86.40% |
| Margin | 93.46% | 92.95% | 73.55% | 98.15% | 90.25% | 88.25% |
| NeuAL-NTK-F | 54.69% | 75.15% | 48.05% | 51.30% | 51.10% | 71.00% |
| NeuAL-NTK-D | 92.89% | 93.65% | 73.80% | 97.70% | 90.15% | 84.05% |
| ALPS | 91.47% | 93.25% | 71.45% | 95.70% | 86.95% | 85.40% |
| I-NeurAL | **94.22%** | **95.75%** | **77.45%** | **99.15%** | **94.45%** | **89.00%** |

Table 3: Test Accuracy with **3%** budget.

|  | Phishing | IJCNN | Letter | Fashion | MNIST | CIFAR-10 |
|---|---|---|---|---|---|---|
| Random | 93.93% | 96.70% | 79.70% | 97.30% | 90.90% | 89.00% |
| Margin | 94.98% | 97.10% | 81.50% | 98.60% | 94.40% | 89.45% |
| NeuAL-NTK-F | 54.69% | 87.65% | 48.05% | 51.30% | 51.10% | 70.95% |
| NeuAL-NTK-D | 92.99% | 96.90% | 80.55% | 98.70% | 94.85% | 89.05% |
| ALPS | 92.89% | 96.20% | 78.05% | 97.50% | 93.00% | 89.35% |
| I-NeurAL | **95.64%** | **97.90%** | **83.95%** | **99.30%** | **97.20%** | **91.75%** |

Table 4: Test Accuracy with **10%** label budget.

|  | Phishing | IJCNN | Letter | Fashion | MNIST | CIFAR-10 |
|---|---|---|---|---|---|---|
| Random | 93.93% | 96.70% | 81.90% | 98.15% | 93.00% | 89.50% |
| Margin | 95.17% | 98.15% | 82.45% | 98.90% | 95.05% | 89.75% |
| NeuAL-NTK-F | 54.69% | 89.95% | 48.05% | 51.30% | 51.10% | 72.15% |
| NeuAL-NTK-D | 94.98% | 97.75% | 82.35% | 99.30% | 96.15% | 90.90% |
| ALPS | 94.41% | 97.05% | 83.20% | 98.30% | 94.45% | 88.95% |
| I-NeurAL | **95.64%** | **98.35%** | **84.65%** | **99.30%** | **97.95%** | **91.80%** |

Table 5: Test Accuracy with **20%** label budget.

|              | Phishing | IJCNN  | Letter | Fashion | MNIST  | CIFAR-10 |
|--------------|----------|--------|--------|---------|--------|----------|
| Random       | 94.98%   | 97.75% | 86.05% | 98.95%  | 96.40% | 90.75%   |
| Margin       | 95.73%   | 98.40% | 86.35% | 99.05%  | 96.20% | 91.25%   |
| NeuAL-NTK-F  | 54.69%   | 90.85% | 48.05% | 51.30%  | 51.10% | 72.35%   |
| NeuAL-NTK-D  | 95.83%   | 98.00% | 83.35% | 99.30%  | 97.15% | 90.55%   |
| ALPS         | 94.50%   | 97.80% | 87.10% | 99.05%  | 96.70% | 90.85%   |
| I-NeurAL     | **96.02**% | **98.75**% | 86.05% | **99.35**% | **97.80**% | **92.30**% |

Table 6: Test Accuracy with **50%** label budget.