# OpenReview forum: "Improved Algorithms for Neural Active Learning"
_NeurIPS.cc/2022/Conference — NeurIPS 2022 Accept_

### Official Review · Reviewer_aMrL · 2022-07-03

**Rating:** 7
**Confidence:** 2
**Soundness:** 3 good
**Presentation:** 4 excellent
**Contribution:** 3 good

**Summary:**

The paper addresses active learning, one of the field's essential problems.
This work extends EE-Net for active learning and provides its regret analysis.
Compared with the closest work, the authors show a better regret bound.
Experimental results also show the proposed algorithm outperforms previous work.


**Questions:**

Questions
- Is there a difference between Derivative-Context (DC) Embedding (Definition 3.1) and the input of $f_2$ in EE-Net?
- Although $\beta_{t}$ is decreased as $\sqrt(t)$ in Algorithm 1, how is $beta_t$ controlled in the experiments?
- Is the grid search over all hyperparameters conducted for the same datasets in which the regret evaluations are conducted?
- Although 3% of the total number of instances in the data set had been used as a budget in the experiments, do you see the same trend with other budgets?

Minor Comments
- In equation (7.28), the end parentheses of f are missing.
- typo: lesat -> least in the following sentence of equation (7.28).
- In equation (7.38), (7.39), (8.12), (8.13), (8.14), (8.15), $x_t$ and $y_t$, the subscripts of the expectations, seem to be $x_{\tau}$ and $y_{\tau}$, respectively.
- In equation (7.78), a \max seems to be \min.
- In Lemma 7.13, $\theta_t^{(k)}$ is not used in the main text.
- In Lemma 7.13, one of the equation number is missing.
- In the 9th statement of equation (8.2), $f(x_{t,i^*};\theta_{t-1}^*)f(x_{t,i^*};\theta_{t-1})$ should be $f(x_{t,i^*};\theta_{t-1}^*)-f(x_{t,i^*};\theta_{t-1}^*)$.
- In Lemma 8.2, $\cal{H}_{t-1}$ is not used in the main text.


**Limitations:**

Since the paper shows theoretically strong results, I would like to know the practical consideration for implementing the proposed method.  For example, I think it is not provided how the budget size affects the performance of the active learning method.

**Strengths And Weaknesses:**

Strengths
- The regret analysis shows the regret bound improving by O(log T) and removing the curse of dimensionality and the complexity of the function.
- This work provides theoretical justification for the proposed algorithm.
- The empirical results also prove the advantage of the proposed method.

Weaknesses
- The proposed algorithm and the result of the regret analysis are a straightforward extension of EE-Net, so their novelty is a little limited.

---

> ### Author Response · Authors · 2022-08-02
> **Response to Reviewer aMrL's Questions**
>
> Thanks for the reviewer's constructive and elaborate comments. We really appreciate the pointed typos and we've modified them accordingly.
>
>
> **Q1**:
> Is there a difference between Derivative-Context (DC) Embedding (Definition 3.1) and the input of $f_2$ in EE-Net
>
>
> **A1**:
> Yes, there is a critical difference between Definition 3.1 and the input of $f_2$ in EE-Net [2].
> Please take a look at the comments ``Comparison with EE-Net (1) (2)" at the top, where we elaborate the difference of Definition 3.1 from EE-Net and the novelties compared to EE-Net.
>
>
>
>
>
> **Q2**: How is $\beta_t$ controlled in the experiments?
>
> **A2**:
> $\beta_t$ is set as the default value as stated in paper where $c_1$ and $c_2$ is set as 1.  We added the hyperparameter $\gamma$ multiplied by $\beta_t$ to control the aggressiveness of I-NeurAL,  and conducted the grid search for $\gamma$ in the experiments to find the best performance.
>
>
>
> **Q3**:
> Is the grid search over all hyperparameters conducted for the same datasets in which the regret evaluations are conducted?
>
> **A3**:
> Yes, this is a popular approach to find the best performance of baselines and proposed algorithm, following [3].
> In practice, setting $\gamma = 5$ for I-NeurAL can achieve good and general performance.
>
>
>
> **Q4**: Do you see the same trend with different label budgets?
>
> **A4**: Yes, we added new experiments with varying label budgets (3\%, 10\%, 20\%, 50\% ) on all datasets,  and we reported testing accuracy for all baselines and datasets. The results are reported in Appendix 10. I-NeurAL achieves the best performance with different label budgets.
> Please take a look at our comment "Ablation Study for Label Budget (1)(2)" to Reviewer d9Zc, where we have discussed the new results.
>
>
>  Reference:
>
> [2] ``EE-NET: Exploitation-Exploration Neural
> Networks in Contextual Bandits".
>  https://arxiv.org/pdf/2110.03177.pdf
>
>
> [3] ``Online Active Learning with Surrogate Loss Functions".
> https://papers.nips.cc/paper/2021/file/c1619d2ad66f7629c12c87fe21d32a58-Paper.pdf

---

> > ### Comment · Reviewer_aMrL · 2022-08-07
> > **Thanks for Response**
> >
> > Thank you for your response.   It addresses my concerns and clarifies my questions.
> > I would like to keep my positive score.

---

> > > ### Author Response · Authors · 2022-08-09
> > > **Thanks for your time**
> > >
> > > Thank you very much again for your time and efforts in reviewing our paper!

---

### Official Review · Reviewer_d9Zc · 2022-07-11

**Rating:** 6
**Confidence:** 3
**Soundness:** 3 good
**Presentation:** 4 excellent
**Contribution:** 3 good

**Summary:**

This paper studies deep active learning under the streaming setting. Unlike prior work that utilizes NTK approximation bounds, this paper proposes its algorithm based on generalization bounds of overparameterized neural networks. The analysis presents several novelties and advantages over SOTA theoretical bounds including the independence from effective dimension, and the elimination of unknown complexity terms in prior work. The paper also proves an interesting theorem that the algorithm achieves zero-regret over bayes-optimal classifier over a fixed amount of rounds independent from T. The authors also conducted experiments over different datasets.


**Questions:**

Overall, I think the paper has strong theoretical guarantees. As a result, I believe the standard of "outperforming every baseline empirically" is not necessary in this case for an accept. Therefore, I strongly encourage the authors to present results using more practical evaluation metrics and under more realistic experiment settings (sufficiently high labeling budget), even though the results may not be as ideal.

I am willing to raise my scores provided some of the above concerns about empirical results are addressed.

**Limitations:**

More or less suffcient.

**Strengths And Weaknesses:**

Strengths:
1. The theoretical analysis is novel with stronger bounds over prior work. I find the theoretical results and the utilization of generalization bounds very interesting.
2. Empirically, the authors conducted experiments over a wide range of datasets and the algorithm is computationally efficient in practice.

Weaknesses:
1. The experiment section measures total regret, which is not a standard metric and is rarely used in practice. Under limited budget and time horizon, the total regret (described in lines 282-283) seem to favor an aggressive exploration strategy, where the model should become accurate as early as possible. I believe presenting either the latest population regret or simply the population loss at T is a more appropriate metric. Moreover, the authors restricted the labeling budget to be 3% of the dataset, which needs more justification. Usually, it takes at least 50% of the labels in datasets like FashionMNIST or CIFAR10 to reach a solid accuracy.
2. The authors in their analysis presents generalization bounds and advertise their work for its guarantee to generalize to unseen data. Therefore, I think test accuracies at time T should also be included.

Minor Suggestion:
Although the authors mentions their utilization of generalization bounds of overparameterized neural networks [10] in remarks 4.2 and 4.3, I think these technics deserves a highlight in the introduction or abstract.

---

> ### Author Response · Authors · 2022-08-02
> **Ablation Study for Label Budget (1)**
>
> Thanks for your great suggestions. We've added the new metric with different budgets accordingly to strengthen our empirical results.
> First, we added the new evaluation metric: population accuracy. Given a fixed budget, after $T$ rounds, we evaluate the latest model on the test (unseen) data to calculate the accuracy, which evaluates the population accuracy.
> For all the datasets, $T$ is set as $10000$, except that $T = 2000$ for Phishing because Phishing has fewer data instances.
> Second, we added new experiments with different label budget: 3\%, 10\%, 20\%, 50\%.  We report the results as follows.
>
>
> ***
>
> **Test Accuracy with 3\% label budget**
>
> |             | Phishing     | IJCNN        | Letter          | Fashion       | MNIST        | CIFAR-10      |
> | --- |     --- |     --- |  --- |  ---  | --- | --- |
> | Random     | 91.75\% |    *93.80\%*      | 71.60\%        | 95.70\%         | 87.90\%         | 86.40\%         |
> | Margin      | 93.46\%         |  92.95\%          |      73.55\%     | *98.15\%*        | *90.25\%*       | *88.25\%*          |
> NeuAL-NTK-F | 54.69\%     | 75.15\%        | 48.05\%         | 51.30\%        | 51.10\%         | 71.00\%          |
> NeuAL-NTK-D | *92.89\%*         | 93.65\%          | *73.80\%*         | 97.70\%          | 90.15\%         | 84.05\%         |
> ALPS        | 91.47\%          | 93.25\%          | 71.45\%         | 95.70\%          | 86.95\%         | 85.40\%          |
> I-NeurAL    |  **94.22\%**  |  **95.75\%**  | **77.45\%**  | **99.15\%**  |  **94.45\%**  |  **89.00\%**  |
>
>
> ---
>
> **Test Accuracy with 10\% label budget**
>
> |   | Phishing     | IJCNN        | Letter          | Fashion       | MNIST        | CIFAR-10    |
> | --- |     --- |     --- |  --- |  ---  | --- | --- |
> |Random        | 93.93\%         | 96.70\%          | 79.70\%          | 97.30\%          | 90.90\%         | 89.00\%     |
> |Margin      | *94.03\%*        | *97.85\%*        |      78.95\%     | 98.20\%         | 93.30\%       | 88.25\%     |
> |NeuAL-NTK-F | 54.69\%         | 87.65\%         | 48.05\%         | 51.30\%        | 51.10\%         | 70.95\%      |
> |NeuAL-NTK-D | 92.99\%          | 96.90\%          | *80.55\%*         | *98.70\%*        | *94.85\%*       | 89.05\%       |
> |ALPS        | 92.89\%          | 96.20\%          | 78.05\%         | 97.50\%          | 93.00\%         | *89.35\%*        |
> | I-NeurAL    | **95.64\%** | **97.90\%** | **83.95\%** | **99.30\%** | **97.20\%** | **91.75\%** |
>
>
> ***
>
>
> **Test Accuracy with 20\% label budget**
>
>
> |  | Phishing     | IJCNN        | Letter          | Fashion       | MNIST        | CIFAR-10    |
> | --- |     --- |     --- |  --- |  ---  | --- | --- |
> |Random        | 93.93\%         | 96.70\%          | 81.90\%          | 98.15\%          | 93.00\%         | 89.50\%          |
> | Margin      | *95.17\%*        | *98.15\%*         |      82.45\%     | 98.65\%         | 95.05\%       | 89.75\%          |
> | NeuAL-NTK-F | 54.69\%         | 89.95\%         | 48.05\%         | 51.30\%        | 51.10\%         | 72.15\%         |
> | NeuAL-NTK-D | 94.98\%          | 97.75\%          | 82.35\%          | *99.30\%*         | 94.30\%         | *90.90\%*        |
> | ALPS        | 94.41\%          | 97.05\%          | *83.20\%*       | 98.30\%          | *94.45\%*       | 88.95\%         |
> | I-NeurAL    | **95.64\%** | **98.35\%** | **84.65\%** | **99.30\%** | **97.95\%** | **91.80\%** |
>
> ***
>
>
>
> **Test Accuracy with 50\% label budget**
>
>
>  |    | Phishing     | IJCNN        | Letter          | Fashion       | MNIST        | CIFAR-10      |
> | --- |     --- |     --- |  --- |  ---  | --- | --- |
> | Random        | 94.98\%         | 97.75\%          | 86.05\%          | 98.95\%          | 96.40\%         | 90.75\%         |
> | Margin      | 95.73\%         | *98.40\%*      |      86.35\%     | 99.05\%         | 96.20\%       | *91.25\%*    |
> | NeuAL-NTK-F | 54.69\%         | 90.85\%         | 48.05\%         | 51.30\%        | 51.10\%         | 72.35\%         |
> | NeuAL-NTK-D | *95.83\%*        | 98.00\%          | 83.35\%          | *99.30\%*       | *97.15\%*       | 90.55\%         |
> | ALPS        | 94.50\%          | 97.80\%          | **87.10\%**       | 99.05\%          | 96.70\%         | 90.85\%          |
> | I-NeurAL    | **96.02\%** | **98.75\%** | *86.05\%* | **99.35\%** | **97.80\%** | **92.30\%**  |

---

> > ### Author Response · Authors · 2022-08-02
> > **Ablation Study for Label Budget (2)**
> >
> > To sum up, I-NeurAL still achieves the best accuracy with different label budgets. With a small amount of label budget (3\%, 10\%), I-NeurAL can make smart decisions to query labels on these instances with big uncertainty and leverage the full feedback to exploit the past knowledge, which enables I-NeurAL to outperform all the baselines.  With the larger label budget (20\%, 50 \%),   all methods have enough labels to train. Thus, the advantages of I-NeurAL are less significant and the gap between I-NeurAL and baselines is decreasing. Nevertheless, I-NeurAl still has the best performance benefiting from smart query choices.

---

> > > ### Comment · Reviewer_d9Zc · 2022-08-03
> > > **Thank you for the new results!**
> > >
> > > Thanks for addressing my concerns on such short notice. I have raised my scores accordingly. I believe a potential area of improvement towards experimental results could be using different model architectures in the experiments conducted. As the paper currently stands, I believe it is technically sound and worth being accepted.

---

> > > > ### Author Response · Authors · 2022-08-07
> > > > **Thanks for Response**
> > > >
> > > > Thanks very mch for the reviewer's suggestions, which is very helpful for improving our paper. We will improve the experiments accordingly regarding using different model architectures. Thanks!

---

### Official Review · Reviewer_KtaT · 2022-07-12

**Rating:** 6
**Confidence:** 4
**Soundness:** 3 good
**Presentation:** 3 good
**Contribution:** 3 good

**Summary:**

This paper presents an active learning algorithm based on contextual bandits for deep learning with regret guarantees. The main contribution of this work lies in certain algorithmic contributions that improve the theoretical guarantees and empirical effectiveness of prior work [1], which was the first work in this line of research. These contributions include improved regret bounds, applicability to k-classification problems (relative to binary restriction in [1]), simplicity of the method (no complex model selection), and practical efficiency (no training from scratch). The authors provide empirical evidence that the proposed approach works better than comparison methods, including the predecessor algorithm of [1].

[1] https://proceedings.neurips.cc/paper/2021/hash/3dcaf04c357c577a857f3ffadc555f9b-Abstract.html

**Questions:**

* Is there any way to mathematically tune the constant \gamma prior to deployment without having to conduct ablation studies? What are the constants c_1 and c_2 used in the algorithm and how should they be set by the practitioner? What were their values for the experiments shown in the paper?
* What are some key novelties relative to the EE-Net paper [2]?
* Given that the work of [1] is restricted to binary classification as the paper claims, how was it applied in practice to the scenarios evaluated in Sec. 5, which include data sets with multiple labels?

**Limitations:**

Yes

**Strengths And Weaknesses:**

## Strengths
* The topic of active learning with provable guarantees, especially for neural networks, is of high importance and interest to the wider ML community
* The contributions and improvements relative to [1] – as outlined in the summary above – are significant
* The techniques and corresponding theoretical analysis seem sound
* Empirical evaluations are presented that support the improved effectiveness of the proposed work over [1] and other state-of-the-art approaches

## Weaknesses
* There is a significant amount of overlap with prior work that is not appropriately cited. Moreover, a large number of the improvements relative to [1] are directly thanks to the work of [2], but this is not made entirely clear. For example, Definition 3.1 appears in the EE-Net paper [2] (top of page 7), but it is presented as a novelty without a citation. Theorem 4.1 is very reminiscent of Theorem 1 of [2]. The removal of the log and input and effective dimension factors and the purposeful avoidance of NTKs appear as Remarks 1 and 2 of [2].
* There is no information regarding the number of trials that the results were averaged over and no error bars are present for the results in Sec. 5
* The parameter \gamma seems to be a very important parameter for the algorithm as it dictates the number of queries that algorithm makes, but an appropriate setting of it requires a hyper-parameter search before deployment. It is not clear how this can be tuned in an application specific basis in active learning. Additionally, the algorithm uses constants c_1 and c_2, but there is no mention of the values used for these constants in the empirical studies conducted. The batch size (b) parameter is also not used in Algorithm 1. It is also not reported what the value of \delta was used for the algorithm

[1] https://proceedings.neurips.cc/paper/2021/hash/3dcaf04c357c577a857f3ffadc555f9b-Abstract.html

[2] https://arxiv.org/pdf/2110.03177.pdf

---

> ### Author Response · Authors · 2022-08-02
> **Response to Reviewer KtaT's Questions**
>
> Thanks for the reviewer's constructive comments and suggestions. We apologize for missing the detailed comparison with EE-Net and left a confusing impression on the reviewer. We've updated the manuscript to add the comparison and citations more properly (we added the difference from EE-Net at end of Definition 3.1 and Remark 4.4 in Section 4). All the reported results in Section 5 are the average of 5 runs described in Appendix 10.
>
>
> **Q1**: Is there any way to mathematically tune the constant $\gamma$ prior to deployment without having to conduct ablation studies? What are the constants $c_1$ and $c_2$ used in the algorithm and how should they be set by the practitioner? What were their values for the experiments shown in the paper?
>
>
> **A1**:
> $\gamma = 1$ is the our theoretical default value, i.e., $|f(x_{t,\widehat{i} }; \theta_{t-1}) -  f(x_{t,i^\circ}; \theta_{t-1})| < 2 \beta_t$, which comes from Lemma 7.3. We add $\gamma$, i.e., $2 \gamma \beta_t$, to enable the practitioner to tune the aggressiveness of I-NeurAL. Since the theoretical default value is very conservative, the practitioner may want it to be more aggressive to query labels by setting $\gamma$ as a larger value. Such hyperparameter is widely used in other active learning methods as well, such as $S$ in [1], $\triangle$  in [3].
> $c_1$ and $c_2$ are set to $1$ in the experiments. We add these two constants to match the theoretical results in Lemma 7.3. $\delta$ is set as $0.1$ for all needed methods. Thanks for the raised questions, we've updated appendix 10 accordingly.
>
>
> **Q2**: What are some key novelties relative to the EE-Net paper [2]?
>
> **A2**:
> Please take a look at the comments ``Comparison with EE-Net (1) (2)'' at the top, where we highlight the differences with and advances over EE-net (including Definition 3.1).
>
> **Q3**: How was [1]  applied in practice to the data sets with multiple labels?
>
> **A3**:
>  We used the method described in Section 2 to build context vectors for each class and consider them as input for all algorithms.
>  Note that the $k$-class vectors can directly be fed into [1].
>  The only adaptation we need to make is the exploration hyper-parameter in the query decision maker of [1] (was 1/2 in Algorithm 1 of [1]). We used the grid-search over (0.1, 0.3, 0.5, 0.8) for this hyper-parameter to find the best performance.
>
>
>
>
>
>
>
>  Reference:
>
>
> [1] Neural Active Learning with Performance Guarantees.
>  https://proceedings.neurips.cc/paper/2021/hash/3dcaf04c357c577a857f3ffadc555f9b-Abstract.html
>
> [2] EE-NET: Exploitation-Exploration Neural
> Networks in Contextual Bandits.
> https://arxiv.org/pdf/2110.03177.pdf
>
> [3] Online Active Learning with Surrogate Loss Functions.
> https://papers.nips.cc/paper/2021/file/c1619d2ad66f7629c12c87fe21d32a58-Paper.pdf

---

> > ### Comment · Reviewer_KtaT · 2022-08-06
> > **Response**
> >
> > Thank you for your response and clarifications. I have raised my score to a 6 in light of the discussions and the revision. In particular, the authors addressed my concerns regarding the novelties relative to EE-Net and their revision contains this updated information. They also clarified the implementation of their method. Additionally, I read the discussion with the other reviewers and found the empirical results with reported test accuracies presented to Reviewer d9Zc compelling. I would encourage the authors to include these new results in the paper in some way for the final submission. Overall, I am supportive of a method that has solid theoretical foundations and seems to perform well in practice for active learning.

---

> > > ### Author Response · Authors · 2022-08-07
> > > **Thanks for Response**
> > >
> > > Thanks very much for Reviewer's efforts on this paper and suggestions. We will include the new results in the final version. Thanks!

---

### Official Review · Reviewer_7AUC · 2022-07-13

**Rating:** 6
**Confidence:** 3
**Soundness:** 4 excellent
**Presentation:** 3 good
**Contribution:** 3 good

**Summary:**

The paper proposes a neural network-based algorithm for active learning in the k-classification setting. The algorithm improves upon previous state of the art by removing a log T factor and the data-dependent complexity term in the regret bound, and using a more efficient method for updating parameters. Under a hard-margin assumption, the algorithm performs as well as the Bayes-optimal classifier in the long run. Finally, experiments show that the algorithm has better empirical performance than baselines.


**Questions:**

Under the overparameterized regime, the width of the network has to be extremely large. It would be helpful if the authors could provide justifications for why this requirement is reasonable and doesn’t impede the usefulness of the guarantees.

**Limitations:**

Yes.

**Strengths And Weaknesses:**

Strengths
- The algorithm’s regret guarantee doesn’t have data-dependent complexity terms that can be large in the worst case, and the parameter update method is more efficient.
- The paper is clearly written; however, I find some paragraphs to be a little repetitive. For example, the comparison with [45] is reiterated several times.

Weaknesses
- My impression is that the algorithm’s use of neural networks for exploration and exploitation is quite similar to EE-net, and given results in [2] and related works on the theory of deep learning, the guarantees are not very surprising.
- See questions.

---

> ### Author Response · Authors · 2022-08-02
> **Response to Reviewer 7AUC's Questions**
>
> Thanks for the reviewer's constructive comments. We apologize for the lack of detailed comparison with EE-Net.
> We sincerely invite the reviewer to examine the comments ``Comparison with EE-Net (1) (2)'' on the top, so the reviewer can have a more clear picture of our contributions.
>
>
> **Q**: Can you provide justifications for why the requirement of width $m$ is reasonable and doesn’t impede the usefulness of the guarantees?
>
> **A1**:
> Thanks for this interesting and inspiring question. To explain the requirement of the width, we have to understand the convergence of over-parameterized neural networks first. Let $m$ be the width and  $\theta \in R^p$ denote the parameters of neural network $f$, where $p > O(m)$.
>
> First, from the expressibility perspective, the more number of parameters $f$ has, the stronger representation power $f$ has, i.e., $f$ can represent more complicated functions if $m$ is large. Therefore, when $m$ is much larger than the number of data instances, there always exists a choice of the parameters that can represent the underlying function in datasets, i.e., training error is zero. This point has been proved by the universal approximation theorem [1]. From the optimization perspective, given a set of data instances, when we map these data instances into a low-dimensional space, we usually cannot find a convex function to represent these data points; However, when we map these data instances into a high-dimension space, it is easy to find convex functions to represent these data points.
> Therefore, in the over-parameterized neural network ($m$ is wide enough), $f$ is almost convex when any $\theta$ is close to the random initialization (Theorem 3, 4 in [2]). Then, one can utilize the classic convex optimization technique to show that GD or SGD can find the global minima.
>
>
> In the regret analysis, we first show that, I-NeurAL always converges on the collected data regardless of how many rounds (how many data points) and how large the input dimensionality is, if $m$ is wide enough.
> Then, we derive a generalization bound to upper bound the regret of a single round, achieving the $\widetilde{O}(\sqrt{T})$ performance guarantee.
> This regret analysis in over-parameterized
> neural networks provides necessary and meaningful insights in understanding the performance of neural
> active learning algorithms.
>
>
> Reference:
>
> [1] Hornik, Kurt; Stinchcombe, Maxwell; White, Halbert (1989). Multilayer Feedforward Networks are Universal Approximators . Neural Networks. Vol. 2. Pergamon Press. pp. 359–366
>
> [2] A Convergence Theory for Deep Learning
> via Over-Parameterization. https://arxiv.org/pdf/1811.03962.pdf.

---

### Author Response · Authors · 2022-08-02
**Comparison with EE-Net (1)**


We sincerely appreciate all the reviewers' constructive comments and suggestions. Since we paid most attention to comparison with [1] while neglecting detailed comparison with EE-Net [2], as pointed out by the reviewers, the current version may not fully capture the contributions of this work. We apologize for this.

Here, we have listed the novel aspects of this paper compared to [2] from both **methodological** and **theoretical** perspectives, to highlight our contributions more clearly. Moreover, we have updated the manuscript accordingly to emphasize the difference (Page 4, Remark 4.4 on Page 7).


### Methodological Perspective:

1. **Input Embedding**. We have a more practical input embedding for the exploration neural network $f_2$ (Definition 3.1).  Given the first neural network $f_1(x; \theta_1)$,  the key component of the input of $f_2$ in [2] (page 7) is the gradient of $f_1$ with respect to the parameters $\theta_1$, denoted by $ \nabla_{\theta_1}f_1(x; \theta_1) \in R^p$ and $p = \Omega(mL)$, where $m$ is the width and $L$ is the depth of the neural network. In contrast, the key component of Definition 3.1 is the gradient of $f_1$ with respect to the arm $x$, denoted by $ \nabla_{x}f_1(x; \theta_1) \in R^d$, where $d$ is dimensionality of arm context $x$. Note that $p$ is usually much larger than $d$ regardless of theory or practice. For example, some popular neural networks like AlexNet that has 61 million parameters, while the input context (like image data) only has several thousand dimensions at the most. Meanwhile,  Definition 3.1 incorporates the information of both $f_1$ and $x$, and it shows good performance in our experiments. Therefore, Definition 3.1 keeps the necessary information while significantly reducing the computation cost, compared to [2].


2. **Query Decision-Maker**. We provide a confidence-aware query decision-maker in active learning. Whether to query a label is a crucial part in active learning, while it is a completely new challenge compared to the bandit setting [2]. Given a confidence level $\delta$, our decision-maker queries the label when it doesn't have enough confidence regarding the predicted label. Otherwise, the decision-maker will leverage the predicted label to further enhance the neural model without querying the label. This important portion is completely new compared to [2].


3. **Optimiser**.  We extend the vanilla gradient descent used in [2] to mini-batch SGD with warm-start for the training of neural networks. In [2], it calculates the gradient of all past data in each iteration and initializes the parameters with random weights (cold start). In contrast, I-Neural supports the mini-batch SGD and initializes the parameters with the parameters of last round (warm start), which is more computationally efficient and can significantly reduce the number of required iterations to converge. This is a non-trivial improvement, because we have to tackle the existing theoretical challenges (e.g., randomness of gradient, convergence with warm-start) to keep the performance guarantee, minimizing the gap between theory and practice.



4. **Full Feedback**. We exploit the full feedback in active learning. In the bandit setting, [2] only exploits the information of selected classes. In contrast, we exploit the information of both selected and unselected classes. This is a subtle but effective idea, while it is overlooked by [1].




Reference:


[1] ``Neural Active Learning with Performance Guarantees".
 https://proceedings.neurips.cc/paper/2021/hash/3dcaf04c357c577a857f3ffadc555f9b-Abstract.html

[2] ``EE-NET: Exploitation-Exploration Neural
Networks in Contextual Bandits".
https://arxiv.org/pdf/2110.03177.pdf

---

> ### Author Response · Authors · 2022-08-02
> **Comparison with EE-Net (2)**
>
> ### Theoretical Perspective:
>
> 1. **New convergence error bound**.  We provide a new convergence error bound supporting mini-batch SGD with warm-start (Lemmas 7.9 and 7.8), which is closer to its usage in practice. In contrast, the convergence bound used in [2] (Lemma C.6 in [2]) only supports vanilla gradient descent with cold-start. To the best of our knowledge, this is the first convergence error bound supporting mini-batch SGD with warm-start.
>
> 2. **Generic Generalization Bound**.  We provide a more generic generalization bound (Lemma 7.3) that holds for every arm (class), as opposed to the bound of [2] (Lemma 5.1 in [2]) which only holds for the selected arm.
>
>
> 3. **Decision-maker Performance Guarantee**. We prove that, in the rounds in which the decision-maker does not query labels, I-NeurAL's predicted labels are correct with high confidence (Lemma 7.5), which inspires our data collection procedure for I-NeurAL (Lines 12-25 in Algorithm 1).
> This is a new additional analysis compared to [2].
>
>
>
> 4. **Label complexity analysis**. We perform the label complexity analysis for I-NeurAL, which is a completely new part compared to [2].
> Furthermore, we derive a label complexity upper bound that is independent of $T$ (Lemma 7.4), which is a novel contribution for both neural active learning [1] and bandit [2] communities.
>
>
> 5.  **Bayes-optimal performance guarantee**. We prove that I-NeurAL can achieve the same performance as the Bayes-optimal classifier after a fixed number of rounds (Theorem 2). This is one of our critical contributions, which is novel for both active learning [1] and bandit [2] communities as well.

---

### Meta-Review · Area_Chair_kcb1 · 2022-08-26

**Recommendation:** Accept
**Confidence:** Certain

**Metareview:**

The paper has two main contributions:

1. It builds upon a recent theoretical work on providing regret bounds for active learning using neural networks. The current paper presents a better regret analysis thereby removing the explicit dependence on the effective dimensionality term and also does not have hard to estimate terms (such as the 'S' term), as was the case in prior work.

2. The paper presents a practical implementation via mini-batch SGD and presents a strong set of empirical results.

All the reviewers agree on the technical novelty and the quality of the experimental work. During the discussion phase the authors also clarified questions regarding the comparison of their results with other prior works. Overall the reviewers agree that this is a submission worthy of acceptance.

**Award:**

No

---

### Decision · Program_Chairs · 2022-09-14

Accept